# Task-Adaptive Parameter-Efficient Fine-Tuning for Weather Foundation Models

**Shilei Cao**[1,*] **Hehai Lin**[2,*] **Jiashun Cheng**[3], **Yang Liu**[4], **Guowen Li**[1] **Xuehe Wang**[1],
**Juepeng Zheng**[1,5,†] **Haoyuan Liang**[1], **Meng Jin**[6], **Chengwei Qin**[2], **Hong Cheng**[4], **Haohuan Fu**[5,7]
[1]Sun Yat-sen University, [2]The Hong Kong University of Science and Technology (Guangzhou)
[3]The Hong Kong University of Science and Technology, [4]The Chinese University of Hong Kong
[5]National Supercomputing Center in Shenzhen, [6]Huawei Technologies Co., Ltd, [7]Tsinghua University

## Abstract

While recent advances in machine learning have equipped Weather Foundation Models (WFMs) with substantial generalization capabilities across diverse downstream tasks, the escalating computational requirements associated with their expanding scale increasingly hinder practical deployment. Current Parameter-Efficient Fine-Tuning (PEFT) methods, designed for vision or language tasks, fail to address the unique challenges of weather downstream tasks, such as variable heterogeneity, resolution diversity, and spatiotemporal coverage variations, leading to suboptimal performance when applied to WFMs. To bridge this gap, we introduce WeatherPEFT, a novel PEFT framework for WFMs incorporating two synergistic innovations. First, during the forward pass, Task-Adaptive Dynamic Prompting (TADP) dynamically injects the embedding weights within the encoder to the input tokens of the pre-trained backbone via internal and external pattern extraction, enabling context-aware feature recalibration for specific downstream tasks. Furthermore, during backpropagation, Stochastic Fisher-Guided Adaptive Selection (SFAS) not only leverages Fisher information to identify and update the most task-critical parameters, thereby preserving invariant pre-trained knowledge, but also introduces randomness to stabilize the selection. We demonstrate the effectiveness and efficiency of WeatherPEFT on three downstream tasks, where existing PEFT methods show significant gaps versus Full-Tuning, and WeatherPEFT achieves performance parity with Full-Tuning using fewer trainable parameters. The code of this work is available at https://github.com/ShileiCao/WeatherPEFT.

## 1 Introduction

In an era marked by intensifying global climate change, the frequency and severity of extreme weather events, such as droughts (Fabian et al., 2023; Deng et al., 2023) and floods (Hirabayashi et al., 2013), have been steadily increasing. Consequently, developing accurate and timely weather modeling systems is crucial for enhancing our understanding of climate change (Beddington et al., 2011; Connor, 2015). For decades, physics-based models (Kimura, 2002; Lynch, 2008; Coiffier, 2011; Bauer et al., 2015; Ravindra et al., 2019) have served as cornerstones for weather research. However, their computational demands, stemming from resolving complex physical constraints, present significant challenges regarding efficiency and scalability (Ren et al., 2021). Over the last decade, the widespread adoption of machine learning models in weather research has led to significant advances in prediction accuracy and computational efficiency (Schultz et al., 2021; Chen et al., 2023c; Shi et al., 2025). Nevertheless, most of these models remain task-specific, requiring bespoke architectures and training protocols for distinct applications, limiting their generalizability.

This limitation has spurred interest in Weather Foundation Models (WFMs), large-scale pre-trained models that leverage massive data to acquire generalized representations of atmospheric processes (Nguyen et al., 2023a; Bodnar et al., 2025; Schmude et al., 2024; Zhao et al., 2024b). Fine-tuning

---

[*]Equal contribution.
[†]Corresponding author: zhengjp8@mail.sysu.edu.cn

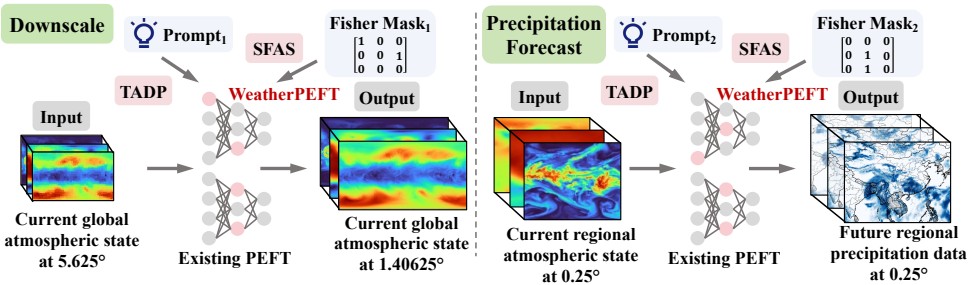

Figure 1: Unlike the uniform adaptation of existing PEFT methods, WeatherPEFT is adaptive to heterogeneous weather tasks like global downscaling (left) and regional precipitation forecasting (right) with Task-Adaptive Dynamic Prompting and Stochastic Fisher-Guided Adaptive Selection.

is then applied to transfer the pre-trained model's knowledge, enabling it to achieve promising performance on downstream tasks. Nevertheless, as the scale of these models increases (Bodnar et al., 2025; Schmude et al., 2024), so too does the challenge of fine-tuning them effectively and efficiently for downstream tasks. Full fine-tuning, which adjusts the entire model per task, is computationally prohibitive due to escalating resource demands. Furthermore, maintaining distinct parameter sets per task creates storage bottlenecks when scaling to large models with multi-task scenarios.

In light of these challenges, Parameter-Efficient Fine-Tuning (PEFT) techniques prevalent in natural language processing and computer vision have shown great promise (Hu et al., 2022; Jia et al., 2022; Zhang et al., 2025), which seeks to potentially match or even exceed the performance of full fine-tuning with a minimal number of trainable parameters updates. These methods not only facilitate more effective adaptation to novel tasks but also preserve the pre-existing knowledge within the foundation models. However, the weather downstream tasks are inherently diverse, encompassing a wide range of objectives. This diversity poses significant challenges when adapting pre-trained models to these tasks, as the varying characteristics of each task make it difficult to apply a one-size-fits-all approach. Unlike the standardized three-channel RGB inputs of vision models or the unified word embedding space of language models, meteorological data involve a wide variety of variables (*e.g.*, temperature and humidity), resolutions (*e.g.*, 1.40625° and 0.25°), and spatiotemporal coverage (*e.g.*, global versus regional) across tasks. First, these variables are distinct physical quantities governed by fluid dynamics equations. Crucially, the correlations between these variables change depending on the task. Second, resolution in weather is not merely a spatial dimension but a physical regime. Changing resolution, *e.g.*, from 5.625° to 0.25°, fundamentally alters the governing physics, transmitting from hydrostatic, large-scale dynamics to non-hydrostatic, convective-scale processes. Third, weather data is inherently spherical and multi-dimensional, often requiring simultaneous reasoning across vast spatiotemporal scales. Tasks at different spatiotemporal scales impose distinctly different demands on the model's feature hierarchies.

These complexities require models to adapt to the varying characteristics inherent in each downstream task. Moreover, a critical limitation of most existing PEFT approaches is their tendency to apply the same set of trainable parameters across different downstream tasks (Figure 1) , which uniformly updates the entire PEFT module across all inputs. These methods fail to account for the fact that different parameters may play varying roles in different tasks. For example, parameters relevant for regional precipitation forecasting may differ from those critical for meteorological downscaling. While task-specific selection methods exist in the broader PEFT literature, they primarily focus on reducing fine-tuning costs in general domains through static selection mechanisms (Xu et al., 2021; Fu et al., 2023; Zhao et al., 2024a). Consequently, they fail to dynamically recalibrate for the complex, variable-specific couplings and physical regime shifts that characterize meteorological data, leading to suboptimal performance.

To fill this gap, we propose WeatherPEFT, a novel PEFT framework for WFMs comprising Task-Adaptive Dynamic Prompting (TADP), which adapts the model's forward pass to task-specific characteristics, and Stochastic Fisher-Guided Adaptive Selection (SFAS), which governs the subsequent parameter updates during backpropagation. Since the encoder's embedding layer captures the task-specific information about input variables, resolutions, and weather phenomena, **TADP** extracts and

integrates this information by transforming its weights into the input token space of the pre-trained backbone. Specifically, TADP first employs three specialized adapters to model the internal patterns within the data dimension. Subsequently, it utilizes self-attention to capture the external patterns by modeling the coupling between physical variables and spatial resolution features, forming a cohesive representation. This dual approach effectively conditions the model on the specific characteristics of the current task. **SFAS** provides a principled approach to identify optimal task-specific parameter subsets, as the relevance and impact of specific parameters can vary significantly across different weather downstream tasks. SFAS utilizes the Fisher information matrix to quantify the sensitivity of parameters to the learning objective. It further integrates an annealed stochastic component to prioritize updates for task-critical parameters with higher possibilities while preserving foundational pre-trained knowledge. The injected randomness serves to stabilize the selection, mitigating the risk of prioritizing parameters influenced by initial noise. Our main contributions are summarized as:

- This work pioneers in exploring generalizing WFMs to downstream tasks. Particularly, we highlight the efficiency issues in tuning WFMs, tackling the diverse demands of weather applications.

- We propose WeatherPEFT, a novel PEFT framework that integrates Task-Adaptive Dynamic Prompting (TADP) and Stochastic Fisher-guided Adaptive Selection (SFAS). TADP utilizes task-related soft prompts extracted from the encoder and SFAS filter task-adaptive parameters based on Fisher information, enabling efficient and adaptive adaptation to weather downstream tasks.

- We evaluate WeatherPEFT on three downstream tasks where existing PEFT methods exhibit a significant performance gap versus Full-Tuning. Our results demonstrate that WeatherPEFT closes this gap, achieving performance on par with Full-Tuning while using fewer trainable parameters. Remarkably, WeatherPEFT outperforms Full-Tuning on regional precipitation forecasting.

## 2 RELATED WORKS

### 2.1 WEATHER FOUNDATION MODELS

The increasing scale of available meteorological data has spurred the application of machine learning (ML) techniques in weather and climate modeling (Shi et al., 2025; Chen et al., 2023c; Schultz et al., 2021; Liu et al., 2025a;b; Zheng et al., 2025). Most notably, several models (Bi et al., 2023; Lam et al., 2023; Chen et al., 2023b;a; Price et al., 2023; Chen et al., 2023b) have demonstrated superior performance in medium-range weather forecasting, surpassing traditional NWPs in terms of accuracy and computational efficiency. Beyond forecasting, ML techniques show promise in various tasks, including bias correction (Gregory et al., 2024; Bretherton et al., 2022), downscaling (Mardani et al., 2024; 2023), data assimilation (Huang et al., 2024; Xiao et al., 2024), and post-processing (Ashkboos et al., 2022; Rasp & Lerch, 2018). Despite these successes, these models are typically designed for specific tasks and often trained on data in particular formats, lacking general-purpose utility for weather and climate modeling.

Foundation Models (FMs) offer a promising solution due to their ability to learn extensive prior knowledge from pre-training on large datasets (Devlin et al., 2019; Brown et al., 2020; Chowdhery et al., 2023; Radford et al., 2021; Yuan et al., 2021; Wang et al., 2023b). Therefore, recent studies have begun exploring WFMs (Bodnar et al., 2025; Nguyen et al., 2023a; Schmude et al., 2024; Zhao et al., 2024b). For instance, Aurora (Bodnar et al., 2025) is pretrained on ten sources of weather datasets and has demonstrated its adaptability to a range of tasks, capable of handling weather data at arbitrary pressure levels for an arbitrary set of variables. Furthermore, Prithvi WxC (Schmude et al., 2024), a 2.3 billion parameter foundation model developed using 160 variables, demonstrates its generalization abilities across a set of challenging downstream tasks. However, as size grows, often encompassing billions of parameters, the computational and storage demands increase substantially. This makes the standard approach of Full-Tuning for each downstream task unsustainable. Therefore, more efficient and resource-saving fine-tuning solutions are urgently needed for WFMs.

### 2.2 PARAMETER-EFFICIENT FINE-TUNING

PEFT has emerged as a promising paradigm for adapting foundation models to novel downstream tasks while maintaining their intrinsic knowledge (Yu et al., 2022; Hu et al., 2022; Zhou et al., 2024; Han et al., 2024; Xin et al., 2024; Zhang et al., 2025; Li & Liang, 2021). Current PEFT can be

broadly categorized into four principal classes: Selective, Additive, Prompt-based, and Reparameterization approaches. Selective PEFT strategically optimizes partial parameter subsets of foundation models (Xu et al., 2021; Zaken et al., 2022; Sung et al., 2021). Additive PEFT incorporates trainable modules into the backbone and only fine-tunes these additional networks (Chen et al., 2023d; Gao et al., 2023). For instance, AdaptFormer (Chen et al., 2022) incorporates a lightweight down-and-up module into the model's backbone. Similarly, SSF (Lian et al., 2022) applies scaling and shifting to the features generated by each layer. Prompt-based PEFT involves learning soft constraints in the input token or the attention layer to adapt models to new tasks like VPT (Jia et al., 2022) and Aprompt (Wang et al., 2023a). Reparameterization PEFT transforms the initial parameters into a low-dimensional representation during training while seamlessly converting the weights back to their original form for inference. LoRA (Hu et al., 2022) is a widely recognized method that decomposes the updated weight into two low-rank matrices, and DoRA (Liu et al., 2024) further advances the decomposition by separating them into a magnitude vector and a direction matrix. However, the inherent heterogeneity of weather downstream tasks, with their varied variables, resolutions, and spatiotemporal coverage, renders conventional homogeneous PEFT approaches suboptimal. While task-specific selection methods exist (Xu et al., 2021; Fu et al., 2023; Zhao et al., 2024a), they primarily focus on reducing fine-tuning costs in general domains, often relying on static selection determined prior to training. These static mechanisms fail to dynamically recalibrate for complex, variable-specific couplings and physical regime shifts inherent in weather tasks. In contrast, WeatherPEFT introduces a dynamic, annealed selection mechanism (SFAS) combined with context-aware dynamic prompting (TADP) to explicitly address the meteorological challenges.

## 3 BACKGROUND AND PRELIMINARIES

**Weather Downstream Tasks.** This work focuses on gridded prediction tasks, which are formalized as spatiotemporal modeling to map input states (historical) to target states (future or derived quantities). Specifically, the input is denoted as a three-dimensional array $\mathbf{X} \in \mathbb{R}^{V \times H \times W}$, where $V$ represents the number of physical variables, such as temperature and geopotential, and $H \times W$ denotes the spatial resolution, determined by how the globe is gridded. The target is to predict an output states $\hat{\mathbf{Y}} \in \mathbb{R}^{\hat{V} \times \hat{H} \times \hat{W}}$. Similarly, $\hat{V}$ and $\hat{H} \times \hat{W}$ are the variables and spatial resolution of the task-dependent output. For example, a global downscaling task involves mapping the $5.625°$ low-resolution data ($32 \times 64$ grid points) to $1.40625°$ high-resolution data ($128 \times 256$ grid points).

**Parameter-Efficient Fine-Tuning.** The foundation model is first pre-trained on extensive source data and then is fine-tuned to perform a variety of downstream tasks $\mathcal{T} = \{\mathcal{T}^i\}_{i=1}^{|\mathcal{T}|}$, where $\mathcal{T}^i = \{(\mathbf{X}_j^i, \mathbf{Y}_j^i)\}_{j=1}^{|\mathcal{T}_i|}$ serves as input-label pairs of each downstream task. Let the pre-trained model $M_\theta$ be parametrized by $\theta$, the goal of fine-tuning is to adapt $\theta$ to different downstream tasks. While the standard full fine-tuning need to update all parameters in $\theta$ to obtain $\theta^i$ for each downstream task $\mathcal{T}^i$, PEFT aims to introduce minimal parameter updates $\Delta\theta^i$ with $|\Delta\theta^i| \ll |\theta^i|$. For each task $\mathcal{T}^i$, the objective is to optimize the task-specific loss $\mathcal{L}^i$ with output $\hat{\mathbf{Y}}_j^i$ from the model $M_{\theta+\Delta\theta_i}$:

$$\min_{\Delta\theta^i} \mathbb{E}_{(\mathbf{X}_j^i, \mathbf{Y}_j^i) \in \mathcal{T}^i} \mathcal{L}^i(M_{\theta+\Delta\theta^i}(\hat{\mathbf{Y}}_j^i | \mathbf{X}_j^i), \mathbf{Y}_j^i). \tag{1}$$

Since our method is applicable to all tasks, we omit task index superscript $^i$ hereafter for simplicity.

## 4 METHODS

Figure 2 presents an overview of the proposed WeatherPEFT, which integrates two synergistic innovations operating at distinct stages of the fine-tuning process. The Task-Adaptive Dynamic Prompting (TADP) makes the model task-aware on the forward pass, while Stochastic Fisher-Guided Adaptive Selection (SFAS) governs the resulting parameter updates during backpropagation.

### 4.1 TASK-ADAPTIVE DYNAMIC PROMPTING

The encoder embedding layer serves as a rich repository of task-specific knowledge, implicitly encoding the distinct characteristics of tasks. To explicitly extract and leverage this information, we

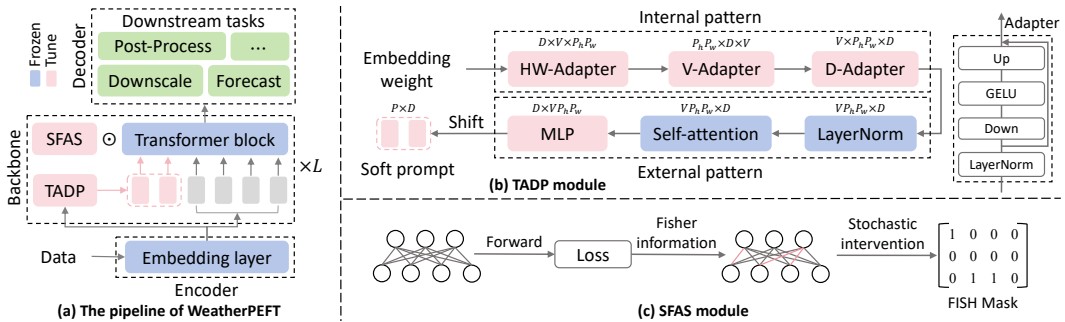

Figure 2: (a) Overview of WeatherPEFT, with TADP and SFAS applied to backbone. (b) TADP generates task-aware prompts by extracting internal and external patterns from the encoder. (c) SFAS uses Fisher information and a stochastic intervention to update task-critical parameters.

propose TADP. This method employs adapters that process the embedding weights to identify both internal and external patterns. These patterns are subsequently used to generate task-aware prompts that condition the forward pass, enabling the model to adapt to specific downstream applications.

**Internal Pattern Extraction.**  The internal patterns within the encoder represent the intrinsic feature learned from data dimensions. The embedding weights $\mathbf{E} \in \mathbb{R}^{D \times V \times P_h \times P_w}$ capture these relationships by mapping the input into tokens, with $P_h \times P_w$ the kernel size involving spatial and resolution information, $V$ the number of variables, $D$ the hidden dimension revealing meteorological characteristics. To harness the patterns, we sequentially extract features using three specialized adapters arranged in a progressive, low-to-high-level hierarchy. Each adapter consists of a Layer-Norm layer, a down-projection layer, a GELU activation, and an up-projection layer. Specifically,

- **HW-Adapter**: We first process the spatial and resolution information ($P_h \times P_w$) that governs localized interactions. The HW-adapter learns patterns from neighboring areas, thereby establishing the fundamental context of how features behave and interact across spatial locations.

- **V-Adapter**: Building upon the spatially-refined features processed by the HW-Adapter, the V-Adapter models the complex interdependencies and relationships among different physical input variables ($V$) such as temperature and humidity, within the established spatial context.

- **D-Adapter**: The D-Adapter processes the abstract attributes represented by the weather characteristics ($D$). It integrates the outputs from the previous spatial and physical processing stages to capture high-level, universal patterns that holistically explain atmospheric response mechanisms.

Formally, we first flatten the spatial dimension of the embedding weights $\mathbf{E}$ to $\hat{\mathbf{E}} \in \mathbb{R}^{D \times V \times P_h P_w}$. Subsequently, $\hat{\mathbf{E}}$ is passed through the adapter sequence to extract the respective internal patterns:

$$\mathbf{E}_{HW} = (\text{Adapter}_{HW}(\hat{\mathbf{E}}))^{\pi}, \quad \mathbf{E}_V = (\text{Adapter}_V(\mathbf{E}_{HW}))^{\pi}, \quad \mathbf{E}_D = \text{Adapter}_D(\mathbf{E}_V), \quad (2)$$

where $\mathbf{E}_{HW} \in \mathbb{R}^{P_h P_w \times D \times V}$, $\mathbf{E}_V \in \mathbb{R}^{V \times P_h P_w \times D}$, and $\mathbf{E}_D \in \mathbb{R}^{V \times P_h P_w \times D}$ are the respective outputs of adapters, and $^{\pi}$ denotes an operation that shifts the last dimension of a tensor to the first.

**External Pattern Integration.**  The next step involves integrating the patterns to form a cohesive, task-specific representation. To achieve this, we capture external patterns by establishing a coupling analysis between the physical quantities ($V$) and spatial resolution features ($P_h P_w$). We first merge the first two dimension of $\mathbf{E}_D$ to $\hat{\mathbf{E}}_D \in \mathbb{R}^{V P_h P_w \times D}$ and then apply the self-attention operation $\text{SA}(\cdot)$ to $\hat{\mathbf{E}}_D$, followed by a linear projection to generate the final soft prompt tokens $\mathbf{E}_P$:

$$\text{SA}(\cdot) = \text{Softmax}(\frac{\mathbf{E}_{query}\mathbf{E}_{key}}{\sqrt{D}})\mathbf{E}_{value}, \quad \mathbf{E}_{SA} = (\text{SA}(\hat{\mathbf{E}}_D))^{\pi}, \quad \mathbf{E}_P = (\text{MLP}(\mathbf{E}_{SA}))^{\pi}, \quad (3)$$

where $\mathbf{E}_{SA} \in \mathbb{R}^{D \times V P_h P_w}$, $\mathbf{E}_P \in \mathbb{R}^{P \times D}$, $P$ is the prompt length, and $\mathbf{E}_{query}, \mathbf{E}_{key}, \mathbf{E}_{value}$ are the query, key, and value, respectively. Specifically, the final step is to inject these task-adaptive prompt tokens into the backbone. The input $\mathbf{X}$ is first encoded into a sequence of $M$ tokens $\mathbf{T} \in \mathbb{R}^{M \times D}$ by

the encoder. The generated soft prompt tokens $\mathbf{E}_P$ are then concatenated with the input tokens $\mathbf{T}$ before being fed into each block of the pretrained backbone. This ensures that the model processes the input data in the context of the task-specific information at every stage of computation.

## 4.2 STOCHASTIC FISHER-GUIDED ADAPTIVE SELECTION

The diversity of weather downstream tasks implies that parameters are not uniformly relevant across all applications. Some parameters may encode chaotic patterns for precipitation forecasting, while some focus on spatial relationships for downscaling. Consequently, we propose SFAS that adopts the Fisher information (Kirkpatrick et al., 2017) as the metric to update the task-critical parameters.

A parameter's significance can be determined by evaluating the extent to which altering the parameter influences the output. Consider a model parameterized by $\theta \in \mathbb{R}^{|\theta|}$ that defines a predictive distribution $P_\theta(\mathbf{Y}|\mathbf{X})$ with input $\mathbf{X}$. The sensitivity of this distribution to a small parameter perturbation $\delta \in \mathbb{R}^{|\theta|}$ can be measured using the Kullback-Leibler divergence $\mathrm{D}_{KL}(P_\theta(\mathbf{Y}|\mathbf{X}) \parallel P_{\theta+\delta}(\mathbf{Y}|\mathbf{X}))$. Abbass et al. (2022); Sung et al. (2021) shows that as $\delta \to 0$, the following relationship holds:

$$\mathbb{E}_{\mathbf{X}}\left[\mathrm{D}_{KL}(P_\theta(\mathbf{Y}|\mathbf{X}) \parallel P_{\theta+\delta}(\mathbf{Y}|\mathbf{X}))\right] = \delta^T F_\theta \delta + O(\delta^3), \tag{4}$$

where $F_\theta \in \mathbb{R}^{|\theta| \times |\theta|}$ is the Fisher information matrix (Fisher, 1922), defined as:

$$F_\theta = \mathbb{E}_{\mathbf{X}}\left[\mathbb{E}_{\mathbf{Y} \sim P_\theta(\mathbf{Y}|\mathbf{X})} \nabla_\theta \log P_\theta(\mathbf{Y}|\mathbf{X}) \nabla_\theta \log P_\theta(\mathbf{Y}|\mathbf{X})^T\right]. \tag{5}$$

Evidently, the Fisher information matrix is intrinsically linked to the change in parameters induced by the small perturbation $\delta$. Therefore, we leverage Fisher information to guide the adaptive parameter selection process. However, the $|\theta| \times |\theta|$ size of $F_\theta$ renders it computationally infeasible to compute the Fisher information matrix exactly in practice. Consequently, prior work often approximates $F_\theta$ with its diagonal matrix, or equivalently, as a vector in $\mathbb{R}^{|\theta|}$. Especially, when we sample $N$ data $\mathbf{X}_1, \mathbf{X}_2, ..., \mathbf{X}_N$ from data distribution $P(\mathbf{X})$, Eq. 5 can be effectively approximated as:

$$\hat{F}_\theta = \frac{1}{N} \sum_{j=1}^{N} \mathbb{E}_{\mathbf{Y} \sim P_\theta(\mathbf{Y}|\mathbf{X}_j)}(\nabla_\theta \log P_\theta(\mathbf{Y}|\mathbf{X}_j))^2. \tag{6}$$

Here $\hat{F}_\theta \in \mathbb{R}^{|\theta|}$ and Eq. 6 demonstrates that a larger $\hat{F}_\theta$ corresponds to a more influential parameter. Furthermore, in a supervised learning framework, we have the data pairs $(\mathbf{X}_j, \mathbf{Y}_j)$ and can access the ground-truth label $\mathbf{Y}_j$ for each $\mathbf{X}_j$. So we can approximate Eq. 6 as:

$$\hat{F}_\theta = \frac{1}{N} \sum_{j=1}^{N}(\nabla_\theta \log P_\theta(\mathbf{Y}_j|\mathbf{X}_j))^2. \tag{7}$$

This approximation improves computational efficiency and performance. However, due to the significant heterogeneity among weather downstream tasks, substantial noise exists during early fine-tuning, distorting Fisher information. For example, in the early epochs, parameters with high Fisher scores may capture transient noise artifacts rather than task-relevant features. To stabilize the training process, we introduce an annealed stochastic component with a linear decay factor:

$$\bar{F}_\theta = \gamma \times (1 - \frac{ns}{ts}) \odot M_{sc} + \hat{F}_\theta, \tag{8}$$

where $\gamma$ is the initial factor, $M_{sc} \sim \mathrm{Uniform}(0, 1)$ is the stochastic vector, and $ns$ and $ts$ are the current step and total step respectively. Each batch is treated as a step, and during training we select the Top-$k$ parameters with the highest $\bar{F}_\theta$ for optimization. The hyperparameter $k$ governs the sparsity of the Fish Mask. The Fish Mask entries for the Top-$k$ parameters are set to one, while the rest are zero, thereby excluding less significant parameters and updating only the Top-$k$ parameters.

## 5 EXPERIMENTS

We evaluate WeatherPEFT on downscaling, ensemble forecast post-processing, and regional precipitation prediction. These tasks are selected to span diverse weather challenges, including variable

Table 1: The RMSE and Mean Bias on downscaling experiments from ERA5 (5.625°) to ERA5 (1.40625°). We adopt the Aurora (Bodnar et al., 2025) as the foundation model and only count the trainable parameters in the backbone for all fine-tuning methods.

| Method | Trainable Params (M) | T2m RMSE | T2m Mean Bias | U10 RMSE | U10 Mean Bias | V10 RMSE | V10 Mean Bias | T850 RMSE | T850 Mean Bias | Z500 RMSE | Z500 Mean Bias |
|---|---|---|---|---|---|---|---|---|---|---|---|
| Nearest | 0.00 | 3.007 | **0.001** | 2.695 | -0.039 | 2.717 | 0.038 | 2.010 | 0.007 | 295.493 | **-0.054** |
| Bilinear | 0.00 | 2.284 | **0.001** | 2.118 | -0.038 | 2.176 | 0.038 | 1.439 | 0.007 | 149.662 | **-0.053** |
| U-Net | 20.10 | 1.915 | -0.111 | 1.174 | 0.031 | 1.152 | -0.033 | 1.773 | -0.059 | 120.045 | -11.118 |
| ResNet | 34.78 | 2.164 | 0.095 | 1.562 | -0.087 | 1.513 | 0.013 | 1.513 | -0.067 | 105.101 | 10.229 |
| ViT | 315.43 | 2.972 | 0.018 | 1.931 | -0.024 | 1.837 | 0.006 | 2.143 | -0.218 | 201.027 | -27.900 |
| FourCastNet | 63.53 | 2.036 | -0.016 | 1.535 | -0.001 | 1.492 | -0.003 | 1.494 | -0.032 | 160.271 | -4.184 |
| ClimaX | 116.65 | 2.512 | -0.043 | 1.691 | 0.005 | 1.649 | 0.009 | 2.000 | -0.102 | 163.806 | -12.55 |
| Aurora | 1256.27 | 1.227 | 0.006 | 1.126 | 0.006 | 1.134 | -0.012 | 1.192 | 0.002 | 99.764 | -0.996 |
| Linear-Probing | 0.00 | 1.291 | 0.014 | 1.227 | -0.002 | 1.198 | 0.003 | 1.078 | 0.002 | 58.085 | 0.598 |
| Bias-Tuning | 0.78 | 1.242 | 0.013 | 1.168 | -0.003 | 1.148 | **0.000** | 1.026 | 0.004 | 53.049 | 0.108 |
| LoRA | 3.63 | 1.190 | 0.006 | 1.130 | **0.000** | 1.118 | -0.002 | 0.998 | -0.001 | 50.421 | 0.084 |
| DoRA | 3.75 | 1.228 | 0.010 | 1.140 | 0.001 | 1.120 | -0.001 | 1.024 | **0.000** | 50.061 | 0.984 |
| AdaptFormer | 4.64 | 1.737 | -0.065 | 1.505 | -0.050 | 1.412 | 0.002 | 1.429 | -0.083 | 106.667 | -21.029 |
| SSF | 3.92 | 1.180 | 0.009 | 1.106 | -0.001 | 1.094 | -0.001 | 0.987 | 0.002 | 48.342 | 0.936 |
| VPT | 3.75 | 1.241 | 0.008 | 1.163 | -0.002 | 1.144 | 0.001 | 1.031 | 0.005 | 52.453 | 0.998 |
| APrompt | 4.34 | 1.228 | 0.010 | 1.151 | -0.002 | 1.132 | **0.000** | 1.025 | 0.008 | 51.587 | 1.099 |
| TADP Only | 2.22 | 1.183 | 0.005 | 1.118 | **0.000** | 1.105 | -0.001 | 0.996 | 0.003 | 49.809 | 1.491 |
| SFAS Only | 1.26 | 1.161 | 0.010 | 1.090 | -0.001 | 1.081 | -0.002 | 0.973 | 0.002 | 47.000 | 0.848 |
| WeatherPEFT | 3.48 | **1.119** | 0.003 | **1.057** | **0.000** | **1.051** | -0.001 | **0.950** | 0.004 | **44.922** | 0.413 |
| Full-Tuning | 1239.94 | **0.906** | 0.002 | 0.882 | **0.000** | 0.884 | -0.001 | 0.836 | **0.000** | 35.821 | **0.314** |
| LoRA | 57.80 | 1.131 | 0.004 | 1.069 | 0.001 | 1.060 | 0.001 | 0.961 | 0.004 | 45.914 | 1.110 |
| DoRA | 57.92 | 1.236 | 0.009 | 1.147 | -0.002 | 1.126 | -0.001 | 1.030 | -0.001 | 50.405 | 1.289 |
| AdaptFormer | 61.68 | 1.590 | -0.007 | 1.376 | -0.012 | 1.331 | -0.003 | 1.282 | 0.006 | 81.465 | 1.739 |
| WeatherPEFT | 52.47 | **0.916** | **0.000** | **0.873** | -0.001 | **0.875** | -0.002 | **0.834** | -0.002 | **35.076** | 0.504 |

variations, resolution shifts, and spatiotemporal coverage heterogeneity. Additional ablation studies on hyperparameters and fine-grained comparisons are provided in Appendix B.1 and B.3.

**Implementation Details.** We mainly leverage Aurora (Bodnar et al., 2025), a 1.3B-parameter pre-trained foundation model with a 3D Swin Transformer U-Net backbone for the fine-tuning experiments. We also evaluate our method on another larger backbone, Prithvi-WxC (Schmude et al., 2024), provided in Appendix B.2. More experimental settings will be discussed in Appendix F.1.

**Baselines.** Generally, we adopt three types of baselines. **Firstly**, we include models trained from scratch from vision and weather domains, *i.e.*, U-Net (Ronneberger et al., 2015), ResNet (He et al., 2016), and ViT (Dosovitskiy et al., 2020), FourCastNet (Pathak et al., 2022), ClimaX (Nguyen et al., 2023a), and Aurora (Bodnar et al., 2025). This comparison helps to highlight the advantages of fine-tuning over training from the ground up. **Secondly**, to demonstrate the efficiency of PEFT, we select three conventional fine-tuning approaches, including Linear-Probing, Bias-Tuning, and Full-Tuning. **Thirdly**, we chose six state-of-the-art PEFT methods, including LoRA (Hu et al., 2022), DoRA (Liu et al., 2024), AdaptFormer (Chen et al., 2022), SSF (Lian et al., 2022), VPT (Jia et al., 2022), APrompt (Wang et al., 2023a). The architectural details are provided in the Appendix E.

## 5.1 DOWNSCALING

Downscaling, the process of mapping coarse-resolution data to a higher resolution, is critical for analyzing local phenomena. In this experiment, we downscale 5.625° ERA5 data to 1.40625° ERA5 data (Hersbach et al., 2020) globally with WeatherBench dataset (Rasp et al., 2020). We simultaneously downscale the 68 atmospheric input variables to test the model's ability to learn the cross-variable interactions required for accurate high-resolution outputs. Additionally, we compare WeatherPEFT with nearest and bilinear interpolation. We evaluate all methods on latitude-weighted Root Mean Squared Error (RMSE) and Mean Bias, which are common metrics in downscaling works (Nguyen et al., 2023b). We select 2-meter temperature ($T2m$), 10-meter zonal wind ($U10$), 10-meter meridional wind ($V10$), 500 hPa geopotential ($Z500$), and 850 hPa temperature ($T850$) as the primary verification fields as they collectively ensure a holistic evaluation of model performance (Rasp et al., 2020). Details of the task configurations and metrics are in the Appendix F.2.

Visualizations are included in Appendix F.2.3. Table 1 shows downscaling results, indicating that

- Models trained from scratch generally exhibit poorer performance compared to fine-tuning approaches. For example, Aurora achieves an RMSE of 1.227 for T2m, which is significantly worse

Table 2: The CRPS and EECRPS on ensemble weather forecast post-processing with ten ensemble members. We adopt the Aurora (Bodnar et al., 2025) as the foundation model.

| Method | Trainable Params (M) | T2m CRPS | T2m EECRPS | U10 CRPS | U10 EECRPS | V10 CRPS | V10 EECRPS | T850 CRPS | T850 EECRPS | Z500 CRPS | Z500 EECRPS |
|---|---|---|---|---|---|---|---|---|---|---|---|
| RAW | 0.00 | 0.732 | 0.250 | 0.889 | 0.304 | 0.899 | 0.304 | 0.719 | 0.246 | 78.222 | 28.766 |
| U-Net | 19.88 | 0.661 | 0.226 | 0.859 | 0.292 | 0.872 | 0.292 | 0.672 | 0.230 | 74.158 | 27.260 |
| ResNet | 33.95 | 0.682 | 0.232 | 0.865 | 0.294 | 0.880 | 0.295 | 0.689 | 0.235 | 75.562 | 27.750 |
| ViT | 311.10 | 0.646 | 0.221 | 0.856 | 0.291 | 0.872 | 0.292 | 0.672 | 0.229 | 73.503 | 26.956 |
| FourCastNet | 73.56 | 0.679 | 0.231 | 0.859 | 0.291 | 0.872 | 0.292 | 0.687 | 0.234 | 74.552 | 27.342 |
| ClimaX | 114.55 | 0.636 | 0.217 | 0.854 | 0.290 | 0.870 | 0.292 | 0.669 | 0.229 | 72.916 | 26.751 |
| Aurora | 1256.46 | 0.619 | **0.211** | 0.847 | 0.287 | 0.863 | 0.288 | 0.662 | 0.226 | 80.852 | 29.616 |
| Linear-Probing | 0.00 | 0.649 | 0.222 | 0.850 | 0.288 | 0.866 | 0.290 | 0.662 | 0.226 | 73.151 | 26.847 |
| Bias-Tuning | 0.78 | 0.644 | 0.220 | 0.849 | 0.288 | 0.865 | 0.290 | 0.661 | 0.226 | 73.009 | 26.827 |
| LoRA | 3.63 | 0.637 | 0.218 | 0.849 | 0.288 | 0.865 | 0.289 | 0.661 | 0.226 | 72.798 | 26.719 |
| DoRA | 3.75 | 0.638 | 0.218 | 0.847 | 0.287 | 0.864 | 0.289 | 0.660 | 0.225 | 72.827 | 26.735 |
| AdaptFormer | 4.64 | 0.647 | 0.221 | 0.862 | 0.294 | 0.878 | 0.295 | 0.666 | 0.227 | 73.312 | 26.869 |
| SSF | 3.92 | 0.629 | 0.215 | 0.847 | 0.287 | 0.862 | 0.289 | 0.659 | 0.225 | 73.025 | 26.832 |
| VPT | 3.75 | 0.635 | 0.217 | 0.846 | 0.287 | 0.862 | 0.288 | 0.659 | 0.225 | 72.883 | 26.774 |
| APrompt | 4.34 | 0.632 | 0.216 | 0.846 | 0.287 | 0.862 | 0.288 | 0.660 | 0.225 | 73.022 | 26.820 |
| TADP Only | 1.92 | 0.632 | 0.216 | 0.848 | 0.288 | 0.863 | 0.289 | 0.659 | 0.226 | 72.715 | 26.731 |
| SFAS Only | 1.26 | 0.629 | 0.215 | 0.849 | 0.288 | 0.864 | 0.289 | 0.660 | 0.226 | 72.716 | 26.715 |
| WeatherPEFT | 3.18 | **0.618** | **0.211** | **0.844** | **0.286** | **0.860** | **0.287** | **0.657** | **0.224** | **72.701** | **26.665** |
| Full-Tuning | 1239.94 | 0.604 | 0.206 | **0.838** | **0.284** | **0.854** | **0.285** | 0.653 | 0.223 | 73.760 | 27.051 |
| LoRA | 57.80 | 0.630 | 0.215 | 0.847 | 0.287 | 0.862 | 0.288 | 0.66 | 0.225 | 72.805 | 26.710 |
| DoRA | 57.92 | 0.631 | 0.216 | 0.845 | 0.287 | 0.861 | 0.288 | 0.659 | 0.225 | 72.987 | 26.779 |
| AdaptFormer | 61.68 | 0.638 | 0.218 | 0.860 | 0.293 | 0.874 | 0.293 | 0.662 | 0.226 | 73.114 | 26.815 |
| WeatherPEFT | **52.18** | **0.601** | **0.205** | **0.838** | **0.284** | **0.854** | 0.286 | **0.650** | **0.222** | 72.745 | 26.683 |

than the 0.906 RMSE of Full-Tuning. This performance gap arises from the task's nature, which necessitates simultaneous downscaling of 68 variables, posing significant challenges for models trained from scratch to effectively capture the complex interdependencies among these variables.

- While the PEFT methods significantly reduce trainable parameters, they incur a certain degree of accuracy degradation compared to Full-Tuning. For example, DoRA shows ∼36% higher T2m RMSE compared to Full-Tuning with only 3.75M parameters (1.228 vs. 0.906). These results underscore the limitations of existing PEFT strategies in specialized scientific domains. Notably, WeatherPEFT effectively balances parameter efficiency and performance, outperforming existing PEFT baselines in terms of RMSE using the fewest parameters, with only 3.48M parameters, demonstrating its ability to adapt the foundation model to the task of downscaling.

- The ablation study provides further evidence of the effectiveness of our framework. TADP and SFAS individually perform well but slightly underperform versus the full WeatherPEFT, underscoring the synergistic benefits of both modules during the forward and backpropagation passes.

- To ensure a comprehensive and fair comparison, we also evaluated the PEFT methods with an increased parameter budget (∼4%). Even in this setting, existing PEFT methods like LoRA and DoRA still fail to approach the performance of Full-Tuning. Remarkably, WeatherPEFT nearly closes the gap, achieving results nearly on par with, and in some cases better than, the Full-Tuning.

## 5.2 ENSEMBLE WEATHER FORECAST POST-PROCESSING

Existing ensemble weather predictions have biases (Toth & Kalnay, 1993), prompting post-processing methods to improve forecast reliability by correcting prediction distributions. Our evaluation uses the ENS-10 benchmark (Ashkboos et al., 2022), which pairs 10-member ECMWF IFS (ECMWF, 2022) ensemble predictions with ERA5 targets at $0.5°$ resolution. The dataset includes 25 surface and atmospheric variables. An additional baseline ('RAW') is included, which refers to using the raw ensemble mean and standard deviation. Performance is quantified using the Continuous Ranked Probability Score (CRPS) and Extreme Event Weighted Continuous Ranked Probability Score (EECRPS) (Ashkboos et al., 2022). We train the model to simultaneously correct the five same target variables as Section 5.1. Implementation specifics are included in the Appendix F.3.

Table 2 presents the results of post-processing across five target variables, indicating that

- Unlike the downscaling task, the performance gap between Full-Tuning and training-from-scratch baselines narrows in the post-processing task. For example, ClimaX achieves a Z500 CRPS of 72.916, marginally better than Full-Tuning's 73.760. This might suggest a significant task shift

Table 3: The SEEPS, ACC, RMSE (1e-2) on regional precipitation forecasting, focusing on China region. We adopt the Aurora (Bodnar et al., 2025) as the foundation model and only count the trainable parameters in the backbone for all fine-tuning methods.

| Method | Trainable Params (M) | 12 Hours SEEPS | ACC ↑ | RMSE | 24 Hours SEEPS | ACC ↑ | RMSE | 36 Hours SEEPS | ACC ↑ | RMSE |
|---|---|---|---|---|---|---|---|---|---|---|
| Persistence | 0.00 | 0.695 | 0.265 | 0.371 | 0.720 | 0.168 | 0.387 | 0.855 | 0.088 | 0.416 |
| U-Net | 19.89 | 0.467 | 0.639 | 0.225 | 0.591 | 0.468 | 0.263 | 0.685 | 0.352 | 0.281 |
| ResNet | 33.99 | 0.551 | 0.499 | 0.259 | 0.664 | 0.342 | 0.283 | 0.767 | 0.210 | 0.300 |
| ViT | 311.30 | 0.560 | 0.499 | 0.257 | 0.646 | 0.389 | 0.276 | 0.717 | 0.292 | 0.290 |
| FourCastNet | 63.94 | 0.640 | 0.376 | 0.279 | 0.756 | 0.213 | 0.299 | 0.824 | 0.126 | 0.310 |
| ClimaX | 117.32 | 0.590 | 0.487 | 0.260 | 0.695 | 0.328 | 0.285 | 0.759 | 0.231 | 0.297 |
| Aurora | 1239.94 | 0.470 | 0.589 | 0.241 | 0.578 | 0.449 | 0.268 | 0.660 | 0.351 | 0.283 |
| Linear-Probing | 0.00 | 0.581 | 0.464 | 0.266 | 0.720 | 0.265 | 0.293 | 0.790 | 0.171 | 0.303 |
| Bias-Tuning | 0.78 | 0.573 | 0.474 | 0.265 | 0.715 | 0.271 | 0.292 | 0.783 | 0.177 | 0.302 |
| LoRA | 3.63 | 0.495 | 0.592 | 0.24 | 0.634 | 0.415 | 0.273 | 0.723 | 0.294 | 0.289 |
| DoRA | 3.75 | 0.513 | 0.574 | 0.244 | 0.662 | 0.372 | 0.279 | 0.748 | 0.246 | 0.294 |
| AdaptFormer | 4.62 | 0.499 | 0.577 | 0.243 | 0.643 | 0.378 | 0.278 | 0.731 | 0.258 | 0.293 |
| SSF | 3.92 | 0.459 | 0.631 | 0.231 | 0.588 | 0.474 | 0.264 | 0.680 | 0.356 | 0.281 |
| VPT | 3.75 | 0.522 | 0.550 | 0.25 | 0.666 | 0.356 | 0.281 | 0.750 | 0.235 | 0.296 |
| APrompt | 4.34 | 0.521 | 0.554 | 0.249 | 0.650 | 0.387 | 0.277 | 0.733 | 0.271 | 0.292 |
| Covpass | 4.92 | 0.485 | 0.606 | 0.237 | 0.615 | 0.439 | 0.269 | 0.697 | 0.326 | 0.285 |
| FacT-TT | 2.73 | 0.525 | 0.553 | 0.249 | 0.662 | 0.371 | 0.279 | 0.747 | 0.246 | 0.294 |
| RepAdapter | 3.75 | 0.534 | 0.532 | 0.254 | 0.675 | 0.340 | 0.283 | 0.757 | 0.222 | 0.297 |
| SCT | 3.94 | 0.481 | 0.607 | 0.237 | 0.616 | 0.439 | 0.269 | 0.706 | 0.316 | 0.286 |
| Child-Tuning$_D$ | 3.39 | 0.407 | 0.694 | 0.214 | 0.565 | 0.500 | 0.259 | 0.672 | 0.364 | 0.280 |
| MoA | 8.62 | 0.515 | 0.563 | 0.246 | 0.665 | 0.354 | 0.281 | 0.749 | 0.235 | 0.296 |
| HydraLoRA | 5.77 | 0.510 | 0.571 | 0.245 | 0.650 | 0.393 | 0.276 | 0.734 | 0.268 | 0.292 |
| VeRA | 0.98 | 0.524 | 0.551 | 0.250 | 0.663 | 0.365 | 0.280 | 0.744 | 0.256 | 0.293 |
| SAM | 3.39 | 0.421 | 0.673 | 0.220 | 0.598 | 0.457 | 0.267 | 0.704 | 0.299 | 0.289 |
| TADP Only | 2.12 | 0.549 | 0.523 | 0.256 | 0.676 | 0.357 | 0.282 | 0.750 | 0.247 | 0.295 |
| SFAS Only | 1.26 | 0.459 | 0.634 | 0.231 | 0.612 | 0.443 | 0.269 | 0.716 | 0.294 | 0.289 |
| WeatherPEFT | 3.38 | **0.368** | **0.742** | **0.198** | **0.515** | **0.559** | **0.247** | **0.615** | **0.443** | **0.268** |
| Full-Tuning | 1246.77 | 0.304 | 0.797 | 0.178 | 0.452 | 0.586 | 0.241 | 0.542 | 0.481 | 0.263 |
| LoRA | 57.80 | 0.449 | 0.648 | 0.226 | 0.59 | 0.474 | 0.263 | 0.681 | 0.353 | 0.282 |
| DoRA | 57.92 | 0.512 | 0.576 | 0.244 | 0.659 | 0.383 | 0.277 | 0.746 | 0.254 | 0.293 |
| AdaptFormer | 61.68 | 0.458 | 0.623 | 0.232 | 0.599 | 0.438 | 0.269 | 0.691 | 0.324 | 0.286 |
| WeatherPEFT | 52.37 | **0.302** | **0.805** | **0.174** | **0.437** | **0.615** | **0.235** | **0.526** | **0.518** | **0.256** |

between the pre-training objectives and the probabilistic correction required for post-processing, which could hinder the transfer of knowledge learned during the pre-training phase.

- While PEFT methods such as SSF demonstrate competitive results, they still lag behind Full-Tuning. Despite the challenging task shift, WeatherPEFT achieves near-Full-Tuning performance with only 3.18M parameters. Especially on Z500, WeatherPEFT outperforms Full-Tuning (72.701 vs. 73.760 CRPS and 26.665 vs. 27.051 EECRPS). This result suggests that WeatherPEFT is capable of handling the specific challenges posed by this post-processing task, even when the pre-training knowledge does not directly align with the task's variable characteristics.

- Furthermore, the ablation study demonstrates the importance of combining both modules, which synergistically to adapt the foundation model's parameters to the specific task at hand.

- Similarly, the results in the increased parameter setting further underscore our method's superiority. WeatherPEFT, with 52.18M parameters, not only exceeds the performance of its PEFT counterparts but also surpasses the 1.2B Full-Tuning method across most key metrics.

## 5.3 REGIONAL PRECIPITATION FORECASTING

Precipitation forecasting is vital for agriculture, water management, and disaster prevention. However, global predictions are often unfeasible, especially with only regional data available. To address this, we formulate a regional precipitation forecasting task to predict the future six-hour accumulation of total precipitation (TP-6hr) based on the regional weather conditions. For this task, we introduce a new dataset ERA5-CH from the ERA5 data at $0.25°$, which includes five surface variables and five upper variables but focuses exclusively on the China region. Following WeatherBench2 (Rasp et al., 2024), we employ the latitude-weighted Stable Equitable Error in Probability Space (SEEPS) (Rodwell et al., 2010), Anomaly Correlation Coefficient(ACC), and RMSE as the evaluation metrics. Specifically, we focus on short-term forecasting with lead times of 12, 24, and 36 Hours. "Persistence" represents utilizing the input as the prediction. Complete experimental details are listed in Appendix F.4, and a case study on extreme precipitation is presented in Appendix B.4.

To rigorously evaluate WeatherPEFT, we include an expanded suite of PEFT baselines, including vision PEFTs (ConvPass (Jie et al., 2024), FacT (Jie & Deng, 2023), RepAdapter (Luo et al., 2023)) and task-selective methods (SCT (Zhao et al., 2024a), Child-Tuning (Xu et al., 2021), SAM (Fu et al., 2023)), LoRA variants (HydraLoRA (Tian et al., 2024), VeRA (Kopiczko et al., 2024)), and Mixture of Adapter (MOA). Table 3 presents the following results of precipitation forecasting:

- Full-Tuning significantly achieves superior performance over training-from-scratch models, confirming that knowledge transfer from pre-training is highly effective for this task.

- Moreover, standard PEFT methods show significant gaps versus Full-Tuning. For example, LoRA's 12h SEEPS is 62.8% higher than Full-Tuning, indicating poorer calibration of rainfall events. This underperformance is due to the unique challenges of precipitation, including its sparse nature and highly localized patterns, which conventional PEFT methods fail to adequately capture. In contrast, the WeatherPEFT significantly surpasses PEFT baselines, and significantly narrows the gap with Full-Tuning when constrained to a minimal parameter budget ($\sim$0.3%).

- Task-adaptive selection methods (SCT, SAM, Child-Tuning$_D$) consistently outperform other baselines like LoRA. This validates the intuition that selecting task-relevant parameters is crucial for heterogeneous weather tasks. Despite these improvements, WeatherPEFT significantly surpasses all competitors. This confirms that adaptivity alone is insufficient and coupling it with the domain-specific context awareness provided by TADP is essential for meteorological adaptation.

- The ablation experiments provide insights into the effectiveness of the two components in WeatherPEFT, indicating that SFAS is more critical than prompting for precipitation's sparse signals.

- Despite the increased trainable parameters, PEFT baselines' performance improves marginally but remains inferior to Full-Tuning. Notably, WeatherPEFT, with $\sim$4% parameters, even surpasses the performance of Full-Tuning across all metrics. This demonstrates that our method is not only more efficient but also more effective at adapting the foundation model for this complex task.

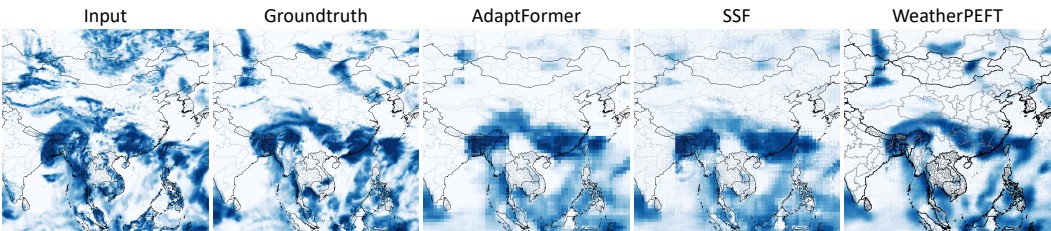

Figure 3: Visualization of a 12-hour forecast for TP-6hr over China (2020-05-20 12 UTC).

**Visualization** We visualize the input, ground truth, and prediction of AdaptFormer, SSF, and WeatherPEFT in Figure 3 to provide an intuitive comparison. The complete visualization of PEFT methods is provided in the Appendix F.4.3. It distinctly reveals that deep learning approaches employing pixel-wise MAE loss exhibit over-smoothed characteristics in their precipitation predictions, which are particularly noticeable in their failure to preserve fine-grained spatial patterns. However, our proposed WeatherPEFT demonstrates superior alignment with the ground truth compared to other PEFT baselines, highlighting the importance of WeatherPEFT's task-adaptive feature.

## 6  CONCLUSION

This paper proposes WeatherPEFT, the first exploration of efficient fine-tuning for weather foundation models. WeatherPEFT is a novel PEFT framework that integrates two synergetic modules, *i.e.*, Task-Adaptive Dynamic Prompting (TADP) and Stochastic Fisher-Guided Adaptive Selection (SFAS). In the forward pass, TADP dynamically encodes task-specific characteristics into contextual prompts, enabling feature recalibration tailored to diverse meteorological inputs without altering the core pre-trained knowledge. During backpropagation, SFAS integrates randomness with Fisher information to identify and update parameters sensitive to downstream objectives with higher possibilities, preserving invariant physical priors while optimizing task-critical weights. Experiment results on three downstream tasks demonstrate the effectiveness and efficiency of WeatherPEFT over existing PEFT methods, highlighting its adaptability to weather-related data.

## ACKNOWLEDGMENTS

This work was supported in part by the National Natural Science Foundation of China under Grant T2125006; in part by GuangDong Basic and Applied Basic Research Foundation under Grant 2025A1515510016; in part by Shenzhen Science and Technology Program under Grant KCXFZ20240903093759004 and Grant KJZD20230923115106012; in part by the Open Research Fund of Pengcheng Laboratory under No.2025KF1B0040; and in part by the Research Grants Council of the Hong Kong Special Administrative Region, China (No. CUHK 14206625). Yang Liu is supported in part by the Postdoctoral Fellowship Scheme of The Chinese University of Hong Kong.

## ETHICS STATEMENT

The authors have read and adhered to the ICLR Code of Ethics. We believe this work presents no major ethical concerns and offers significant societal benefits. The primary goal of our research is to develop more efficient methods for fine-tuning Weather Foundation Models. This work contributes positively to human well-being by making advanced weather forecasting more accessible, which is critical for applications in disaster preparedness (*e.g.*, flood and extreme weather warnings), agriculture, and water resource management. Our research exclusively utilizes publicly available meteorological datasets (*e.g.*, ERA5 and WeatherBench), which do not contain personally identifiable or sensitive human data, thereby avoiding privacy and security issues. In line with our commitment to scientific transparency and reproducibility, we have provided our code and will make it publicly available. This work has been conducted in adherence to the ICLR Code of Ethics, with the goal of fostering responsible and beneficial scientific advancement.

## REPRODUCIBILITY STATEMENT

We are committed to ensuring the reproducibility of our research. Source code and a README.md file with detailed instructions for environment setup, data preparation, and script execution are available at https://github.com/ShileiCao/WeatherPEFT and also provided in the supplementary material. The appendix offers comprehensive details to support our claims. Appendix E describes the implementation of our proposed WeatherPEFT and all baseline models. Appendix F details the setup for each downstream task, including data sources, problem settings, and formal definitions for all evaluation metrics. Furthermore, Appendix B presents extensive hyperparameter ablation studies and a generalization study to justify our main experimental choices and demonstrate the robustness of our method.

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

# Task-Adaptive Parameter-Efficient Fine-Tuning for Weather Foundation Models

## Appendix

CONTENTS

## A OVERVIEW

We provide additional details and analysis in this technical Appendix. In Section B, we furnish additional studies on hyperparameter, backbone, module fine-grained comparison, and a real-world case. In Section C, we discuss the limitations and prospective directions of our research. In Section E, we provide model implementation details on WeatherPEFT and other methodologies. In Section F, we furnish additional details and visualization examples for the downstream tasks.

## B ADDITIONAL STUDIES

### B.1 HYPERPARAMETER ABLATION STUDY

Table 4: Ablation study on key hyperparameters for the regional precipitation forecasting task, using Aurora (Bodnar et al., 2025) as the foundation model. The analyzed hyperparameters include the rank ($r$) for LoRA (Hu et al., 2022), the parameter selection percentage ($k$) and initial linear decay factor ($\gamma$) for SFAS of WeatherPEFT, and the number of soft prompt tokens ($P$) with adapter hidden dimensions ($HW_h, V_h, D_h$) for TADP of WeatherPEFT.

| Hyperparameter | Trainable Params (M) | 12 Hours | | | 24 Hours | | | 36 Hours | | |
|---|---|---|---|---|---|---|---|---|---|---|
| | | SEEPS | ACC ↑ | RMSE | SEEPS | ACC ↑ | RMSE | SEEPS | ACC ↑ | RMSE |
| Full-Tuning | 1246.77 | 0.304 | 0.797 | 0.178 | 0.452 | 0.586 | 0.241 | 0.542 | 0.481 | 0.263 |
| LoRA-$r$ = 256 | 92.01 | 0.495 | 0.591 | 0.241 | 0.633 | 0.423 | 0.272 | 0.716 | 0.306 | 0.288 |
| LoRA-$r$ = 160 | 57.80 | **0.449** | **0.648** | **0.226** | **0.590** | **0.474** | **0.263** | **0.681** | **0.353** | **0.282** |
| LoRA-$r$ = 128 | 46.39 | 0.491 | 0.592 | 0.240 | 0.627 | 0.425 | 0.271 | 0.714 | 0.307 | 0.288 |
| LoRA-$r$ = 64 | 23.59 | 0.479 | 0.606 | 0.237 | 0.641 | 0.403 | 0.274 | 0.728 | 0.282 | 0.290 |
| LoRA-$r$ = 8 | 3.63 | 0.495 | 0.592 | 0.240 | 0.634 | 0.415 | 0.273 | 0.723 | 0.294 | 0.289 |
| $k$ = 0.040 | 52.37 | **0.302** | **0.805** | **0.174** | **0.437** | 0.615 | 0.235 | **0.526** | 0.518 | 0.256 |
| $k$ = 0.035 | 46.09 | 0.303 | 0.804 | 0.175 | 0.439 | 0.615 | 0.235 | 0.528 | **0.520** | 0.256 |
| $k$ = 0.030 | 39.81 | 0.305 | 0.803 | 0.175 | 0.440 | 0.616 | **0.234** | 0.530 | 0.519 | **0.255** |
| $k$ = 0.025 | 33.53 | 0.306 | 0.803 | 0.175 | 0.441 | 0.616 | **0.234** | 0.532 | **0.520** | **0.255** |
| $k$ = 0.020 | 27.25 | 0.309 | 0.802 | 0.176 | 0.444 | **0.617** | **0.234** | 0.535 | 0.519 | **0.255** |
| $k$ = 0.015 | 20.96 | 0.312 | 0.800 | 0.177 | 0.448 | **0.617** | **0.234** | 0.540 | 0.516 | 0.256 |
| $k$ = 0.010 | 14.68 | 0.315 | 0.796 | 0.178 | 0.453 | 0.614 | **0.234** | 0.548 | 0.514 | 0.256 |
| $k$ = 0.005 | 8.40 | 0.328 | 0.785 | 0.182 | 0.468 | 0.604 | 0.237 | 0.565 | 0.499 | 0.258 |
| $k$ = 0.001 | 3.38 | 0.368 | 0.742 | 0.198 | 0.515 | 0.559 | 0.247 | 0.615 | 0.443 | 0.268 |
| $\gamma$ = 1.0 | 3.38 | 0.369 | 0.742 | **0.198** | 0.518 | 0.556 | **0.247** | 0.616 | 0.439 | 0.269 |
| $\gamma$ = 0.8 | 3.38 | 0.369 | **0.743** | **0.198** | 0.520 | 0.553 | 0.248 | 0.619 | 0.436 | 0.269 |
| $\gamma$ = 0.6 | 3.38 | 0.376 | 0.736 | 0.200 | 0.521 | 0.552 | 0.248 | 0.622 | 0.434 | 0.269 |
| $\gamma$ = 0.4 | 3.38 | 0.371 | 0.740 | 0.199 | 0.517 | 0.556 | **0.247** | 0.617 | 0.440 | **0.268** |
| $\gamma$ = 0.2 | 3.38 | **0.368** | 0.742 | **0.198** | **0.515** | 0.559 | **0.247** | **0.615** | 0.443 | **0.268** |
| $P$ = 100 | 4.98 | 0.376 | 0.736 | 0.200 | 0.524 | 0.550 | 0.248 | 0.622 | 0.434 | 0.269 |
| $P$ = 80 | 4.58 | 0.381 | 0.728 | 0.202 | 0.526 | 0.548 | 0.249 | 0.622 | 0.438 | 0.269 |
| $P$ = 60 | 4.18 | 0.375 | 0.736 | 0.200 | 0.529 | 0.544 | 0.250 | 0.631 | 0.422 | 0.272 |
| $P$ = 40 | 3.78 | 0.400 | 0.707 | 0.209 | 0.545 | 0.528 | 0.253 | 0.645 | 0.409 | 0.273 |
| $P$ = 20 | 3.38 | **0.368** | 0.742 | 0.198 | 0.515 | 0.559 | 0.247 | **0.615** | 0.443 | 0.268 |
| $P$ = 10 | 2.98 | 0.387 | 0.720 | 0.205 | 0.531 | 0.544 | 0.250 | 0.630 | 0.428 | 0.270 |
| $HW_h, V_h, D_h$ = 32, 13, 512 | 27.87 | 0.376 | 0.736 | 0.200 | 0.521 | 0.552 | 0.248 | 0.619 | 0.437 | 0.269 |
| $HW_h, V_h, D_h$ = 8, 6, 16 | 3.38 | **0.368** | **0.742** | **0.198** | **0.515** | **0.559** | **0.247** | **0.615** | **0.443** | **0.268** |

To rigorously assess the impact of key hyperparameters within WeatherPEFT, we conduct an ablation study on the regional precipitation forecasting task, with results presented in Table 4. First, we investigate the influence of $k$, the percentage of selected parameters in SFAS. The findings reveal that WeatherPEFT can achieve performance comparable to, and even superior to, full fine-tuning (1246.77M parameters) using only approximately 3% of the trainable parameters. With $k$ = 0.030, yielding 39.81M parameters, we observe SEEPS/ACC/RMSE of 0.0.440/0.616/0.234 for the 24-hour forecast, versus Full-Tuning's 0.452/0.586/0.241. Additionally, a trend indicates that as k increases, model performance generally improves across all forecast horizons (12, 24, and 36 hours). However, the magnitude of these improvements diminishes with larger k values, suggesting a point of diminishing returns where adding more trainable parameters yields only marginal gains. For

Table 5: Ablation study on key hyperparameters for the downscaling task, using Aurora (Bodnar et al., 2025) as the foundation model. The analyzed hyperparameters include parameter selection percentage ($k$) for SFAS of WeatherPEFT.

| Hyperparameter | Trainable Params (M) | T2m | | U10 | | V10 | | T850 | | Z500 | |
|---|---|---|---|---|---|---|---|---|---|---|---|
| | | RMSE | Mean Bias | RMSE | Mean Bias | RMSE | Mean Bias | RMSE | Mean Bias | RMSE | Mean Bias |
| Full-Tuning | 1239.94 | **0.906** | 0.002 | 0.882 | **0.000** | 0.884 | **-0.001** | 0.836 | **0.000** | 35.821 | **0.314** |
| $k = 0.04$ | 52.47 | 0.916 | **0.000** | **0.873** | -0.001 | **0.875** | -0.002 | **0.834** | -0.002 | **35.076** | 0.504 |
| $k = 0.03$ | 39.91 | 0.929 | **0.000** | 0.882 | -0.001 | 0.883 | -0.002 | 0.840 | -0.002 | 35.511 | 0.502 |
| $k = 0.02$ | 27.34 | 0.949 | -0.002 | 0.898 | -0.002 | 0.898 | -0.002 | 0.851 | -0.001 | 36.284 | 0.630 |
| $k = 0.01$ | 14.82 | 0.987 | -0.001 | 0.928 | -0.001 | 0.927 | -0.003 | 0.869 | -0.002 | 37.826 | 0.355 |
| $k = 0.001$ | 3.48 | 1.119 | 0.003 | 1.057 | **0.000** | 1.051 | **-0.001** | 0.950 | 0.004 | 44.922 | 0.413 |

Table 6: Ablation study on key hyperparameters for the ensemble weather forecast post-processing task, using Aurora (Bodnar et al., 2025) as the foundation model. The analyzed hyperparameters include parameter selection percentage ($k$) for SFAS of WeatherPEFT.

| Hyperparameter | Trainable Params (M) | T2m | | U10 | | V10 | | T850 | | Z500 | |
|---|---|---|---|---|---|---|---|---|---|---|---|
| | | CRPS | EECRPS | CRPS | EECRPS | CRPS | EECRPS | CRPS | EECRPS | CRPS | EECRPS |
| Full-Tuning | 1239.94 | 0.604 | 0.206 | **0.838** | **0.284** | **0.854** | **0.285** | 0.653 | 0.223 | 73.760 | 27.051 |
| $k = 0.04$ | 52.47 | **0.601** | **0.205** | **0.838** | **0.284** | **0.854** | 0.286 | **0.650** | **0.222** | 72.745 | 26.683 |
| $k = 0.03$ | 39.91 | 0.605 | 0.207 | **0.838** | **0.284** | **0.854** | **0.285** | 0.652 | 0.223 | 74.102 | 27.247 |
| $k = 0.02$ | 27.34 | 0.606 | 0.207 | 0.839 | **0.284** | 0.855 | 0.286 | 0.652 | 0.223 | 73.757 | 27.082 |
| $k = 0.01$ | 14.82 | 0.608 | 0.208 | 0.841 | 0.285 | 0.857 | 0.287 | 0.654 | 0.223 | 73.438 | 26.958 |
| $k = 0.001$ | 3.48 | 0.618 | 0.211 | 0.844 | 0.286 | 0.860 | 0.287 | 0.657 | 0.224 | **72.701** | **26.665** |

fair comparisons with other PEFT methodologies in this paper, we select $k$=0.001 for most of the experiments, ensuring a comparable parameter budget. To unleash the potential of WeatherPEFT and ensure a fair comparison with Full-Tuning, we supplement experiments with $k$ set to 0.04. To explicitly validate this consistency across all tasks, we have conducted the same ablation study on the hyperparameter $k$ for the other two downstream tasks: Downscaling and Ensemble Weather Forecast Post-Processing. The results are presented in Tables 5 and 6, which demonstrate a clear and consistent trend across two tasks. WeatherPEFT's performance scales with trainable parameters, matching or surpassing Full-Tuning when using 3% of the model's parameters. Beyond this, performance gains gradually plateau. Additionally, we conduct a hyperparameter sweep on LoRA's rank on Aurora. The results on the precipitation forecasting task show LoRA's performance is insensitive to its parameter count and remains significantly inferior to WeatherPEFT even when it uses fewer parameters. This confirms that WeatherPEFT's superiority stems from a fundamental architectural advantage, not from suboptimal baseline tuning.

Furthermore, our ablation on $\gamma$, the initial value of the linear decay factor in SFAS, demonstrates that the model exhibits relative insensitivity to this hyperparameter, with $\gamma = 0.2$ yielding the optimal or jointly optimal results across most metrics and forecast horizons. This could be attributed to the weights progressively decaying towards zero, making the initial value of $\gamma$ less critical to the final results. Regarding the prompt length $P$ in TADP, experiments show that increasing the number of soft prompt tokens beyond P=20 does not lead to further performance improvements and, in some cases, results in slight degradation (e.g., 12-hour SEEPS increased from 0.368 at $P = 20$ to 0.400 at P=40). Given that longer prompts also increase the trainable parameter count (from 3.38M at $P = 20$ to 4.98M at $P = 100$), $P = 20$ is identified as the most reasonable setting for this task, suggesting that excessive prompt lengths may introduce redundant parameters or make optimization more challenging without contributing additional descriptive power. Finally, the ablation on the hidden dimensions $(HW_h, V_h, D_h)$ of the three adapters in TADP indicates that increasing these dimensions from a compact (8, 6, 16) to a larger (32, 13, 512) configuration (*i.e.* forgoing dimensionality reduction to retain dimensions of the original features), which drastically increase trainable parameters from 3.38M to 27.87M, do not yield performance benefits and, in fact, led to a decline in metrics. This suggests that larger adapter capacities may be prone to overfitting on the downstream task or are not necessary for capturing the task-specific information for regional precipitation forecasting, making the smaller dimensions more efficient and effective. These analyses affirm the selected hyperparameter values for achieving a strong balance between performance and efficiency.

## B.2 BACKBONE GENERALIZATION STUDY

Table 7: Generalization study on the backbone for regional precipitation forecasting in the China region. Performance is evaluated using SEEPS, ACC, and RMSE (1e-2). Prithvi-WxC (Schmude et al., 2024) is adopted as the foundation model, and for fine-tuning methods, we report only the trainable parameters within the backbone.

| Method | Trainable Params (M) | 12 Hours | | | 24 Hours | | | 36 Hours | | |
|---|---|---|---|---|---|---|---|---|---|---|
| | | SEEPS | ACC ↑ | RMSE | SEEPS | ACC ↑ | RMSE | SEEPS | ACC ↑ | RMSE |
| Prithvi WxC | 1979.10 | 0.435 | 0.649 | 0.226 | 0.542 | 0.505 | 0.259 | 0.630 | 0.404 | 0.275 |
| Full-Tuning | 1978.47 | **0.398** | **0.678** | **0.218** | **0.517** | 0.521 | 0.256 | **0.604** | 0.419 | 0.273 |
| LoRA | 86.47 | 0.647 | 0.406 | 0.273 | 0.760 | 0.231 | 0.297 | 0.813 | 0.149 | 0.307 |
| WeatherPEFT | **81.99** | 0.405 | **0.678** | **0.218** | 0.523 | **0.522** | **0.255** | 0.605 | **0.428** | **0.272** |

In WeatherPEFT, SFAS is universally applicable to any model trained with gradient-based optimization. Regarding TADP, its core concept involves applying a series of projection transformations to the encoder's embedding space. This extracts task-specific representations, which are concatenated as soft prompts to the input of each layer in the backbone model. TADP offers broad applicability across diverse settings due to three key factors:

- **Unified Operation Target**: Embedding space is irrespective of architecture (e.g., Transformers (Vaswani et al., 2017), Convolutional Neural Networks (Krizhevsky et al., 2012), or Graph Neural Networks Scarselli et al. (2008)), and all models involve an embedding operation mapping input data to a continuous feature space for subsequent computation. Consequently, TADP can be applied to various embedding networks/encoders by identifying their corresponding embedding weight matrices.

- **Consistent Feature Processing Across Architectures**: Fundamentally, diverse model architectures perform multi-layered computations on input feature vectors to produce outputs. TADP concatenates soft prompts to the input feature map at each layer. Therefore, adapting TADP to different backbones simply requires minor adjustments based on the specific characteristics of the extracted feature maps.

- **Extension to Multi-modal Inputs**: Handling multi-modal inputs typically involves transitioning from a single-modal encoder to multiple single-modal encoders. TADP can integrate the embedding weight of multi-modal encoders. Task-specific representations are subsequently derived from this integrated space and concatenated as soft prompts to the backbone's inputs at each layer.

In summary, WeatherPEFT demonstrates strong generalization capability across variations in model architecture, embedding methods, and input modalities. To provide concrete evidence for these claims, we further evaluate our method on a different, larger foundation model: Prithvi-WxC (Schmude et al., 2024). The results on the regional precipitation forecasting task are shown in Table 7. We note that this model is pre-trained on data sources that are more dissimilar to our downstream tasks compared to Aurora (Bodnar et al., 2025), which makes effective fine-tuning more challenging. As the table demonstrates, WeatherPEFT still achieves similar performance, matching Full-Tuning using only 4% of the parameters. Crucially, the generic PEFT baseline, LoRA, performs very poorly on this architecture. This result strongly underscores the necessity of a weather-specific and adaptive PEFT method like WeatherPEFT, as generic approaches are not guaranteed to be effective across different WFMs.

## B.3 MODULE FINE-GRAINED COMPARISON STUDY

To precisely evaluate the individual mechanisms of WeatherPEFT, we conduct a fine-grained ablation study on the downscaling task (Table 8), dissecting components of Task-Adaptive Dynamic Prompting (TADP) and Stochastic Fisher-Guided Adaptive Selection (SFAS).

Within TADP, ablating either the 'Internal' pattern extraction (designed for task-specific physical constraints) or the 'External' pattern extraction (for coupling physical quantities with spatial resolution features) consistently leads to performance degradation compared to the full WeatherPEFT. For instance, T2m RMSE increases from 1.119 in the full model to 1.140 (w/o Internal) and 1.130 (w/o

Table 8: Fine-grained Ablation study on TADP and SFAS. 'External' and 'Internal' represent the external and internal pattern extraction in TADP, while 'Randomness' denotes the stochastic component in SFAS. We adopt the Aurora (Bodnar et al., 2025) as the foundation model. Experiments are done on the downscaling task under the limited (top) and increased (bottom) parameter budgets.

| Method | T2m | | U10 | | V10 | | T850 | | Z500 | |
|---|---|---|---|---|---|---|---|---|---|---|
| | RMSE | Mean Bias | RMSE | Mean Bias | RMSE | Mean Bias | RMSE | Mean Bias | RMSE | Mean Bias |
| w/o Internal | 1.140 | 0.007 | 1.076 | 0.001 | 1.069 | -0.002 | 0.964 | **0.004** | 46.027 | 1.048 |
| w/o External | 1.130 | 0.006 | 1.068 | 0.001 | 1.062 | -0.003 | 0.958 | **0.004** | 45.292 | 0.787 |
| w/o Randomness | 1.130 | 0.005 | 1.069 | **0.000** | 1.062 | **0.000** | 0.956 | 0.005 | 45.808 | 0.714 |
| WeatherPEFT | **1.119** | **0.003** | **1.057** | **0.000** | **1.051** | -0.001 | **0.950** | 0.004 | **44.922** | **0.413** |
| w/o Internal | 0.970 | **0.000** | 0.913 | -0.002 | 0.912 | **-0.002** | 0.860 | -0.002 | 36.870 | 0.611 |
| w/o External | 0.958 | **0.000** | 0.903 | **-0.001** | 0.903 | -0.003 | 0.854 | **-0.001** | 36.415 | 0.640 |
| w/o Randomness | 0.954 | **0.000** | 0.900 | -0.002 | 0.901 | **-0.002** | 0.852 | **-0.001** | 36.277 | 0.620 |
| WeatherPEFT | **0.916** | **0.000** | **0.873** | -0.001 | **0.875** | -0.002 | **0.834** | -0.002 | **35.076** | **0.504** |

External), highlighting the importance of these components for adapting to input data characteristics, particularly vital for downscaling. Similarly, for SFAS, removing the 'Randomness' (stochastic component), intended to stabilize parameter selection, results in higher RMSE values for most variables (*e.g.* T2m RMSE increased to 1.130), underscoring the need for stabilizing the parameter selection. However, we observe that in the low-parameter regime, the performance differences, distinct yet relatively small. This phenomenon is likely attributable to "performance saturation", where the optimization landscape is tightly constrained by the minimal trainable parameter budget, compressing the variance between methods. The results of larger-parameter setting demonstrate that the performance gaps become significantly more pronounced as capacity increases. These findings collectively demonstrate that the Internal and External pattern extraction mechanisms are essential for robust scaling. They allow the model to efficiently utilize additional capacity to capture complex meteorological dynamics, preventing the premature plateauing observed in the ablated variants.

The complete WeatherPEFT consistently achieves the overall best performance (*e.g.*, lowest RMSE for T2m, U10, T850, Z500). This demonstrates that each evaluated sub-component contributes meaningfully and synergistically to WeatherPEFT's robust and efficient adaptation capabilities.

## B.4 REAL-WORLD CASE STUDY

Table 9: Real-world case study on the extreme 2020 China Mei-yu flood event. Performance is evaluated using the 50th and 75th percentile Threat Score (TS) and SEEPS with forecast initialized from 7.1 12:00 on the China region. Aurora (Bodnar et al., 2025) is adopted as the foundation model, and for fine-tuning methods, we report only the trainable parameters within the backbone.

| Method | Trainable Params (M) | 12 Hours | | | 24 Hours | | | 36 Hours | | |
|---|---|---|---|---|---|---|---|---|---|---|
| | | 50%TS | 75%TS | SEEPS ↓ | 50%TS | 75%TS | SEEPS ↓ | 50% TS | 75%TS | SEEPS ↓ |
| Full-Tuning | 1246.77 | 0.64 | **0.50** | **0.34** | 0.70 | 0.45 | **0.67** | 0.57 | 0.34 | 0.68 |
| LoRA | 57.80 | 0.58 | 0.37 | 0.49 | 0.68 | 0.34 | 0.86 | 0.52 | 0.26 | 0.83 |
| DoRA | 57.92 | 0.54 | 0.32 | 0.55 | 0.65 | 0.31 | 0.89 | 0.49 | 0.19 | 0.94 |
| AdaptFormer | 61.68 | 0.59 | 0.40 | 0.45 | 0.68 | 0.36 | 0.83 | 0.54 | 0.25 | 0.82 |
| WeatherPEFT | **52.37** | **0.65** | **0.50** | **0.34** | **0.72** | **0.46** | **0.67** | **0.58** | **0.37** | **0.66** |

To demonstrate practical utility, we further conduct a case study on the extreme 2020 China Mei-yu (plum rain) flood, which is documented as a period of record-breaking flooding (Ding et al., 2021; Volonté et al., 2021). We initialize a forecast at 12:00 UTC on July 1, 2020, during an intensely active phase of this event, evaluating performance with decision-relevant metrics such as the 50th and 75th percentile Threat Score (TS) and SEEPS. The results in Table 9 show that with only 4% of the parameters, WeatherPEFT's performance on heavy rainfall forecasts is comparable to Full-Tuning. Crucially, it also outperforms the generic PEFT baselines, including LoRA (Hu et al., 2022), DoRA (Liu et al., 2024), and AdaptFormer (Chen et al., 2022), despite their larger number of trainable parameters. This demonstrates that our method's targeted approach offers tangible efficiency benefits for real-world extreme event prediction.

## B.5 SYNERGISTIC ANALYSIS

Table 10: Synergistic analysis study on the backbone for regional precipitation forecasting in the China region. Performance is evaluated using SEEPS, ACC, and RMSE (1e-2). Aurora (Bodnar et al., 2025) is adopted as the foundation model, and for fine-tuning methods, we report only the trainable parameters within the backbone.

| Method | Trainable Params (M) | 12 Hours | | | 24 Hours | | | 36 Hours | | |
|--------|---------------------|----------|--|--|----------|--|--|----------|--|--|
| | | SEEPS | ACC ↑ | RMSE | SEEPS | ACC ↑ | RMSE | SEEPS | ACC ↑ | RMSE |
| AdaptFormer+SFAS | 5.88 | 0.475 | 0.608 | 0.236 | 0.617 | 0.419 | 0.272 | 0.705 | 0.302 | 0.288 |
| LoRA+SFAS | 4.89 | 0.446 | 0.647 | 0.227 | 0.592 | 0.464 | 0.265 | 0.701 | 0.316 | 0.286 |
| VPT+SFAS | 5.01 | 0.395 | 0.708 | 0.209 | 0.537 | 0.533 | 0.252 | 0.639 | 0.41 | 0.273 |
| WeatherPEFT | **3.38** | **0.368** | **0.742** | **0.198** | **0.515** | **0.559** | **0.247** | **0.615** | **0.443** | **0.268** |

To rigorously validate whether the proposed TADP module provides significant architectural value beyond simply applying sparse adaptive parameter selection to existing methods, we conduct a comparative study. We integrate the proposed SFAS mechanism with representative generic PEFT methods, including LoRA (Hu et al., 2022), AdaptFormer (Chen et al., 2022), and VPT (Jia et al., 2022), and compare them with WeatherPEFT on the Regional Precipitation Forecasting task.

As presented in the Table 10, simply adding SFAS to generic adapters yields suboptimal results compared to WeatherPEFT. While adding SFAS to methods like VPT does improve performance relative to their standard counterparts (Table 3), they still consistently lag behind WeatherPEFT. Notably, WeatherPEFT achieves the best performance while utilizing fewer parameters compared to the combinatorial baselines. These results suggest that generic adapters, even when optimized with Fisher-guided selection, fail to adequately capture the complex variable-specific couplings and physical regime shifts inherent in weather data. By explicitly modeling internal and external patterns through TADP, WeatherPEFT provides a more effective initialization for the selection process. This empirically demonstrates that TADP is not merely a supplementary module but a critical architectural component that works synergistically with SFAS to achieve superior adaptation.

## B.6 COMPUTATIONAL EFFICIENCY ANALYSIS

Table 11: Comparison of training times across different tasks.

| Methods | Training Time | | |
|---------|---------------|--|--|
| | Downscaling | Post-Processing | Precipitation Forecasting |
| LoRA | 5h09m | 1h09m | 1h42m |
| AdaptFormer | 5h05m | 1hs06m | 1h40m |
| Ours | 5h33m | 1h20m | 1h58m |

The sequential implementation of the three specialized adapters in TADP and the parameter selection mechanism in SFAS might introduce a degree of computational overhead compared to simpler techniques. To quantitatively evaluate this trade-off between algorithmic complexity and computational efficiency, we measure the total training time for WeatherPEFT against representative PEFTs (LoRA (Hu et al., 2022) and AdaptFormer (Chen et al., 2022)) across all three downstream tasks.

As shown in the Table 11, WeatherPEFT incurs a modest training time increase of approximately 10% compared to LoRA. This marginal increase in wall-clock training time is a highly favorable trade-off given the substantial performance gains demonstrated in the main experiments. It enables WFMs to accurately solve complex downstream tasks where generic, faster PEFT methods fail to capture the necessary physical dynamics.

## B.7 ADDITIONAL DOMAIN SPECIFICITY ANALYSIS

To verify that the performance gains of WeatherPEFT stem from addressing meteorological challenges, we evaluate it on a standard vision task. Specifically, we conduct experiments on the Cityscapes (Cordts et al., 2016) → ACDC (Sakaridis et al., 2021) domain generalization benchmark for semantic segmentation, which encompasses the Night, Snow, Fog, and Rain as the target

Table 12: The mIoU (%) on Cityscapes to ACDC domain generalization benchmark for semantic segmentation. We adopt the DINOv2-L (Oquab et al., 2024) as the foundation model.

| Methods | Trainable Params (M) | ACDC (Target) Night | Snow | Fog | Rain | Mean |
|---|---|---|---|---|---|---|
| Full-Tuning | 304.20 | 52.4 | 70.5 | 80.9 | 74.4 | 69.5 |
| Linear-Probing | 0.00 | 54.3 | 69.3 | 79.1 | 68.0 | 67.6 |
| Convpass | 3.64 | **56.0** | 71.7 | 80.2 | **74.9** | 70.7 |
| FacT-TT | 2.85 | 56.1 | 71.3 | 81.0 | 72.9 | 70.3 |
| MOA | 6.39 | 53.2 | 70.6 | 80.3 | 72.8 | 69.3 |
| LoRA | 3.14 | 52.3 | 74.4 | 79.5 | 74.0 | 70.1 |
| AdaptFormer | 3.17 | 53.8 | **74.8** | 80.3 | 74.6 | **70.9** |
| VPT | 3.15 | 53.4 | 74.4 | 80.4 | 70.5 | 69.7 |
| Ours | 2.90 | **56.0** | 70.9 | **81.2** | 74.5 | 70.7 |

| Methods | params (M) | **Natural** Cifar100 | Caltech101 | DTD | Flower102 | Pets | SVHN | Sun397 | **Specialized** Camelyon | EuroSAT | Resisc45 | Retinopathy | **Structured** Clevr-Count | Clevr-Dist | DMLab | KITTI-Dist | dSpr-Loc | dSpr-Ori | sNORB-Azim | sNORB-Ele | Average |
|---|---|---|---|---|---|---|---|---|---|---|---|---|---|---|---|---|---|---|---|---|---|
| Full-Tuning | 85.8 | 68.9 | 87.7 | 64.3 | 97.2 | 86.9 | 87.4 | 38.8 | 79.7 | 95.7 | 84.2 | 73.9 | 56.3 | 58.6 | 41.7 | 65.5 | 57.5 | 46.7 | 25.7 | 29.1 | 68.9 |
| Linear-Probing | 0.00 | 64.4 | 85.0 | 63.2 | 97.0 | 86.3 | 36.6 | 51.0 | 78.5 | 87.5 | 68.5 | 74.0 | 34.3 | 30.6 | 33.2 | 55.4 | 12.5 | 20.0 | 9.6 | 19.2 | 57.6 |
| Convpass | 0.33 | 72.3 | 91.2 | 72.2 | 99.2 | 90.9 | **91.3** | 54.9 | 84.2 | 96.1 | 85.3 | 75.6 | 82.3 | 67.9 | 51.3 | 80.0 | **85.9** | 53.1 | 36.4 | 44.4 | **76.6** |
| FacT-TK | 0.07 | 70.6 | 90.6 | 70.8 | 99.1 | 90.7 | 88.6 | 54.1 | 84.8 | **96.2** | 84.5 | 75.7 | 82.6 | 68.2 | 49.8 | **80.7** | 80.8 | 47.4 | 33.2 | 43.0 | 75.6 |
| RepAdapter | 0.22 | 72.4 | 91.6 | 71.0 | 99.2 | 91.4 | 90.7 | **55.1** | 85.3 | 95.9 | 84.6 | 75.9 | 82.3 | 68.0 | 50.4 | 79.9 | 80.4 | 49.2 | **38.6** | 41.0 | 76.1 |
| SSF | 0.24 | 69.0 | **92.6** | 75.1 | **99.4** | **91.8** | 90.2 | 52.9 | **87.4** | 95.9 | **87.4** | 75.5 | 75.9 | 62.3 | **53.3** | 80.6 | 77.3 | **54.9** | 29.5 | 37.9 | 75.7 |
| SCT | 0.11 | 75.3 | 91.6 | 72.2 | 99.2 | 91.1 | 91.2 | 55.0 | 85.0 | 96.1 | 86.3 | 76.2 | 81.5 | 65.1 | 51.7 | 80.2 | 75.4 | 46.2 | 33.2 | **45.7** | 76.0 |
| LoRA | 0.29 | 67.1 | 91.4 | 69.4 | 98.8 | 90.4 | 85.3 | 54.0 | 84.9 | 95.3 | 84.4 | 73.6 | **82.9** | **69.2** | 49.8 | 78.5 | 75.7 | 47.1 | 31.0 | 44.0 | 74.5 |
| AdaptFormer | 0.16 | 70.8 | 91.2 | 70.5 | 99.1 | 90.9 | 86.6 | 54.8 | 83.0 | 95.8 | 84.4 | **76.3** | 81.9 | 64.3 | 49.3 | 80.3 | 76.3 | 45.7 | 31.7 | 41.1 | 74.7 |
| VPT | 0.53 | **78.8** | 90.8 | 65.8 | 98.0 | 88.3 | 78.1 | 49.6 | 81.8 | 96.1 | 83.4 | 68.4 | 68.5 | 60.0 | 46.5 | 72.8 | 73.6 | 47.9 | 32.9 | 37.8 | 72.0 |
| Ours | 0.29 | 73.1 | 92.2 | 71.9 | 99.2 | 90.2 | 89.2 | 53.5 | 83.3 | 95.0 | 83.4 | 73.6 | 81.3 | 68.0 | 46.8 | 74.8 | 72.3 | 45.2 | 28.3 | 37.6 | 74.0 |

Table 13: Results on VTAB-1K (Zhai et al., 2019) Benchmark with ViT-B/16 (Dosovitskiy et al., 2020) backbone.

domains. We compare WeatherPEFT against established vision PEFT methods, including ConvPass, FacT, MoA, LoRA, AdaptFormer, and VPT. We utilize Dinov2-L (Oquab et al., 2024) as the backbone and report the mean Intersection over Union (mIoU). The results indicate that while WeatherPEFT remains competitive in the vision domain (comparable to AdaptFormer), it does not demonstrate the dominant superiority observed in the weather tasks. This distinction is pivotal, verifying that WeatherPEFT functions not merely as an enhanced general adapter, but rather as a method specifically optimized for the unique physical semantics of weather data. Notably, the dynamic, annealed selection mechanism of SFAS, combined with context-aware dynamic prompting of TADP, provides distinct advantages in meteorological contexts.

To further investigate the generalizability and domain specificity of our approach, we also evaluate WeatherPEFT on the VTAB-1K benchmark (Zhai et al., 2019), a standard suite for evaluating transfer learning in computer vision. We utilize a ViT-B/16 (Dosovitskiy et al., 2020) backbone pre-trained on ImageNet-21k (Deng et al., 2009). We compare our method against representative visual PEFT methods, including Convpass (Jie et al., 2024), FacT (Jie & Deng, 2023), RepAdapter (Luo et al., 2023), SSF (Lian et al., 2022), SCT (Zhao et al., 2024a), LoRA (Hu et al., 2022), AdaptFormer (Chen et al., 2022), and VPT (Jia et al., 2022). As demonstrated in the Table 13, WeatherPEFT achieves an average accuracy of 74.0%, which is comparable to general PEFT methods like LoRA and AdaptFormer. However, we observe that our method performs slightly below the SOTA on the "Structured" task group (e.g., dSprites, sNORB). We attribute this performance difference to a fundamental distinction between the VTAB-1K experimental setting and the weather domains for which our method was optimized. The core design of our TADP is to extract task-specific characteristics (*e.g.*, variable types and physical resolutions) from the encoder's embedding layer to introduce context-aware feature recalibration. In weather tasks, the embedding layer is rich with varying physical information, allowing TADP to dynamically adapt the model to the specific

"physics" of the input. In contrast, for standard vision tasks like VTAB-1K, the embedding layers of the backbone are typically frozen and process homogeneous RGB data. In this setting, TADP extracts information from a fixed layer, causing the "dynamic prompt" to effectively become a static constant. This neutralizes the primary advantage of TADP's adaptivity, resulting in performance that is competitive with, but not significantly superior to, other baselines. In summary, while WeatherPEFT is capable of handling generic tasks, its superior performance is unlocked in the weather domain, validating our motivation for a domain-specialized design that addresses meteorological challenges.

## C DISCUSSION

Table 14: Scaling trends in weather foundation models.

| Model | Year | Parameters | Training Resources |
|---|---|---|---|
| FourCastNet (Pathak et al., 2022) | 2022 | 64M | 16 hours; 64 A100 GPUs |
| Pangu (Bi et al., 2023) | 2022 | 65M | 16 days; 192 V100 GPUs |
| GraphCast (Lam et al., 2023) | 2022 | 37M | 28 days; 32 TPU v4 |
| ClimaX (Nguyen et al., 2023a) | 2023 | 117M | ∼3 days; 80 V100 GPUs |
| FengWu (Chen et al., 2023a) | 2023 | 158M | 17 days; 32 A100 GPUs |
| Fuxi (Chen et al., 2023b) | 2023 | 157M | ∼8 days; 8 A100 GPUs |
| Aurora (Bodnar et al., 2025) | 2024 | 1.3B | ∼18 days; 32 A100 GPUs |
| Prithvi WxC (Schmude et al., 2024) | 2024 | 2.3B | 64 A100 GPUs |

While WeatherPEFT demonstrates promising advances in PEFT for WFMs, several aspects warrant further discussion:

**Scales of WFMs.** First, one potential limitation of the current work pertains to the existing scale of WFMs. We are currently in the early stages of developing general AI for the weather domain. Current WFMs, including Aurora (Bodnar et al., 2025) and ClimaX (Nguyen et al., 2023a), remain in their infancy compared to mature Computer Vision (CV) or Natural Language Processing (NLP) foundation models. These models are generally smaller and less computationally demanding than their counterparts in NLP or CV, which might initially lessen the perceived urgency for PEFT methods in meteorological science. However, this view is rapidly being challenged by the swift expansion of WFMs. As detailed in Table 14, recent models such as Aurora (1.3B parameters) (Bodnar et al., 2025) and Prithvi WxC (2.3B parameters) (Schmude et al., 2024) already highlight a clear trajectory towards billion-parameter scales and increasing computational requirements. This trend indicates that the computational and storage demands for fine-tuning will soon become unsustainable for many institutions. As a case in point, Environment Canada reported that GPU memory constraints make it "effectively impossible" to fully fine-tune GraphCast Lam et al. (2023) on their in-house systems (Subich, 2025). In this evolving context, WeatherPEFT is presented as a forward-looking initiative. Our work aims to proactively establish efficient adaptation methodologies that will be essential for the accessible and sustainable deployment of these increasingly large and complex future-generation weather foundation models.

**Gnerlization of WeatherPEFT.** Furthermore, WeatherPEFT has only been validated on the transformer-based backbone, including Aurora (Bodnar et al., 2025) and Prithvi WxC (Schmude et al., 2024), but it can be adapted to other architectures with minor modifications as discussed in Appendix B.2. Future work should prioritize its extension to other foundational architectures, such as Convolutional Neural Networks and Graph Neural Networks. Testing its performance across a broader range of downstream tasks will also be crucial for confirming its generalizability.

**Trade-off between Efficiency and Performance.** Moreover, it is a general observation in the PEFT field that a marginal performance gap can sometimes exist when compared to the absolute ceiling achievable by exhaustive full fine-tuning when fine-tuning only a minuscule fraction of parameters (∼0.3%). This potential, slight differential is broadly considered an acceptable trade-off. As demonstrated in Appendix B.1, this performance gap for PEFT methods narrows significantly

as the budget of trainable parameters is increased to ∼3%. Our method, WeatherPEFT, completely closes this gap, achieving performance that is on par with, and on certain metrics even superior to, that of full fine-tuning. Practitioners can select the optimal balance based on their specific application, choosing extreme efficiency with a small performance trade-off or allocating a modest parameter budget to achieve performance parity with full fine-tuning.

**Out of Distribution Scenarios.** While the WeatherPEFT framework does not include an explicit mechanism for general out-of-distribution (OOD) generalization, our experimental results provide evidence of its robustness to specific distribution shifts, namely extreme weather events. This capability is demonstrated by its superior performance on metrics designed to penalize errors on rare phenomena, including EECRPS and SEEPS. Furthermore, we evaluate WeatherPEFT on the real-world case study of the 2020 Mei-yu flood, where it achieves a high Threat Score (TS), a key decision-relevant metric. We attribute this enhanced performance to our adaptive parameter selection method, SFAS. By dynamically identifying and fine-tuning the most task-critical parameters, SFAS more effectively captures the dynamics of events in the tails of the data distribution compared to fixed PEFT strategies. This indicates a promising robustness against the OOD challenges posed by extreme events.

**Physical Mechanisms Incorporation.** Finally, the current WeatherPEFT framework, while adapting effectively through its data-driven components, does not explicitly incorporate domain-specific physical mechanisms or constraints from atmospheric science directly into the PEFT process itself. Future research could investigate domain-specific PEFT methods tailored to weather and climate applications to improve the performance, such as integrating physical mechanisms into the fine-tuning process (*e.g.*, embedding conservation laws or dynamical constraints).

## D  USE OF LARGE LANGUAGE MODELS (LLMS)

In the preparation of this manuscript, Large Language Models (LLMs) are utilized as a general-purpose assistive tool to enhance the quality and clarity of the writing. The core research, experimental design, data analysis, and intellectual contributions remain entirely the work of the authors. The specific applications of LLMs in this work include:

- **Text Polishing and Refinement**: The LLM is employed to review the entire text for grammatical accuracy, improve sentence structure, and ensure consistent phrasing and tone throughout the paper. This process is akin to using an advanced grammar and style checker to improve the overall readability of the manuscript.
- **Coherence and Logical Flow**: We use the LLM to help organize and structure our arguments. By presenting existing drafts of sections to the model, we receive suggestions on how to improve the logical transitions between paragraphs and make the overall narrative more coherent and compelling for the reader.
- **Supplementing and Articulating Ideas**: At various stages, the LLM serves as a sounding board to help supplement our thoughts. It assists in articulating complex ideas more clearly and exploring alternative ways to frame concepts that were already formulated by the authors. The model does not contribute to the original ideation or the generation of novel research findings but rather acts as an aid to express the authors' own thoughts more effectively.

All suggestions and modifications proposed by the LLM are critically reviewed, edited, and approved by the authors to ensure they accurately reflect our research and intended meaning. The final responsibility for the content of this paper rests solely with the authors.

## E  ADDITIONAL MODEL IMPLEMENTATION DETAILS

### E.1  TRAINING-FROM-SCRATCH MODEL ARCHITECTURES

#### E.1.1  RESNET

We build the ResNet (He et al., 2016) architecture based on WeatherBench (Rasp et al., 2020; 2024) and ClimateLearn (Nguyen et al., 2023b), where each residual block consists of two identical convo-

lutional modules: 2D convolution → LeakyReLU with $\alpha = 0.3$ → Batch Normalization → Dropout. Table 15 shows the hyperparameters for ResNet in all of our experiments.

Table 15: Default hyperparameters of ResNet

| Hyperparameter | Meaning | Value |
|---|---|---|
| Padding size | Padding size of each convolution layer | 1 |
| Kernel size | Kernel size of each convolution layer | 3 |
| Stride | Stride of each convolution layer | 1 |
| Hidden dimension | The number of output channels of each residual block | 256 |
| Residual blocks | The number of residual blocks | 28 |
| Dropout | Dropout rate | 0.1 |

### E.1.2  U-NET

We borrow our U-Net (Ronneberger et al., 2015) implementation from ClimateLearn (Nguyen et al., 2023b). We use the following hyperparameters in the Table 16 for UNet in all of our experiments. Similar to ResNet, we use a convolutional layer with a kernel size of 7 at the beginning of the network, and all paddings are periodic in the longitude direction and zeros in the latitude direction.

Table 16: Default hyperparameters of U-net

| Hyperparameter | Meaning | Value |
|---|---|---|
| Padding size | Padding size of each convolution layer | 1 |
| Kernel size | Kernel size of each convolution layer | 3 |
| Stride | Stride of each convolution layer | 1 |
| Hidden dimension | The number of base channels of each block | 64 |
| Channel multiplications | The number of feature channels to scale | (1,2,2) |
| Blocks | The number of blocks | 4 |
| Use attention | If use attention in Down and Up blocks | False |
| Dropout | Dropout rate | 0.1 |

### E.1.3  VIT

We implement the ViT (Dosovitskiy et al., 2020) architecture according to ClimateLearn (Nguyen et al., 2023b), which differs from the standard ViT with some minor modifications. Specifically, the class token is removed with a 1-hidden MLP prediction head incorporating, which is applied to the tokens after the last attention layer to predict the outputs. Table 17 demonstrates the hyperparameters for ViT in all of our experiments based on ViT-B.

Table 17: Default hyperparameters of ViT

| Hyperparameter | Meaning | Value |
|---|---|---|
| Padding size | The patch size to embed the input to the token | 8 |
| Hidden dimension | The number of embedding dimension | 1024 |
| Depth | The number of ViT blocks | 24 |
| Heads | The number of attention heads | 16 |
| MLP ratio | Determine the hidden dimension of the MLP layer in a ViT block | 4 |
| Prediction depth | The number of layers of the prediction head | 4 |
| Drop path | For stochastic depth rate (Huang et al., 2016) | 0.1 |
| Dropout | Dropout rate | 0.1 |

### E.1.4  FOURCASTNET

The FourCastNet is implemented based on the official code of FourCastNet (Pathak et al., 2022). As shown in the Table 18, we employ the following default hyperparameters for FourCastNet.

Table 18: Default hyperparameters of FourCastNet

| Hyperparameter | Meaning | Value |
|---|---|---|
| Padding size | The patch size to embed the input to the token | 4 |
| Sparsity threshold | The threshold of sparsity controlling in the Soft-Thresholding | 0.01 |
| Hidden dimension | The number of embedding dimension | 768 |
| Block number | The number of AFNO (Guibas et al., 2021) blocks | 8 |
| Depth | The number of layers | 12 |
| MLP ratio | Determine the hidden dimension of the MLP layer in a ViT block | 4 |
| Activation layer | The activation function within each layer (Huang et al., 2016) | GELU |
| Dropout | Dropout rate | 0 |

### E.1.5 CLIMAX

The ClimaX is implemented based on the official code of ClimaX (Nguyen et al., 2023a). As shown in the Table 19, we employ the following default hyperparameters for ClimaX in all of our experiments.

Table 19: Default hyperparameters of ClimaX

| Hyperparameter | Meaning | Value |
|---|---|---|
| Padding size | The patch size to embed the input to the token | 4 |
| Hidden dimension | The number of embedding dimension | 1024 |
| Depth | The number of ViT blocks | 8 |
| Heads | The number of attention heads | 16 |
| MLP ratio | Determine the hidden dimension of the MLP layer in a ViT block | 4 |
| Prediction depth | The number of layers of the prediction head | 2 |
| Drop path | For stochastic depth rate (Huang et al., 2016) | 0.1 |
| Dropout | Dropout rate | 0.1 |

### E.1.6 AURORA

The Aurora is implemented based on the official code of Aurora (Bodnar et al., 2025). As shown in the Table 20, we employ the following default hyperparameters for Aurora in all of our experiments.

Table 20: Default hyperparameters of Aurora

| Hyperparameter | Meaning | Value |
|---|---|---|
| Patch size | The patch size to embed the input to the token | 4 |
| Hidden dimension | Embedding dimension size | 512 |
| Encoder depths | The number of blocks per encoder layer | (6, 10, 8) |
| Decoder depths | The number of blocks per decoder layer | (8, 10, 6) |
| Heads | The number of attention heads | 16 |
| MLP ratio | MLP hidden dimension ratio | 4.0 |
| Encoder depth | The number of Perceiver (Jaegle et al., 2021) blocks in encoder | 1 |
| Decoder depth | The number of Perceiver (Jaegle et al., 2021) blocks in decoder | 1 |
| Latent levels | The number of latent pressure levels | 4 |
| Window size | 3D Swin window dimensions | (2, 6, 12) |
| Drop path | For stochastic depth rate (Huang et al., 2016) | 0 |
| Dropout | Dropout rate | 0 |

### E.1.7 PRITHXI WxC

The Prithvi WxC is implemented based on the official code of Prithvi-WxC (Schmude et al., 2024). As shown in the Table 21, we employ the following default hyperparameters for Prithvi WxC in all of our experiments.

Table 21: Default hyperparameters of Prithvi WxC

| Hyperparameter | Meaning | Value |
|---|---|---|
| Patch size | The patch size to embed the input to tokens | (2, 2) |
| Hidden dimension | Embedding dimension size | 2560 |
| Encoder blocks | The number of local-global transformer pairs | 12 |
| Heads | The number of attention heads | 16 |
| MLP ratio | MLP hidden dimension ratio | 4.0 |
| Drop path | For stochastic depth rate (Huang et al., 2016) | 0.0 |
| Dropout | Dropout rate | 0.0 |

### E.2 PEFT METHODS

The PEFT methods are implemented within the backbone of Aurora (Bodnar et al., 2025), which is first loaded with the official pretrained weights on over a million hours of diverse weather and climate data, and Prithvi WxC (Schmude et al., 2024), which is first loaded with official pretrained weights of the backbone.

### E.2.1 WEATHERPEFT

As shown in Table 22, we depict some hyperparameter values in our experiment. We denote the downscaling, post-processing, and forecasting tasks as Tasks 1, 2, and 3, respectively.

Table 22: Default hyperparameters of WeatherPEFT

| Hyperparameter | Module | Meaning | Value (Task 1/2/3) |
|---|---|---|---|
| $P$ | TADP | The number of soft prompt tokens | 30/5/20 |
| $P_h$ | TADP | The heigth of the patch embedding's window | 4/4/4 |
| $P_w$ | TADP | The width of the patch embedding's window | 4/4/4 |
| $V$ | TADP | The number of input variables | 11/21/13 |
| $D$ | TADP | The hidden dimension of the encoder's embedding layer | 512/512/512 |
| $HW_h$ | TADP | The hidden dimension of HW-Adapter | 8/8/8 |
| $V_h$ | TADP | The hidden dimension of V-Adapter | 5/10/6 |
| $D_h$ | TADP | The hidden dimension of D-Adapter | 16/16/16 |
| $E_h$ | TADP | The hidden dimension of $E^{VP_hP_w \times D}$-Adapter | 16/16/16 |
| $k$ | SFAS | The percentage of selected parameters | 0.001/0.001/0.001 |
| $\gamma$ | SFAS | The initial value of linear decay factor | 0.2/0.2/0.2 |

### E.2.2 OTHER PEFT BASELINES

We implement six state-of-the-art PEFT methods, including LoRA (Hu et al., 2022), DoRA (Liu et al., 2024), AdaptFormer (Chen et al., 2022), SSF (Lian et al., 2022), VPT (Jia et al., 2022), and APrompt (Wang et al., 2023a), based on their original paper. The default hyperparameters in our experiment are listed in Table 23.

## F ADDITIONAL DOWNSTREAM TASK DETAILS

### F.1 EXPERIMENTAL SETTINGS

We train all the models and WeatherPEFT using the same training framework. Each model is trained with the AdamW optimizer, employing a weight decay of 0.05. We employ a cosine learning rate

Table 23: Default hyperparameters of PEFT baselines.

| Method | Hyperparameter | Meaning | Value |
|---|---|---|---|
| LoRA | Rank | The rank of the low rank matrix | 8 |
| LoRA | Alpha | The alpha value | 1 |
| LoRA | Dropout | Dropout rate | 0 |
| DoRA | Rank | The rank of the low rank matrix | 8 |
| DoRA | Alpha | The alpha value | 1 |
| DoRA | Dropout | Dropout rate | 0 |
| AdaptFormer | Skip connection | Whether to use residual connection within the adapter | False |
| AdaptFormer | Mlp ratio | The ratio of down sample | 0.25 |
| AdaptFormer | Activation function | The activation function within the adapter | GELU |
| SSF | Layer number | The number of SSF layer | 12 |
| VPT | Prompt length | The number of soft prompt tokens | 50 |
| APrompt | Prompt length | The number of soft prompt tokens | 50 |
| APrompt | QKV length | The number of soft attention tokens | 10 |

scheduler with a warm-up phase during the first three epochs to stabilize training. For the three distinct downstream tasks, models are trained on eight 80GB NVIDIA A800 GPUs. The specific parameters for these tasks are: learning rates of 7e-4, 1e-3, and 3e-3; batch sizes of 5, 1, and 4; and 30, 10, and 15 training epochs, respectively. The approximate training times for these respective configurations are 6, 2, and 2 hours. In the subsection, we will elaborate on the details of the implementation of model architectures and PEFT methods.

## F.2 DOWNSCALING

Global weather forecasting models typically operate at coarse spatial resolutions to mitigate computational costs, capturing large-scale atmospheric dynamics at the expense of localized detail. However, such resolutions are insufficient for analyzing regional phenomena such as coastal wind patterns. Downscaling, or statistical super-resolution, addresses this limitation by enhancing coarse-grained model outputs to finer resolutions while preserving physical consistency. In this experiment, we downscale 5.625° ERA5 data to 1.40625° ERA5 data (Hersbach et al., 2020) both at a global scale and 6-hour intervals, leveraging the WeatherBench dataset (Rasp et al., 2020). The training involves 30 epochs over the period from 2007 to 2016, and the test is in 2017 and 2018. Following Nguyen et al. (2023a;b), we first bilinearly interpolate the input to match the resolution of the desired output before feeding it to the model. We use mean square error as the loss function, and the overall surface loss is weighted by 0.25, while the overall upper loss is weighted by 1, following (Bi et al., 2023; Bodnar et al., 2025).

### F.2.1 DATA

Table 24 summarizes the variables we use for our experiments, which total 68 variables.

Table 24: ERA5 variables used in our experiments. Surface represents surface variables, and Upper represents atmospheric properties at the chosen altitudes.

| Type | Variable | Abbrev. | Levels |
|---|---|---|---|
| Surface | 2 metre temperature | T2m | |
| Surface | 10 metre U wind component | U10 | |
| Surface | 10 metre V wind component | V10 | |
| Upper | Geopotential | Z | |
| Upper | U wind component | U | 50, 100, 150, 200, 250, |
| Upper | V wind component | V | 300, 400, 500, 600, 700, |
| Upper | Temperature | T | 850, 925, 1000 |
| Upper | Specific humidity | Q | |

### F.2.2 PROBLEM SETTING

In this $5.625°$ ERA5 data to $1.40625°$ downscaling experiment, the $5.625°$ input data $\mathbf{X} \in \mathbb{R}^{68 \times 32 \times 64}$ is first bilinearly interpolated to $1.40625°$ data $\hat{\mathbf{X}} \in \mathbb{R}^{68 \times 128 \times 256}$ following Nguyen et al. (2023a;b). The machine learning models are trained to correct the biases between the interpolated input data $\hat{\mathbf{X}}$ and ground truth $1.40625°$ data $\mathbf{Y} \in \mathbb{R}^{68 \times 32 \times 64}$.

### F.2.3 VISUALIZATION

We visualize the input, ground truth, and prediction of seven PEFT approaches (our proposed WeatherPEFT and six other state-of-the-art PEFT baselines) to provide an intuitive comparison for further reference.

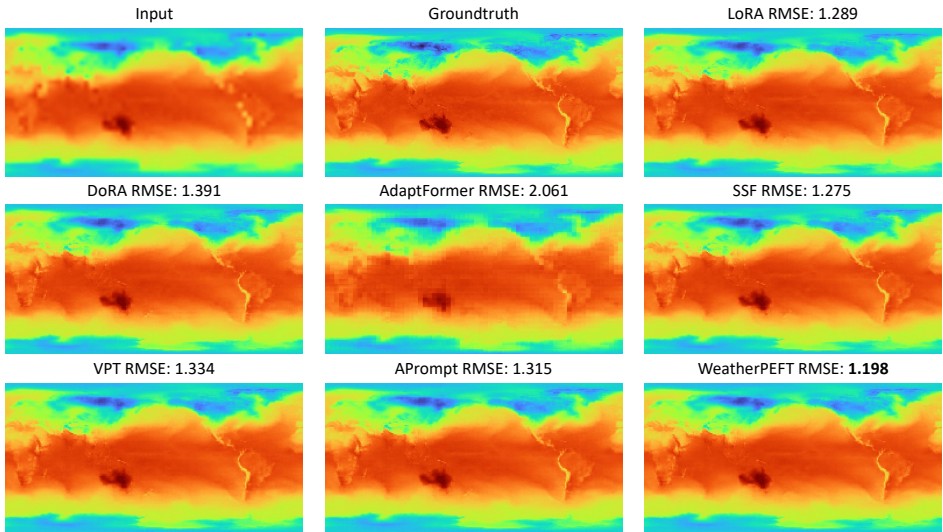

Figure 4: Visualization of PEFT baselines and WeatherPEFT on the variable T2m of downscaling (2018-01-11 06 UTC).

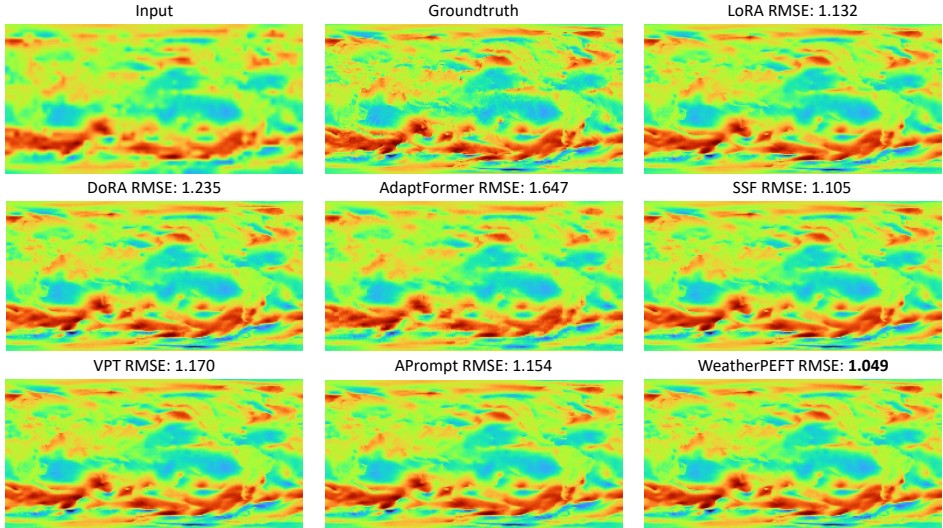

Figure 5: Visualization of PEFT baselines and WeatherPEFT on the variable U10 of downscaling (2017-08-14 06 UTC).

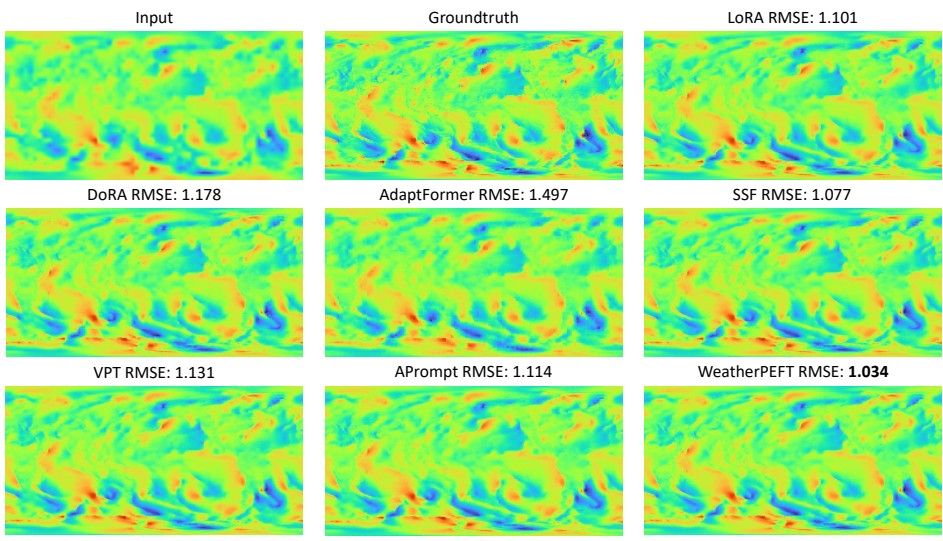

Figure 6: Visualization of PEFT baselines and WeatherPEFT on the variable V10 of downscaling (2017-08-14 06 UTC).

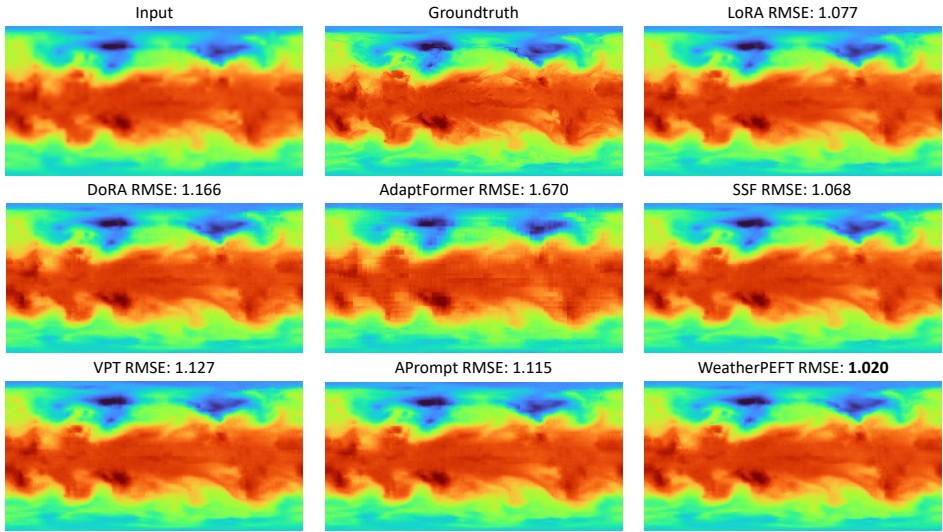

Figure 7: Visualization of PEFT baselines and WeatherPEFT on the variable T850 of downscaling (2018-01-11 06 UTC).

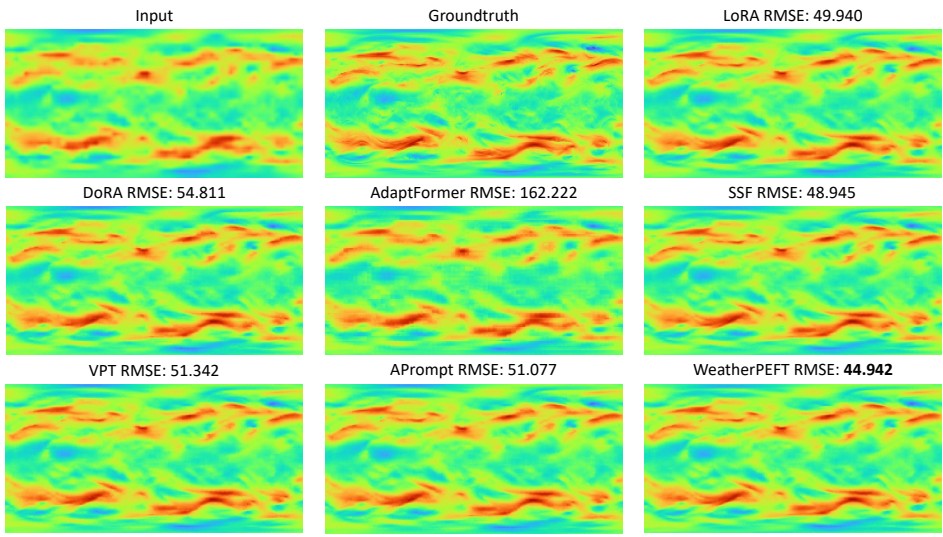

Figure 8: Visualization of PEFT baselines and WeatherPEFT on the variable Z500 of downscaling (2018-03-27 06 UTC).

### F.3 ENSEMBLE WEATHER FORECAST POST-PROCESSING

Existing ensemble weather predictions are subject to systematic errors known as biases (Toth & Kalnay, 1993). Therefore, post-processing approaches have been introduced to forecast skill by correcting the distribution of the ensemble weather prediction to improve the reliability of weather forecasting. Our evaluation employs the ENS-10 benchmark (Ashkboos et al., 2022) for global ensemble forecast post-processing, which pairs 10-member ensemble prediction (48-hour lead time) from the ECMWF Integrated Forecasting System (IFS) (ECMWF, 2022) with ERA5 reanalysis targets at $0.5°$ resolution. The dataset involves two data points per week spanning 20 years, with the years 1998-2015 as the training set and 2016-2017 as the test set. Following (Ashkboos et al., 2022), we utilize the closed-form expression of the Continuous Ranked Probability Score (CRPS) as the loss function, training for 10 epochs.

#### F.3.1 DATA

Table 25 summarizes the variables we use for our experiments, which total 25 variables.

Table 25: ENS-10 variables used in our experiments. Surface represents surface variables, and Upper represents atmospheric properties at the chosen altitudes.

| Type | Variable | Abbrev. | Levels |
|---|---|---|---|
| Surface | Sea surface temperature | SST | |
| Surface | Total column water | TCW | |
| Surface | Total column water vapor | TCWV | |
| Surface | Convective precipitation | CP | |
| Surface | Mean sea level pressure | MSL | |
| Surface | Total cloud cover | TCC | |
| Surface | Skin temperature at surface | SKT | |
| Surface | Total precipitation | TP | |
| Surface | 2 metre temperature | T2m | |
| Surface | 10 metre U wind component | U10 | |
| Surface | 10 metre V wind component | V10 | |
| Upper | Geopotential | Z | |
| Upper | U wind component | U | |
| Upper | V wind component | V | |
| Upper | Temperature | T | 500, 850 |
| Upper | Specific humidity | Q | |
| Upper | Vertical velocity | W | |
| Upper | Divergence | D | |

#### F.3.2 PROBLEM SETTING

For a given time T, the input is a set of ensemble members $X = \{\mathbf{X}_{k,T}\}_{k \in [1,\underline{10}]}$. Each ensemble member $\mathbf{X}_{k,T} \in \mathbb{R}^{25 \times 360 \times 720}$ consists of all surface and upper variables predictions at time steps $T + 24h$. For each target variable, the task is to predict a corrected cumulative distribution function (CDF) $F_{ij}$ at time $T + 48h$ at each grid point $(i, j)$. Following Toth & Kalnay (1993); Grönquist et al. (2021), we assume a Gaussian distribution on the target variable and learn the mean and standard deviation of this distribution. Specifically, the model is provided with the mean and standard deviation of all variables in ENS-10 at a lead time of $T + 48h$. The model outputs two values corresponding to the mean and standard deviation of the target variable. To derive the corrected mean, the first output value is multiplied by the ensemble member's standard deviation and added to the ensemble mean. Similarly, the corrected standard deviation is obtained by taking the exponential of the second output value and multiplying it by the ensemble standard deviation. This normalization ensures accurate calibration of the predicted distribution. We choose to minimize the Continuous Ranked Probability Score (CRPS) between the ensemble prediction and ERA5 ground-truth. In this case, the closed-form expression of CRPS of a Gaussian distribution (Ashkboos et al., 2022) can be

defined as:

$$\mathrm{CRPS}(F_{i,j}, \mathbf{X}) = \sigma \left[ 2\psi \left( \frac{\mathbf{X} - \mu}{\sigma} \right) + \frac{\mathbf{X} - \mu}{\sigma} \left( 2\phi \left( \frac{\mathbf{X} - \mu}{\sigma} \right) - 1 \right) - \frac{1}{\sqrt{\pi}} \right], \qquad (9)$$

where $\mu$ and $\sigma$ are the mean and standard deviation of the distribution, $\psi$ and $\phi$ are the probability density and cumulative density function of a standard Gaussian random variable, respectively.

### F.4 REGIONAL PRECIPITATION FORECASTING

Precipitation forecasting plays a crucial role in agriculture, water resource management, and disaster prevention (Yue et al., 2022; Ward et al., 2011). Among fundamental atmospheric forecast variables, precipitation forecasting presents unique challenges. This is primarily attributed to the multiscale interactions involved in precipitation processes, ranging from cloud microphysics to large-scale circulation (Frank et al., 2024), encompassing complex nonlinear dynamical, water vapor transport, and thermodynamic processes (Trenberth et al., 2003). Moreover, global predictions are not always feasible, particularly when only regional data is available. In this experiment, we evaluate Weather-PEFT on regional six-hour precipitation accumulation forecasts across China, addressing scenarios where only localized observational data is available. To enable this assessment, we introduce ERA5-CH, a specialized dataset derived from ERA5 reanalysis at resolution $0.25°$ exclusively over China. To do this, we first identified the latitude ($58.5°$N-$1.5°$S) and longitude ($74.0°$E-$134.0°$E) range to form a rectangular area that encapsulates China. For each data sample, we then extracted the spatial positions that fall into this range, forming ERA5-CH. We utilize the mean absolute error loss for training and train the model over 15 epochs, with data from 2010–2019 serving as the training set and 2020 as the test set. Both datasets are configured with a 12-hour temporal resolution.

### F.4.1 DATA

Table 24 summarizes the variables we use for our experiments, which total 70 variables.

Table 26: ERA5 variables used in our experiments. Surface represents surface variables, and Upper represents atmospheric properties at the chosen altitudes.

| Type | Variable | Abbrev. | Levels |
|------|----------|---------|--------|
| Surface | Total precipitation of 6 hours | TP | |
| Surface | Mean sea level pressure | MSL | |
| Surface | 2 metre temperature | T2m | |
| Surface | 10 metre U wind component | U10 | |
| Surface | 10 metre V wind component | V10 | |
| Upper | Geopotential | Z | |
| Upper | U wind component | U | 50, 100, 150, 200, 250, |
| Upper | V wind component | V | 300, 400, 500, 600, 700, |
| Upper | Temperature | T | 850, 925, 1000 |
| Upper | Relative humidity | R | |

### F.4.2 PROBLEM SETTING

In this regional precipitation forecasting experiment, the input $\mathbf{X} \in \mathbb{R}^{70 \times 240 \times 240}$ is $0.25°$ data with 70 variables and $240 \times 240$ grids. The machine learning models are trained to predict the six-hour accumulation of precipitation for three lead times of 12 hours, 24 hours, and 36 hours, which is also $0.25°$ data $\mathbf{Y} \in \mathbb{R}^{3 \times 240 \times 240}$ with $240 \times 240$ grids.

### F.4.3 VISUALIZATION

We provide the visualization of PEFT baselines and WeatherPEFT on the variable TP (total precipitation) in Figure 9.

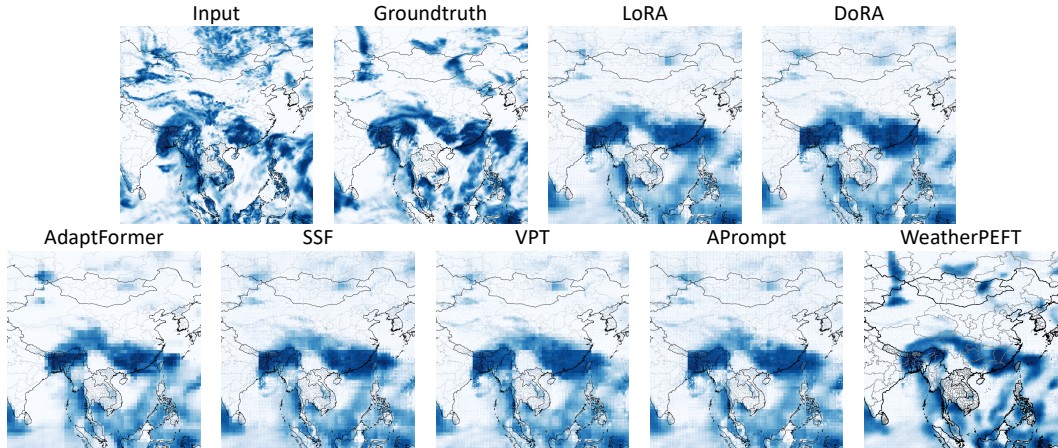

Figure 9: PEFT baselines and WeatherPEFT visualization of a 12-hour forecast for TP-6hr over China (2020-05-20 12 UTC).

## F.5 METRICS

This section defines all the evaluation metrics we employ in the experiment. For arbitrarily variable, we denote $\hat{\mathbf{Y}} \in \mathbb{R}^{N \times H \times W}$ and $\mathbf{Y} \in \mathbb{R}^{N \times H \times W}$ and $\mathbf{Y}$ as the prediction output and the ground truth, both of which have the same shape, where $N$ represents the number of data points, $H$ denotes the number of latitude coordinates, and $W$ is the number of longitude coordinates. $\hat{y}_{k,i,j}$ and $y_{k,i,j}$ indicates scalar values of the prediction tensor $\hat{\mathbf{Y}}$ and the ground-truth tensor $\mathbf{Y}$, respectively. The indices $k$, $i$, and $j$ correspond to the data sample, latitude, and longitude.

### F.5.1 ROOT MEAN SQUARED ERROR (RMSE)

Following WeatherBench, we define the RMSE as the mean latitude-weighted RMSE over all forecasts for each variable:

$$\text{RMSE} = \frac{1}{N} \sum_{k=1}^{N} \sqrt{\frac{1}{H \times W} \sum_{i=1}^{H} \sum_{j=1}^{W} W(i)(\hat{y}_{k,i,j} - y_{k,i,j})^2}, \tag{10}$$

where $W(i)$ is is the latitude weighting factor for the latitude at $i^{th}$ latitude index:

$$W(i) = \frac{\cos(\text{lat}(i))}{\frac{1}{N_{\text{lat}}} \sum_i^{N_{\text{lat}}} \cos(\text{lat}(i))}. \tag{11}$$

### F.5.2 MEAN BIAS

Mean bias quantifies the discrepancy between the spatial average of predictions and ground truth. A positive value indicates systematic overestimation, while a negative value reflects an underestimation of the mean. The Mean Bias for each variable is defined as:

$$\text{Mean Bias} = \frac{1}{N \times H \times W} \sum_{k=1}^{N} \sum_{i=1}^{H} \sum_{j=1}^{W} (\hat{y}_{k,i,j} - y_{k,i,j}). \tag{12}$$

### F.5.3 CONTINUOUS RANKED PROBABILITY SCORE (CRPS)

CRPS generalizes the mean absolute error for probabilistic forecasts. Given a ground truth observation $y$ at grid-point $(i, j)$, the CRPS for the corrected cumulative distribution function $F$ at the same point is defined as:

$$\text{CRPS}(F_{ij}, y) = \int_{-\infty}^{\infty} \left(F_{ij}(x) - \mathbf{1}_{y \leq x}\right)^2 dx, \tag{13}$$

where $\mathbf{1}_{y \leq x}$ is an indicator function that equals 1 if $y \leq x$ and 0 otherwise. This formulation quantifies the discrepancy between the predicted cumulative distribution function and the observed value, providing a robust measure of probabilistic forecast accuracy. We report the mean CRPS over all grid points over the two test years.

### F.5.4 ANOMALY CORRELATION COEFFICIENT (ACC)

ACC measures the spatial correlation between the anomalies of prediction $\hat{\mathbf{Y}}$ and ground truth $\mathbf{Y}$, where both are computed relative to climatological baselines. Formally, ACC is defined as:

$$\text{ACC} = \frac{\sum_{k,i,j} W(i)\hat{y}'_{k,i,j} y'_{k,i,j}}{\sqrt{\sum_{k,i,j} W(i)\hat{y}'^2_{k,i,j} \sum_{k,i,j} W(i) y'^2_{k,i,j}}}, \tag{14}$$
$$\hat{\mathbf{Y}}' = \hat{\mathbf{Y}} - \mathbf{C}, \mathbf{Y}' = \mathbf{Y} - \mathbf{C},$$

where climatology $\mathbf{C}$ is the temporal mean of the ground truth data over the dataset.

### F.5.5 EXTREME EVENT WEIGHTED CONTINUOUS RANKED PROBABILITY SCORE (EECRPS)

A critical objective in bias correction is reducing uncertainty during extreme weather events. To avoid conflating these events with average-case forecast skill, (Ashkboos et al., 2022) introduces a weighted version of CRPS that emphasizes extreme conditions. A widely adopted metric for quantifying forecast irregularity is the Extreme Forecast Index (EFI) (Lalaurette, 2003; Zsótér, 2006), which measures the deviation of ensemble forecasts relative to a probabilistic weather model. The EFI ranges between -1 and 1, with larger absolute values indicating greater deviation from historical meteorological records. Typically, EFI magnitudes between 0.5 and 0.8 are considered unusual, while values above 0.8 signify very unusual conditions and a high likelihood of extreme weather. Given a ground-truth observation y at grid-point $(i, j)$, we weight the CRPS using the absolute value of the EFI at that location, defining the Extreme Event Weighted CRPS (EECRPS) as:

$$\text{EECRPS}(F_{i,j}, y) := |\text{EFI}_{(i,j)}| \times \text{CRPS}(F_{i,j}, y). \tag{15}$$

We report the mean EECRPS over all grid points of the test years. For the calculation of $EFI(i, j)$, please refer to (Ashkboos et al., 2022)

### F.5.6 STABLE EQUITABLE ERROR IN PROBABILITY SPACE (SEEPS)

Traditional deterministic metrics such as RMSE and ACC are inadequate for evaluating precipitation forecasts due to precipitation's highly skewed distribution and spatiotemporal intermittency. These limitations cause conventional metrics to favor overly smooth forecasts. Following (Rasp et al., 2020), we adopt the SEEPS score (Rodwell et al., 2010) for precipitation evaluation. SEEPS categorizes precipitation into three classes: "dry," "light," and "heavy," discouraging smooth forecasts while maintaining stability across parameter choices. For more details about the SEEPS score, please refer to (Rodwell et al., 2010). Here, we describe how we compute the SEEPS score based on (Rasp et al., 2024). For every location, we use a dry threshold of 0.1 mm/day for 6 hourly accumulations. The remaining precipitation values are split into light and heavy categories, with light precipitation days occurring twice as frequently as heavy ones for that location climatologically. We utilize the light-heavy threshold precomputed by (Rasp et al., 2024), which is the 2/3rd quantile of non-dry days based on climatology (Rasp et al., 2024). Forecast-observation pairs are classified into these categories based on the thresholds, generating a $3 \times 3$ joint probability contingency table (Table 27) for each lead time.

The contingency table is then multiplied by the scoring error matrix $S$ based on the climatological occurrence of dry days for each geographical location:

$$S = \frac{1}{2} \begin{bmatrix} 0 & \frac{1}{1-p} & \frac{4}{1-p} \\ \frac{1}{p} & 0 & \frac{3}{1-p} \\ \frac{1}{p} + \frac{3}{2+p} & \frac{3}{2+p} & 0 \end{bmatrix} \tag{16}$$

where $p$ represents the climatological probability of dry days, columns represent observed probabilities, and rows represent forecast probabilities. Following (Zhao et al., 2024b; Rodwell et al., 2010),

Table 27: $3 \times 3$ contingency table of precipitation classification forecast and observation in SEEPS scores.

| Probability | | Observation | | |
|---|---|---|---|---|
| | Category | 1 | 2 | 3 |
| | 1 | $P_{11}$ | $P_{12}$ | $P_{13}$ |
| Forecast | 2 | $P_{21}$ | $P_{22}$ | $P_{23}$ |
| | 3 | $P_{31}$ | $P_{32}$ | $P_{33}$ |

we exclude extreme climates using $0.1 < p < 0.85$ and compute area-weighted mean SEEPS scores. It can be seen in Equation 16 that the SEEPS error scoring matrix is uniquely determined by $p$. For rainy climate regions, where $p$ is smaller, the lower triangular elements of the SEEPS error scoring matrix (corresponding to false negatives for "dry" conditions) are larger. For arid climate regions, where $p$ is larger, the upper triangular elements of the SEEPS error scoring matrix (corresponding to false negatives for "heavy rain") are larger. This indicates that the SEEPS error scoring matrix, which is based on the probability of precipitation occurrence $(1 - p)$, varies across different climate regions or precipitation seasons. Consequently, a key feature of SEEPS is its ability to assign different error scores to the same forecast characteristic (e.g., missing a "heavy rain" event) depending on the climate region or season. In other words, the "penalty" for forecast errors is tied to the climatic probability of precipitation. Thus, SEEPS automatically adapts to site-specific precipitation probabilities across varying climate zones or seasons.

### F.5.7 THREAT SCORE (TS)

The Threat Score (TS), also known as the Critical Success Index (CSI), is a widely used verification metric in meteorology for evaluating the performance of categorical forecasts, particularly for precipitation events (Schaefer, 1990). It measures the fraction of correctly predicted "yes" events out of all instances where the event was either predicted or observed. The TS is particularly valuable as it ignores correct negatives (correctly forecasting no event), making it sensitive to performance on rare or localized phenomena like heavy rainfall.

To calculate the TS, forecast-observation pairs at each grid point are first categorized into a contingency table based on a predefined event threshold. The categories are Hits ($H$), where the event was forecast to occur and did occur, Misses ($M$), where the event was not forecast to occur but did occur, and False Alarms (F), where the event was forecast to occur but did not occur. The Threat Score is then computed using the following formula:

$$TS = \frac{H}{(H + M + F)}.$$ (17)

The score ranges from 0 to 1, where 1 indicates a perfect forecast. In the context of our case study on the 2020 Mei-yu flood, we use percentile-based thresholds to define the precipitation events, allowing for a location-specific evaluation of moderate and heavy rainfall. Specifically, we establish two thresholds for each grid point based on a climatology constructed from precipitation data in June and July between 2010 and 2020:

- **50th Percentile TS**: An event is defined as precipitation exceeding the local 50th percentile of the climatology.

- **75th Percentile TS**: An event is defined as precipitation exceeding the local 75th percentile of the climatology.

This approach ensures that the metric evaluates the model's ability to predict rainfall events that are significantly intense relative to the typical climate of each specific location during that season.

