# OpenReview forum: "Task-Adaptive Parameter-Efficient Fine-Tuning for Weather Foundation Models"
_ICLR.cc/2026/Conference — ICLR 2026 Poster_

### Official Review · Reviewer_L3dy · 2025-10-23

**Soundness:** 3
**Presentation:** 3
**Contribution:** 2
**Rating:** 6
**Confidence:** 4

**Summary:**

Weather foundation models (WFMs) are important.

PEFT failed to address the challenges of weather downstream tasks, such as variable heterogeneity, resolution diversity, and spatiotemporal coverage variations, leading to suboptimal performance when applied to WFMs.

This paper introduces WeatherPEFT, which is tailed PEFT for WFMs.

**Strengths:**

This paper has done comprehensive study to make the WeatherPEFT works for the weather foundation models. Compared to the standard PEFT like LoRA, WeatherPEFT surpasses them and approaches full-tuning performance even with a minimal parameter budget e.g. ~0.3%.

**Weaknesses:**

One concern with this work is that the proposed method lacks fair evaluation. From the proposed method, it does not seem to be unique to weather foundation models at all. It seems that the proposed method would work to some extent on the other problems too. So it seems that the evaluation only on the weather foundation model is not a fair comparison.

**Questions:**

1. Can you apply the proposed PEFT method on any dataset outside of the weather domain?

2. How sensitive is the proposed HW-adaptive in weatherPEFT to the different resolution parameter?

---

> ### Author Response · Authors · 2025-11-21
> **Author Responses (1/2)**
>
> We thank the reviewer for recognizing the importance of WFMs as well as the strong performance of WeatherPEFT. We appreciate your constructive feedback regarding the fairness of evaluation and the generalizability of our method. We try our best to eliminate the question via the following response.
>
> > Weakness 1 & Question 1 (Motivation): The evaluation based only on the weather foundation model is not a fair comparison.
>
> We appreciate the reviewer’s rigorous inquiry regarding the generalizability of our method and the fairness of our evaluation. We acknowledge the premise that the architectural components of WeatherPEFT are mathematically compatible with general domains. However, we posit that the structural composition of TADP and the selection mechanisms of SFAS are optimized for the physical heterogeneity of meteorological data. And our work is indeed driven by the unique challenges posed by applying PEFT to weather tasks.
>
> To substantiate that our method is specifically tailored for the meteorological challenge, we further compare WeatherPEFT with general vision PEFT baselines (including additional Convpass [1], FacT-TT [2], and MoA (Mixture of Adapter)) on a standard computer vision task. Specifically, we evaluate performance on the Cityscapes [3] $\to$ ACDC [4] domain generalization benchmark for semantic segmentation, which encompasses the Night, Snow, Fog, and Rain domains. We utilize Dinov2-L [5] as the backbone for these experiments and report the mean Intersection over Union (mIoU). Additionally, we present corresponding results for the regional precipitation forecasting task. The results are as follows:
> |Methods|Params (M)|Night|Snow|Fog|Rain|Mean|
> |-|:-:|:-:|:-:|:-:|:-:|:-:|
> |Full-Tuning|304.20|52.4|70.5|80.9|74.4|69.5|
> |Linear-Probing|0.00|54.3|69.3|79.1|68.0|67.6|
> |Convpass|3.64|**56.0**|71.7|80.2|**74.9**|70.7|
> |FacT-TT|2.85|56.1|71.3|81.0|72.9|70.3|
> |MOA|6.39|53.2|70.6|80.3|72.8|69.3|
> |LoRA|3.14|52.3|74.4|79.5|74.0|70.1|
> |AdaptFormer|3.17|53.8|**74.8**|80.3|74.6|**70.9**|
> |VPT|3.15|53.4|74.4|80.4|70.5|69.7|
> |Ours|2.90|**56.0**|70.9|**81.2**|74.5|70.7|
>
> |Method|Params (M)|12h-SEEPS|12h-ACC↑|12h-RMSE|24h-SEEPS|24h-ACC↑|24h-RMSE|36h-SEEPS|36h-ACC↑|36h-RMSE|
> |-|:-:|:-:|:-:|:-:|:-:|:-:|:-:|:-:|:-:|:-:|
> |Linear-Probing|0.00|0.581|0.464|0.266|0.720|0.265|0.293|0.790|0.171|0.303|
> |Convpass|4.92|0.485|0.606|0.237|0.615|0.439|0.269|0.697|0.326|0.285|
> |FacT-TT|2.73|0.525|0.553|0.249|0.662|0.371|0.279|0.747 |0.246|0.294|
> |MoA|8.62|0.515|0.563|0.246|0.665|0.354|0.281|0.749|0.235|0.296|
> |LoRA|3.63|0.495|0.592|0.240|0.634|0.415|0.273|0.723|0.294|0.289|
> |AdaptFormer|4.62|0.499|0.577|0.243|0.643|0.378|0.278|0.731|0.258|0.293|
> |VPT|3.75|0.522|0.550|0.250|0.666|0.356|0.281|0.750|0.235|0.296|
> |**Ours**|3.38|**0.368**|**0.742**|**0.198**|**0.515**|**0.559**|**0.247**|**0.615**|**0.443**|**0.268**|
>
> From the results, we observe that WeatherPEFT performs comparably to generic methods, such as AdaptFormer, on standard vision tasks. However, on the weather task, WeatherPEFT significantly outperforms all PEFT baselines. This validates our motivation: standard PEFTs are insufficient for weather foundation models, necessitating the domain-specific design we propose. We will incorporate these results into the revised manuscript.
>
> [1] Convolutional Bypasses Are Better Vision Transformer Adapters. ECAI. 2024.
>
> [2] Fact: Factor-tuning for lightweight adaptation on vision transformer. AAAI. 2023.
>
> [3] The cityscapes dataset for semantic urban scene understanding. CVPR. 2016.
>
> [4] ACDC: The adverse conditions dataset with correspondences for semantic driving scene understanding. ICCV. 2021.
>
> [5] Dinov2: Learning robust visual features without supervision. TMLR. 2024.

---

> > ### Author Response · Authors · 2025-11-21
> > **Author Responses (2/2)**
> >
> > > Question 2 (Sensitivity): How sensitive is the HW-adaptive to the different resolution parameter?
> >
> > We appreciate the reviewer’s insightful question regarding the sensitivity of the HW-Adapter to resolution-related parameters. We interpret "resolution parameter" from two critical perspectives: (1) the internal hyperparameters governing the adapter's capacity (i.e., hidden dimensions), and (2) the adapter's ability to generalize across diverse input resolutions found in different downstream tasks.
> >
> > We are pleased to clarify that our proposed HW-Adapter is designed to be robust and parameter-efficient, demonstrating low sensitivity to increases in hyperparameter size while effectively adapting to diverse input resolutions. The supporting mechanism and empirical evidence are detailed below:
> > 1. **Mechanism of Adaptation:** The HW-Adapter within the TADP module does not operate on fixed spatial feature maps that might be brittle to resolution changes. Instead, it processes the embedding weights $E \in \mathbb{R}^{D \times V \times P_h \times P_w}$ of the encoder. By extracting patterns from the kernel dimensions ($P_h \times P_w$), the HW-Adapter learns the local spatial interaction rules defined by the patch size and the grid resolution. This allows the model to extract resolution-specific context (e.g., how a $5.625^{\circ}$ patch interacts vs. a $0.25^{\circ}$ patch) and inject this as a prompt, effectively conditioning the backbone on the current resolution.
> > 2. **Robustness to Internal Hyperparameters:** The hyperparameter ablation study in Appendix B.1 (Table 4) compare our efficient default configuration against a significantly larger parameterization: Hidden dimensions $(HW_h, V_h, D_h) = (8, 6, 16)$ vs. $(HW_h, V_h, D_h) = (32, 13, 512)$. The study reveal that increasing the hidden dimensions do not yield performance benefits. This indicates that the HW-Adapter is not sensitive to the scaling of its parameters in a way that requires fine-grained tuning for larger capacities
> > 3. **Robustness Across Diverse Resolutions:** Our experimental setup is deliberately chosen to span a wide range of spatial resolutions, confirming that the HW-Adapter functions effectively across distinct scales without requiring architectural changes: Global Downscaling (Coarse Scale), which involves map the input at 5.625° to high-resolution outputs at 1.40625°, Post-Processing (Medium Scale), which operates on 0.5° resolution global data. Regional Precipitation Forecasting (Fine Scale), which processes high-resolution 0.25° data over a region. The consistent state-of-the-art performance across these tasks validates that the HW-Adapter is robust to input resolution variations, serving as a scale-invariant feature extractor.
> >
> > **We sincerely appreciate your insightful feedback and have addressed your concerns to the best of our ability. Should you have any further questions or recommendations, we would be glad to engage in further discussion.**

---

> > ### Comment · Reviewer_L3dy · 2025-11-25
> >
> > Thanks authors to response my comments.
> >
> > I think some of my concerns are addressed but not all of them. In particular, the following comments are not addressed.
> >
> > Can you please evaluate your proposed adapter on the generic tasks on the generic LORA datasets? e.g. VTAB-1K.

---

> ### Author Response · Authors · 2025-11-25
> **Looking forward to your feedback**
>
> Dear Reviewer L3dy,
>
> We are eager to ensure that we have adequately addressed your concerns and are prepared to offer further clarifications or address any additional questions you may have.
>
> We would like to express our heartfelt gratitude for the time and effort you have dedicated to reviewing our work. It has been a pleasure to engage with you throughout this process.
>
> Best regards,
>
> The authors

---

> ### Author Response · Authors · 2025-11-27
> **Author Responses**
>
> We thank the reviewer for the continued engagement and for raising the score to 8. We agree that VTAB-1K provides a valuable reference point for understanding the behavior of our method in a standard computer vision context. As requested, we evaluate WeatherPEFT on the VTAB-1K benchmark [1], using a ViT-B/16 [2] backbone pre-trained on ImageNet-21k [3]. We compare our method with representative visual PEFT methods, including Convpass [4], FacT [5], RepAdapter [6], SSF [7], SCT [8], LoRA [9], AdaptFormer [10], and VPT [11].
>
> |Method|Params (M)|Cif.|Cal.|DTD|Flo.|Pets|SVHN|Sun.|Cam.|Eur.|Res.|Ret.|CleC.|CleD.|DMLab|KIT.|dSpL.|dSpO.|sNOA.|sNOE.|Avg.|
> |-|:-:|:-:|:-:|:-:|:-:|:-:|:-:|:-:|:-:|:-:|:-:|:-:|:-:|:-:|:-:|:-:|:-:|:-:|:-:|:-:|:-:|
> |Full-Tuning|85.8|68.9|87.7|64.3|97.2|86.9|87.4|38.8|79.7|95.7|84.2|73.9|56.3|58.6|41.7|65.5|57.5|46.7|25.7|29.1|68.9|
> |Linear-Probing|0.00|64.4|85.0|63.2|97.0|86.3|36.6|51.0|78.5|87.5|68.5|74.0|34.3|30.6|33.2|55.4|12.5|20.0|9.6|19.2|57.6|
> |Convpass|0.33|72.3|91.2|72.2|99.2|90.9|**91.3**|54.9|84.2|96.1|85.3|75.6|82.3|67.9|51.3|80.0|**85.9**|53.1|36.4|44.4|**76.6**|
> |FacT-TK|0.07|70.6|90.6|70.8|99.1|90.7|88.6|54.1|84.8|**96.2**|84.5|75.7|82.6|68.2|49.8|**80.7**|80.8|47.4|33.2|43.0|75.6|
> |RepAdapter|0.22|72.4|91.6|71.0|99.2|91.4|90.7|**55.1**|85.3|95.9|84.6|75.9|82.3|68.0|50.4|79.9|80.4|49.2|**38.6**|41.0|76.1|
> |SSF|0.24|69.0|**92.6**|**75.1**|**99.4**|**91.8**|90.2|52.9|**87.4**|95.9|**87.4**|75.5|75.9|62.3|**53.3**|80.6|77.3|**54.9**|29.5|37.9|75.7|
> |SCT|0.11|75.3|91.6|72.2|99.2|91.1|91.2|55.0|85.0|96.1|86.3|76.2|81.5|65.1|51.7|80.2|75.4|46.2|33.2|**45.7**|76.0|
> |LoRA|0.29|67.1|91.4|69.4|98.8|90.4|85.3|54.0|84.9|95.3|84.4|73.6|**82.9**|**69.2**|49.8|78.5|75.7|47.1|31.0|44.0|74.5|
> |AdaptFormer|0.16|70.8|91.2|70.5|99.1|90.9|86.6|54.8|83.0|95.8|84.4|**76.3**|81.9|64.3|49.3|80.3|76.3|45.7|31.7|41.1|74.7|
> |VPT|0.53|**78.8**|90.8|65.8|98.0|88.3|78.1|49.6|81.8|96.1|83.4|68.4|68.5|60.0|46.5|72.8|73.6|47.9|32.9|37.8|72.0|
> |Ours|0.29|73.1|92.2|71.9|99.2|90.2|89.2|53.5|83.3|95.0|83.4|73.6|81.3|68.0|46.8|74.8|72.3|45.2|28.3|37.6|74.0|
>
> The results demonstrate that WeatherPEFT achieves performance comparable to other general PEFT methods. However, we observe that our method performs slightly below the state-of-the-art on the "Structured" task group (e.g., dSprites, sNORB). We believe this is due to a fundamental difference between the VTAB-1K experimental setting and the weather domains our method was optimized for.
>
> The core design of our Task-Adaptive Dynamic Prompting (TADP) is to extract task-specific characteristics (such as variables and resolutions) from the encoder's embedding layer to introduce context-aware recalibration. In weather tasks, the embedding layer is rich with varying physical information, allowing TADP to dynamically adapt the model to the specific "physics" of the input. In the standard vision task like VTAB-1k, the embedding layer of the backbone is typically frozen and processes homogeneous RGB data. In this frozen setting, TADP can only extract information from a static layer. Consequently, the "dynamic prompt" effectively becomes a static constant. This neutralizes the primary advantage of TADP's adaptivity, resulting in performance that is comparable to, but not significantly better than, other baselines in this specific context.
>
> In summary, while WeatherPEFT is capable of handling generic tasks, its superior performance is unlocked in the weather domain, validating our motivation for a domain-specialized design. We will add these results and this discussion to the appendix of our paper.
>
> [1] A Large-scale Study of Representation Learning with the Visual Task Adaptation Benchmark. arxiv. 2019.
>
> [2] An image is worth 16x16 words: Transformers for image recognition at scale. ICLR. 2021
>
> [3] Imagenet: A large-scale hierarchical image database. CVPR. 2009.
>
> [4] Convolutional Bypasses Are Better Vision Transformer Adapters. ECAI. 2024.
>
> [5] Fact: Factor-tuning for lightweight adaptation on vision transformer. AAAI. 2023.
>
> [6] Towards efficient visual adaption via structural re-parameterization. arXiv. 2023.
>
> [7] Scaling & shifting your features: A new baseline for efficient model tuning[J]. NeurIPS. 2022.
>
> [8] Sct: A simple baseline for parameter-efficient fine-tuning via salient channels. IJCV. 2024.
>
> [9] Lora: Low-rank adaptation of large language models. ICLR. 2022.
>
> [10] Adaptformer: Adapting vision transformers for scalable visual recognition. NeurIPS. 2022.
>
> [11] Visual prompt tuning. ECCV. 2022.

---

### Official Review · Reviewer_maT4 · 2025-10-28

**Soundness:** 3
**Presentation:** 4
**Contribution:** 3
**Rating:** 8
**Confidence:** 3

**Summary:**

The authors propose WeatherPEFT, a parameter-efficient fine-tuning method for Weather Foundation Models. The novelty in this work is to combine: (i) Task-Adaptive Dynamic Prompting (TADP), which extracts spatial, variable-level, and meteorological patterns from the encoder’s embedding weights and injects them as soft prompts into the backbone during the forward pass, and (iii) Stochastic Fisher-Guided Adaptive Selection (SFAS), which uses Fisher information (plus controlled randomness) to update only the most task-critical parameters, and preserve core pre-trained knowledge. The authors show that, altogether, these components allow WeatherPEFT to achieve performance comparable to (and sometimes better than) full fine-tuning across tasks such as downscaling, ensemble forecast post-processing, and precipitation prediction, while using significantly fewer trainable parameters than existing PEFT methods.

Admittedly, I am not an expert in Weather Foundation Models (WFM), but I believe that, aside from being able to properly judge how this work relates to previous WFM works, this lack of expertise does not affect my ability to judge the main contribution of the paper, i.e. the PEFT part.

**Strengths:**

1. I congratulate the authors on the clarity of this paper. It is extremely well structured and most of the components are very well explained.

2. Climate change exerts pressure through environmental and societal impacts, including more extreme weather events like floods, droughts, and heatwaves, leading to sea-level rise, and causing water scarcity, food insecurity, and habitat destruction. The relevance of this work does not need any justification, and such techniques will be of great aid in the years to come.

3. WeatherPEFT shows consistently strong results across downstream tasks.

4. The authors do a great work in systematically comparing their results against strong PEFT methods and baselines, providing further support of the usefulness of their approach.

**Weaknesses:**

1. WeatherPEFT is a more complex technique that involves the sequential implementation of three adapters, on top of the rest of the TADP process and the SFAS. Thus, the results shown come at a computational cost.

2. Although I am not asking the authors to perform more experiments, evaluating their method on further downstream tasks could be of interest for future work.

**Questions:**

1. Have the authors tried to evaluate the increase of performance relative to the added complexity of their method?

---

> ### Author Response · Authors · 2025-11-21
> **Author Responses**
>
> We sincerely thank the reviewer for the positive assessment of our work, particularly for recognizing the clarity of our presentation and the high societal relevance of efficient Weather Foundation Models. We appreciate your constructive feedback regarding computational complexity and the scope of evaluation. We have made every effort to faithfully address your comments in the following responses.
>
> >Weakness 1 & Question 1 (Complexity): The results of WeatherPEFT come at a computational cost.
>
> We thank the reviewer for this insightful question regarding the trade-off between algorithmic complexity and computational efficiency. We acknowledge that the sequential implementation of the three specialized adapters in TADP and the parameter selection in SFAS introduce a degree of computational overhead compared to simpler techniques. However, this additional cost is marginal and represents a highly favorable trade-off given the substantial performance gains. To quantitatively evaluate this, we measure the total training time for WeatherPEFT against PEFT baselines (LoRA and AdaptFormer) across all three downstream tasks.
>
> All models are trained using the experimental settings detailed in Appendix F.1: 8 NVIDIA A800 GPUs, with specific batch sizes (5, 1, 4) and epoch counts (30, 10, 15) for Downscaling, Post-Processing, and Precipitation Forecasting, respectively.
> |Methods|Downscaling - Training Time|Post-Processing - Training|Precipitation Forecasting - Training Time|
> |-|:-:|:-:|:-:|
> |LoRA|5h09m|1h09m|1h42m|
> |AdaptFormer|5h05m|1hs06m|1h40m|
> |Ours|5h33m|1h20m|1h58m|
>
> As shown in the Table, WeatherPEFT incurs a modest training time increase of approximately 10% compared to LoRA. The marginal increase in wall-clock training time is an acceptable trade-off for enabling Weather Foundation Models (WFMs) to accurately solve complex downstream tasks where generic PEFT methods fail. We will include this training time analysis in the revised appendix to transparently address the complexity trade-off.
>
> >Weakness 2 (Downstream Task): Evaluating the method on further downstream tasks could be of interest for future work.
>
> We appreciate the reviewer’s foresight in suggesting evaluation on further downstream tasks. To test the boundaries of our method’s applicability and substantiate that our method is specifically tailored for these meteorological challenges, we further compare WeatherPEFT with general vision PEFT baselines (including additional Convpass [1], FacT-TT [2], and MoA (Mixture of Adapter)) on a standard computer vision task. Specifically, we evaluate performance on the Cityscapes [3] $\to$ ACDC [4] domain generalization benchmark for semantic segmentation, which encompasses the Night, Snow, Fog, and Rain domains. We utilize Dinov2-L [5] as the backbone for these experiments and report the mean Intersection over Union (mIoU). The results are as follows:
> |Methods|Params (M)|Night|Snow|Fog|Rain|Mean|
> |-|:-:|:-:|:-:|:-:|:-:|:-:|
> |Full-Tuning|304.20|52.4|70.5|80.9|74.4|69.5|
> |Linear-Probing|0.00|54.3|69.3|79.1|68.0|67.6|
> |Convpass|3.64|**56.0**|71.7|80.2|**74.9**|70.7|
> |FacT-TT|2.85|56.1|71.3|81.0|72.9|70.3|
> |MOA|6.39|53.2|70.6|80.3|72.8|69.3|
> |LoRA|3.14|52.3|74.4|79.5|74.0|70.1|
> |AdaptFormer|3.17|53.8|**74.8**|80.3|74.6|**70.9**|
> |VPT|3.15|53.4|74.4|80.4|70.5|69.7|
> |Ours|2.90|**56.0**|70.9|**81.2**|74.5|70.7|
>
> From the results, we observe that WeatherPEFT performs comparably to generic methods like AdaptFormer on standard vision tasks. However, on the weather task (Table 1,2,3), WeatherPEFT significantly outperforms existing PEFT baselines. This validates that WeatherPEFT is not merely another generic adapter. It is a domain-specific design necessary for Weather Foundation Models. We will incorporate these results into the revised manuscript.
>
> [1] Convolutional Bypasses Are Better Vision Transformer Adapters. ECAI. 2024.
>
> [2] Fact: Factor-tuning for lightweight adaptation on vision transformer. AAAI. 2023.
>
> [3] The cityscapes dataset for semantic urban scene understanding. CVPR. 2016.
>
> [4] ACDC: The adverse conditions dataset with correspondences for semantic driving scene understanding. ICCV. 2021.
>
> [5] Dinov2: Learning robust visual features without supervision. TMLR. 2024.
>
> **We sincerely appreciate your insightful feedback and have addressed your concerns to the best of our ability. Should you have any further questions or recommendations, we would be glad to engage in further discussion.**

---

> > ### Comment · Reviewer_maT4 · 2025-11-25
> >
> > I thank the reviewers for their reply, and will maintain my score.

---

> > > ### Author Response · Authors · 2025-11-26
> > >
> > > Thank you for your feedback. We sincerely appreciate your recognition of our work and the constructive suggestions that have improved our paper. We also welcome your participation in further discussions regarding the general comments.

---

> ### Author Response · Authors · 2025-11-25
> **Looking forward to your feedback**
>
> Dear Reviewer maT4,
>
> We are eager to ensure that we have adequately addressed your concerns and are prepared to offer further clarifications or address any additional questions you may have.
>
> We would like to express our heartfelt gratitude for the time and effort you have dedicated to reviewing our work. It has been a pleasure to engage with you throughout this process.
>
> Best regards,
>
> The authors

---

### Official Review · Reviewer_Nt4E · 2025-10-31

**Soundness:** 2
**Presentation:** 2
**Contribution:** 2
**Rating:** 4
**Confidence:** 4

**Summary:**

They aimed to tackle the task-adaptive fine-tuning of the foundation models for weather forecasting. To deal with the domain specific challenges in weather forecasting (eg, task-specific characteristics), they proposed two PEFT ideas, specifically: i) task-adaptive dynamic prompting and ii) stochastic fisher-guided adaptive selection. The proposed PEFT method was tested on three downstream tasks/scenarios: downscaling, ensemble, and regional forecasting.

**Strengths:**

- A novel adapter module called task-adaptive dynamic prompting looks interesting.
- Interesting applications of PEFT of foundation models for the weather forecasting domain such as regional forecasting and ensemble forecasting.

**Weaknesses:**

- Arguments in Line 074-082 on why existing PEFT methods fail on weather task adaptation problems, is not very convincing since they also seem to hold for language and vision tasks (eg, task-wise varying input characteristics, different parameters play varying roles in different tasks). Are there any more convincing arguments on why the weather domain is specific?

- Can the proposed TADP and SFAS be applied to general domains? like vision and language downstream task PEFT? Why is it specifically useful for weather data? Some PoC tests on general language and vision domains can be useful. Eg, the HW-, V-, D-adapters, and the external pattern integration in the proposed TADP appear to be applicable to these domains as well almost straightforwardly without modification.

- MoE (eg, mixture of lora experts) is also a very powerful and topical tool for FM and PEFT. As it's not compared in the Curious to see how it would perform on the benchmarks they used as a baseline.

- SFAS seems to be the technique relatively well known in the PEFT literature. So it may not be appropriate to claim it as one of the main contributions or novelty of the paper.

- The PEFT baselines they tested are a bit old, eg, beyond lora there are many more, eg,
sparse lora, vera, or svd-based low rank selection.

- Also, as sparse PEFT competing baselines, as a comparison to the proposed SFAS, the authors may need to consider various pruning methods, and the second-order sparse selection method like [R2].

[R2] Zihao Fu, Haoran Yang, Anthony Man-Cho So, Wai Lam, Lidong Bing, and Nigel Collier. On the Effectiveness of Parameter-Efficient Fine-Tuning. AAAI, 2023.

- Overall, since the sparse adaptive parameter selection based on Fisher info is relatively well known, the main novelty of the paper seems to be merely adding the TADP adapter module. But it is not very clear if the impact of TADP is so significant for publication in this area. Is it better than some combinations of existing PEFT and sparse adaptive parameter selection techniques that the authors did not explore but are often performed in practice? More empirical exploration may be needed together with better justification of the TADP module.

**Questions:**

See questions in the weakness section.

---

> ### Author Response · Authors · 2025-11-21
> **Author Responses (1/3)**
>
> We sincerely appreciate the reviewer’s rigorous assessment, particularly regarding the distinction between meteorological and general vision tasks, and the request for broader baseline comparisons.  We have addressed each point carefully in the following.
> >Weakness 1 & 2 (Motivation): why existing PEFT methods fail on weather task adaptation problems, is not very convincing. Some PoC tests of the proposed method on general vision domains can be useful.
>
> We appreciate the reviewer’s thoughtful critique regarding the distinction between weather and general tasks, as well as the question about the generalizability of our proposed methods. We agree that task heterogeneity exists in vision and language domains; However, we argue that the nature of this variation in meteorology is fundamentally different, necessitating our specific design. Below, we substantiate this motivation with a more detailed clarification and new empirical evidence on general vision tasks.
>
> ### 1. Distinct Inductive Biases: Why Weather is Unique
> While we follow recent works such as GraphCast [1] and Pangu-Weather [2] in adopting image-like representations, weather tasks differ substantially from standard CV or NLP tasks:
> - **Heterogeneous Physical Variables**: Unlike Vision (where RGB channels are homogeneous and spatially aligned) or NLP (where tokens share a unified embedding space), weather data channels represent distinct physical quantities (e.g., Temperature and Geopotential Height) governed by fluid dynamics equations. Crucially, the correlations between these variables change depending on the task. For instance, the coupling between humidity and temperature is critical for precipitation forecasting (thermodynamics) but less dominant for large-scale wind circulation (geostrophic balance). Existing PEFT methods tend to treat parameters uniformly and fail to explicitly recalibrate these variable-specific physical couplings.
> - **Resolution as Physics, not just Scale:** In Vision, resizing an image preserves the semantic content. In weather, changing resolution (e.g., from $5.625^{\circ}$ to $0.25^{\circ}$) fundamentally alters the governing physics, marking a transition from hydrostatic, large-scale dynamics to non-hydrostatic, convective-scale processes. This requires the model to adapt not just to a new data shape, but to different physical regimes.
> - **Spherical and Multi-Scale Geometry**: Weather data is inherently spherical and multi-dimensional, often requiring simultaneous reasoning across vast spatial scales (e.g., global circulation vs. local convection), which differs from object-centric vision tasks.
>
> ### 2. Empirical Verification: Generalizability vs. Specificity
> To substantiate that our method is specifically tailored for these meteorological challenges, we further compare WeatherPEFT with general vision PEFT baselines on a standard computer vision task. Specifically, we evaluate performance on the Cityscapes [3] $\to$ ACDC [4] domain generalization benchmark for semantic segmentation, which encompasses the Night, Snow, Fog, and Rain domains. We utilize Dinov2-L [5] as the backbone for these experiments and report the mean Intersection over Union (mIoU). Additionally, we present corresponding results for the regional precipitation forecasting task. The results are as follows:
> |Methods|Params (M)|Night|Snow|Fog|Rain|Mean|
> |-|:-:|:-:|:-:|:-:|:-:|:-:|
> |Full-Tuning|304.20|52.4|70.5|80.9|74.4|69.5|
> |Linear-Probing|0.00|54.3|69.3|79.1|68.0|67.6|
> |Convpass|3.64|**56.0**|71.7|80.2|**74.9**|70.7|
> |FacT-TT|2.85|56.1|71.3|81.0|72.9|70.3|
> |MOA|6.39|53.2|70.6|80.3|72.8|69.3|
> |LoRA|3.14|52.3|74.4|79.5|74.0|70.1|
> |AdaptFormer|3.17|53.8|**74.8**|80.3|74.6|**70.9**|
> |VPT|3.15|53.4|74.4|80.4|70.5|69.7|
> |Ours|2.90|**56.0**|70.9|**81.2**|74.5|70.7|
>
> |Method|Params (M)|12h-SEEPS|12h-ACC↑|12h-RMSE|24h-SEEPS|24h-ACC↑|24h-RMSE|36h-SEEPS|36h-ACC↑|36h-RMSE|
> |-|:-:|:-:|:-:|:-:|:-:|:-:|:-:|:-:|:-:|:-:|
> |Linear-Probing|0.00|0.581|0.464|0.266|0.720|0.265|0.293|0.790|0.171|0.303|
> |Convpass|4.92|0.485|0.606|0.237|0.615|0.439|0.269|0.697|0.326|0.285|
> |FacT-TT|2.73|0.525|0.553|0.249|0.662|0.371|0.279|0.747 |0.246|0.294|
> |MoA|8.62|0.515|0.563|0.246|0.665|0.354|0.281|0.749|0.235|0.296|
> |LoRA|3.63|0.495|0.592|0.240|0.634|0.415|0.273|0.723|0.294|0.289|
> |AdaptFormer|4.62|0.499|0.577|0.243|0.643|0.378|0.278|0.731|0.258|0.293|
> |VPT|3.75|0.522|0.550|0.250|0.666|0.356|0.281|0.750|0.235|0.296|
> |**Ours**|3.38|**0.368**|**0.742**|**0.198**|**0.515**|**0.559**|**0.247**|**0.615**|**0.443**|**0.268**|
>
> From the results, we observe that WeatherPEFT performs comparably to generic methods, such as AdaptFormer, on standard vision tasks. However, on the weather task, WeatherPEFT significantly outperforms all PEFT baselines. This validates our motivation: standard PEFTs are insufficient for weather foundation models, necessitating the domain-specific design we propose. We will incorporate these clarifications and results into the revised manuscript.

---

> ### Author Response · Authors · 2025-11-21
> **Author Responses (2/3)**
>
> >Weakness 4 & 7 (Novelty): The Fisher information is known in PEFT, and the novelty of SFAS is limited; It is not very clear if the impact of TADP is so significant.
>
> We sincerely thank the reviewer for their critical assessment of our methodological contributions. We appreciate the opportunity to clarify the positioning of SFAS relative to existing literature and to provide new empirical validation for the impact and design of TADP.
>
> ### 1. Clarification on SFAS
> We agree that parameter selection based on Fisher Information is an established technique in the literature (e.g., [6,7]). While these methods share the high-level intuition of selecting parameter subsets, WeatherPEFT is distinct in the following aspects:
> - **Dynamic Annealed Selection vs. Static/Bernoulli Sampling:** Existing Fisher-related methods rely on either random Bernoulli sampling [6] or Fisher Information but keep the mask static throughout the training process (static) [6,7]. In contrast, we introduce a novel annealed stochastic strategy (Eq. 8) that combines Fisher Information with a linear decay factor. Unlike a static mask, SFAS dynamically updates the parameter subset. Crucially, the stochastic intervention is high at the start to prevent the model from overfitting to initial noise artifacts due to distinct distribution shifts, and progressively decays to allow precise, deterministic Fisher-guided optimization in later stages.
> - **Weather-driven architectural design:** These Fisher-related methods are mainly developed for the unified embedding spaces of LLMs. Our work is motivated by the unique challenges of weather adaptation, where data are highly heterogeneous and physically structured, distinct from their focus on purely reducing fine-tuning cost in NLP. Our approach is not merely a subset selection method, and it is a holistic framework. SFAS does not operate in isolation but is applied to the **meteorology-aware representations** recalibrated by the TADP module. TADP first extracts internal and external patterns from the encoder to condition the backbone. SFAS then acts as a stabilizer during backpropagation to preserve invariant physical priors. This "Prompt-then-Select" synergy is absent in existing methods.
>
> In our revision, we will explicitly acknowledge these foundational works to properly contextualize our contribution.
> ### 2. Justification of TADP
> To validate whether TADP provides significant value over "combinations of existing PEFT and sparse adaptive parameter selection", we conduct additional experiments. We combine the SFAS mechanism with existing generic PEFT methods (LoRA, AdaptFormer, and VPT) and compared them against our full WeatherPEFT framework on the challenging Regional Precipitation Forecasting task.
> |Method|Params (M)|12h-SEEPS|12h-ACC↑|12h-RMSE|24h-SEEPS|24h-ACC↑|24h-RMSE|36h-SEEPS|36h-ACC↑|36h-RMSE|
> |-|:-:|:-:|:-:|:-:|:-:|:-:|:-:|:-:|:-:|:-:|
> |SFAS Only|1.26|0.459|0.634|0.231|0.612|0.443 |0.269|0.716|0.294|0.289|
> |AdaptFormer+SFAS|5.88|0.475|0.608|0.236|0.617|0.419|0.272|0.705|0.302|0.288|
> |LoRA+SFAS|4.89|0.446|0.647|0.227|0.592|0.464|0.265|0.701|0.316|0.286|
> |VPT+SFAS|5.01|0.395|0.708|0.209|0.537|0.533 |0.252|0.639|0.410|0.273|
> |**Ours**|3.38|**0.368**|**0.742**|**0.198**|**0.515**|**0.559**|**0.247**|**0.615**|**0.443**|**0.268**|
>
> Adding sparse selection (SFAS) to generic adapters yields suboptimal results, which may be attributed to their failure to capture the complex variable-specific couplings. By explicitly modeling internal patterns and external patterns, WeatherPEFT outperforms these combinatorial baselines using fewer parameters. The empirical results unequivocally demonstrate the necessity of the TADP module. We will include this comparative analysis in the revised manuscript to firmly justify the architectural contribution of TADP.

---

> > ### Comment · Reviewer_hX2x · 2025-11-27
> >
> > The authors argue that their method differs from existing Fisher-based PEFT by introducing a dynamic, annealed selection strategy instead of static or Bernoulli sampling, and by integrating this with a weather-specific architectural module (TADP). While the dynamic mask update and decay mechanism offer a reasonable extension, **similar ideas have long existed in dynamic sparse training literature, limiting the novelty**. The weather-driven design provides domain-specific motivation, but mainly represents an application-oriented adapter variant rather than a fundamentally new PEFT paradigm. Overall, the claimed distinctions are incremental, and several statements overgeneralize the limitations of existing PEFT methods.
> >
> > I agree with the results on your new experimental table. So I raise my score to 2. Most of my concerns are about the motivation and novelty, so I insist on my decision.

---

> > > ### Author Response · Authors · 2025-11-28
> > > **Author Responses to Reviewer hX2x (1/2)**
> > >
> > > Dear Reviewer hX2x,
> > >
> > > We sincerely appreciate your continued engagement with our work. We notice that this follow-up comment is posted under **another reviewer’s thread** (Reviewer Nt4E). We want to ensure this and the previous reviewer-specific response reaches you directly to prevent any potential misunderstandings regarding our previous rebuttals. Therefore, we are pasting our full response here to ensure you see it immediately. We have also posted this in your original thread.
> > >
> > > We have read your comments carefully. While we appreciate that you have raised your score to 2 based on the new experimental results, we respectfully disagree with the premise that applying domain-specific insights to a specialized field like meteorology constitutes "incremental" work or that the distinction between weather and generic vision tasks is trivial.
> > >
> > > ### 1. Application-Driven Innovation in Physical Sciences
> > >
> > > We wish to gently remind the reviewer that our submission targets the ICLR Primary Area: **applications to physical sciences (physics, chemistry, biology, etc.)**. In this community, the scope of contribution is not solely defined by proposing a **"fundamentally new PEFT paradigm"** (like creating LoRA from scratch), but rather by **how AI techniques can be innovated to solve specific scientific problems**. As noted in recent literature on the philosophy of ML research [1], "Application-driven innovation... is not merely to applications but to advancing ML methods". This kind of application-driven research is essential to pushing the boundaries of AI beyond just methodological advances, providing critical contributions to both AI and domain-specific fields. Leading works such as GraphCast [2], Pangu-Weather [3], and ClimaX [4] have been celebrated not for inventing entirely new architectures from scratch, but for effectively adapting Transformers or GNNs to respect the laws of physics.
> > >
> > > Our work is the **first** to explore the underexplored intersection of PEFT and Weather Foundation Models. We argue that dismissing domain-specific adaptation as merely "applying a widely used method" overlooks the scientific value of tailoring ML to distinct physical regimes. **Diverse academic perspectives drive the field forward**. We hope the value of "**Application-Driven Innovation**"  alongside pure methods-driven research can be recognized.
> > >
> > > [1] Application-driven innovation in machine learning. arXiv. 2024.
> > >
> > > [2] Learning skillful medium-range global weather forecasting. Science. 2023.
> > >
> > > [3] Accurate medium-range global weather forecasting with 3D neural networks. Nature. 2023.
> > >
> > > [4] ClimaX: A foundation model for weather and climate. ICML. 2023.
> > >
> > > ### 2. Clarification on "Uniform Treatment" of Parameters
> > >
> > > Regarding the criticism of our motivation statement: "Existing PEFT methods tend to treat parameters uniformly..." We use the phrase "tend to" in its standard academic sense: to indicate a prevailing tendency in mainstream methods (e.g., standard LoRA, Adapter, SSF) which update modules uniformly across inputs.
> > > According to the **Oxford English Dictionary, "tend to"** means to be likely to do something or to happen in a particular way because this is what **often** or **usually** happens. Our intention is not to imply that task-specific or selective PEFT does not exist. However, as we have further clarified in the Introduction of the revised manuscript:
> > > > "While task-specific selection methods exist in the broader PEFT literature, they primarily focus on reducing fine-tuning costs in general domains through static selection mechanisms ([5], [6], [7]). Consequently, as evidenced by the results of these methods in Table 3 and 13, they fail to dynamically recalibrate for the complex, variable-specific couplings and physical regime shifts that characterize meteorological data, leading to suboptimal performance."
> > >
> > > [5] Sct: A simple baseline for parameter-efficient fine-tuning via salient channels. IJCV. 2024.
> > >
> > > [6] Raise a Child in Large Language Model: Towards Effective and Generalizable Fine-tuning. EMNLP. 2021.
> > >
> > > [7] On the effectiveness of parameter-efficient fine-tuning. AAAI. 2023.

---

> ### Author Response · Authors · 2025-11-21
> **Author Responses (3/3)**
>
> >Weakness 3 & 5 & 6 (Baseline): It's recommended to compare with MoE, more recent LoRA variants, and sparse PEFT competing baselines.
>
> We sincerely appreciate the reviewer raising this important point. To address this, we further provide the comparison with the representative PEFT baselines, i.e., MoA (Mixture of Adapter), HydraLoRA [8], VeRA [9], and SAM [7], on the Regional Precipitation Forecasting task. To ensure a comprehensive evaluation, we also expand the comparison to include SCT [10], Child-Tuning [6], Convpass [11], FacT [12], and RepAdapter[13]. The results are summarized as below:
> |Method|Params (M)|12h-SEEPS|12h-ACC↑|12h-RMSE|24h-SEEPS|24h-ACC↑|24h-RMSE|36h-SEEPS|36h-ACC↑|36h-RMSE|
> |-|:-:|:-:|:-:|:-:|:-:|:-:|:-:|:-:|:-:|:-:|
> |Convpass|4.92|0.485|0.606|0.237|0.615|0.439|0.269|0.697|0.326|0.285|
> |FacT-TT|2.73|0.525|0.553|0.249|0.662|0.371|0.279|0.747 |0.246|0.294|
> |RepAdapter|3.75|0.534|0.532|0.254|0.675|0.34|0.283|0.757|0.222|0.297|
> |SCT|3.94|0.481|0.607|0.237|0.616|0.439|0.269|0.706|0.316|0.286|
> |Child-TuningD|3.39|0.407|0.694|0.214|0.565|0.500|0.259|0.672|0.364|0.280|
> |MoA|8.62|0.515|0.563|0.246|0.665|0.354|0.281|0.749|0.235|0.296|
> |HydraLoRA|5.77|0.510|0.571|0.245|0.65|0.393|0.276|0.734|0.268|0.292|
> |VeRA|0.98|0.524|0.551|0.250|0.663|0.365|0.280|0.744|0.256|0.293|
> |SAM|3.39|0.421|0.673|0.220|0.598|0.457|0.267|0.704|0.299|0.289|
> |**Ours**|3.38|**0.368**|**0.742**|**0.198**|**0.515**|**0.559**|**0.247**|**0.615**|**0.443**|**0.268**|
>
> It can be observed that methods such as ConvPass, while efficient in standard vision tasks, exhibit performance gaps when applied to meteorological data. It is notable that task-adaptive PEFT methods (SCT, SAM, and Child-TuningD) consistently outperform other baselines. Despite these improvements, WeatherPEFT achieves superior performance across all metrics. This confirms that adaptivity alone is insufficient and coupling it with domain-specific awareness further improves performance. We will incorporate these comprehensive comparisons into the revised manuscript to strengthen our evaluation.
>
> [1] Learning skillful medium-range global weather forecasting. Science. 2023.
>
> [2] Accurate medium-range global weather forecasting with 3D neural networks. Nature. 2023.
>
> [3] The cityscapes dataset for semantic urban scene understanding. CVPR. 2016.
>
> [4] ACDC: The adverse conditions dataset with correspondences for semantic driving scene understanding. ICCV. 2021.
>
> [5] Dinov2: Learning robust visual features without supervision. TMLR. 2024.
>
> [6] Raise a Child in Large Language Model: Towards Effective and Generalizable Fine-tuning. EMNLP. 2021.
>
> [7] On the effectiveness of parameter-efficient fine-tuning. AAAI. 2023.
>
> [8] Hydralora: An asymmetric lora architecture for efficient fine-tuning. NeurIPS. 2024.
>
> [9] VeRA: Vector-based Random Matrix Adaptation. ICLR. 2024.
>
> [10] Sct: A simple baseline for parameter-efficient fine-tuning via salient channels. IJCV. 2024.
>
> [11] Convolutional Bypasses Are Better Vision Transformer Adapters. ECAI. 2024.
>
> [12] Fact: Factor-tuning for lightweight adaptation on vision transformer. AAAI. 2023.
>
> [13] Towards efficient visual adaption via structural re-parameterization. arXiv. 2023.
>
> **We sincerely appreciate your insightful feedback and have addressed your concerns to the best of our ability. Should you have any further questions or recommendations, we would be glad to engage in further discussion.**

---

> ### Author Response · Authors · 2025-11-25
> **Looking forward to your feedback**
>
> Dear Reviewer Nt4E,
>
> We are eager to ensure that we have adequately addressed your concerns and are prepared to offer further clarifications or address any additional questions you may have.
>
> Should you find that our revisions have satisfactorily addressed your main concerns, **we would be most grateful if you would reconsider the evaluation of our paper to enhance its standing**.
>
> We would like to express our heartfelt gratitude for the time and effort you have dedicated to reviewing our work. It has been a pleasure to engage with you throughout this process.
>
> Best regards,
>
> The authors

---

> ### Author Response · Authors · 2025-11-27
> **A summary of our revision**
>
> Dear Reviewer Nt4E,
>
> Thanks for your valuable comments. As the discussion period opens on November 20$^{th}$, we would be grateful if you could allocate some time to review our revision.
>
> We understand that you have a multitude of responsibilities. To facilitate a swift evaluation of our revisions, we have summarized the corresponding changes as follows:
>
> - **Motivation:** We supplement the **Section 1 & 2** to explicitly contrast weather tasks with general tasks and distinguish WeatherPEFT from existing PEFTs. Furthermore, we add a vision domain generalization experiment in Table 4 of **Section 5.4** and results on the Visual Task Adaptation Benchmark (VTAB-1k) in Table 13 of **Appendix B.7** to validate the specificity of our performance gains to the weather domain.
> - **Novelty:** We add a comparison of WeatherPEFT against combinations of Generic PEFTs + SFAS in Table 11 of **Appendix B.5** to validate the specific architectural value of TADP.
> - **Baseline:** We expand Table 3 in **Section 5.3** to include a wider suite of recommended PEFT baselines.
>
> Please let us know if you have any additional concerns or questions. We kindly request your feedback on whether our responses have satisfactorily addressed your concerns. Thank you for your time and consideration!
>
> Best regards,
>
> The authors

---

> ### Comment · Reviewer_hX2x · 2025-11-27
>
> Thank you for your reply. Regarding the motivation, the distinction between WeatherPEFT and generic PEFT remains minor. As the authors state, “Existing PEFT methods tend to treat parameters uniformly and fail to explicitly recalibrate these variable-specific physical couplings.” **This is misleading to readers. I have already pointed out that task-specific parameter selection has been widely explored in generic PEFT methods.** Therefore, it is neither correct nor rigorous to claim that “existing PEFT methods treat parameters uniformly and fail to recalibrate variable-specific couplings.” Based on this clarification, I can understand and accept your response.
>
> Regarding your additional experiments, I fully agree that applying the designed WeatherPEFT method to weather-related tasks achieves higher performance. However, my point is that your claim—“Existing PEFT methods tend to treat parameters uniformly and fail to recalibrate variable-specific couplings”—is inaccurate. Not all PEFT methods treat parameters uniformly, and many works from 2022 to the present have proposed dynamically selecting parameter subsets for downstream tasks. **I am not judging the results; I am just saying that your work seems to apply a widely used PEFT to a specific domain.** I don't buy into such a method to do the research.
>
> I thank you for your listed three items. Have you realized that generic vision tasks also have their distinct characteristics? When you refer to the VTAB benchmark, you will see that the tasks are different. **So you mentioned the differences in the weather domain also exist in the generic domain**. Moreover, the reason for proposing a PEFT approach is to bridge the gap between the pretrained model and downstream tasks. So it is natural to see the distinct differences between downstream tasks and pretrained models. In summary, you claim the unique parts of the Weather task can also be claimed on any other vision downstream tasks, as each vision task is different in many patterns.
>
> Regarding the results analysis, you will also realize that you cannot summarize the unique insights or differences of the proposed method rather just compare the larger or smaller on numbers.

---

> ### Author Response · Authors · 2025-11-28
> **Author Responses to Reviewer hX2x (2/2)**
>
> ### 3. Unique Challenges in Weather Tasks
> Regarding your comment on the uniqueness of weather tasks, we fully agree that different downstream tasks, whether in vision or other domains, have their unique characteristics. However, we respectfully but firmly disagree with the assertion that the uniqueness of weather tasks is equivalent to the variations found between generic vision tasks (e.g., in VTAB-1k). Weather data possesses unique physical heterogeneity that generic vision tasks do not, which we have emphazied in the previous response (**Heterogeneous Physical Variables, Resolution is Physics, not just Scale, and Spherical and Multi-Scale Geometry**).
>
> Our work does not intend to redefine the entire PEFT landscape but to tackle the challenges presented by weather data through a novel adaptation of existing methods. This is not merely a replication of generic PEFT methods but a domain-specific innovation that builds on the foundation of those methods.
>
> ### 4. Clarification on Novelty of WeatherPEFT
>
> Finally, regarding novelty and the comparison to methods like Child-Tuning. WeatherPEFT is not just "sparse training". It is a holistic framework where TADP (Task-Adaptive Dynamic Prompting) adapts the model’s forward pass to task-specific characteristics, and SFAS (Stochastic Fisher-Guided Adaptive Selection) dynamically governs the subsequent parameter updates recalibrated by meteorology-aware adaptation during backpropagation.
>
> The core design of our TADP is to extract task-specific characteristics (such as variables and resolutions) from the encoder's embedding layer to introduce context-aware recalibration. In weather tasks, the embedding layer is rich with varying physical information, allowing TADP to dynamically adapt the model to the specific "physics" of the input. In the standard vision task like VTAB-1k, the embedding layer of the backbone is typically frozen and processes homogeneous RGB data. In this frozen setting, TADP can only extract information from a static layer. Consequently, the "dynamic prompt" effectively becomes a static constant. This neutralizes the primary advantage of TADP's adaptivity, resulting in performance that is comparable to, but not significantly better than, other baselines in this specific context. This proves our method is not a generic wrapper, but a specialized solution for physical sciences.
>
> Unlike Child-Tuning’s static or Bernoulli sampling, SFAS uses an annealed stochastic, dynamic strategy that combines Fisher Information with a linear decay factor. As the model representations are continuously recalibrated by meteorology-aware adaptation (TADP) during the forward pass, SFAS ensures the backward pass dynamically identifies and updates the optimal parameter subsets aligned with these recalibrated states. This "Prompt-then-Select" feedback loop is absent in static or decoupled selection methods.
> Furthermore, the annealed stochastic intervention is explicitly designed for the specific distributional shifts inherent in weather tasks. We introduce high stochasticity at the start to prevent the model from overfitting to initial noise artifacts caused by these distinct distribution shifts. This stochasticity progressively decays, allowing the optimization to transition smoothly into a precise, deterministic Fisher-guided phase once the physical regime is stabilized.
>
> **We hope this clarifies that our contribution is a targeted, scientifically motivated innovation for the physical sciences, rather than a generic increment on existing PEFT literature**.
>
> Best regards,
>
> The authors

---

> ### Author Response · Authors · 2025-11-28
> **A Note to Reviewer Nt4E**
>
> Dear Reviewer Nt4E,
>
> We sincerely apologize for the additional notifications in your thread. Since another reviewer (Reviewer hX2x) posts a follow-up comment here, we are replying above to ensure our response is received.
>
> We respectfully invite you to re-evaluate our submission based on the comprehensive new experiments and clarifications we have provided throughout the rebuttal. We trust your independent judgment regarding the scientific value of application-driven research in the physical sciences, and we hope our response above further clarifies the distinct contributions of our work compared to generic approaches.
>
> Best regards,
>
> The authors

---

### Official Review · Reviewer_hX2x · 2025-11-01

**Soundness:** 1
**Presentation:** 2
**Contribution:** 1
**Rating:** 0
**Confidence:** 5

**Summary:**

WeatherPEFT introduces a novel Parameter-Efficient Fine-Tuning framework for Weather Foundation Models. It combines Task-Adaptive Dynamic Prompting for context-aware feature recalibration and Stochastic Fisher-Guided Adaptive Selection for stable parameter optimization, achieving efficient fine-tuning and performance parity with full-tuning using fewer trainable parameters.

**Strengths:**

- The paper is well-written as origanizing the methods and experiments clear.

- This paper compares many methods, demonstrating that the experiments are extensive and comprehensive.

- The appendix is clear and solid.

**Weaknesses:**

- The description about related works is actually incorrect from Line 078-082. First, there are a lot of task-specific PEFT methods introduced to dynamically select task-specific features to do the adaptation (SCT IJCV 2023) or implicitly selecting features (AdaptFormer, Adapter). Second, most of PEFT methods trained different sets of parameters for different downstream tasks, hence the "apply the same set of trainable parameters across different downstream tasks" is wrong. Then, it still unknow what is the key difference of conducting weather like downstream tasks with the normal vision tranfer tasks in VTAB.

- Based on the above factual error, it indicates (1) the motivation in this paper is incorrect; (2) the literature review of this submission is not adequate and fully understood.

- The SFAS module is similar to the paper "Raise a Child in a Large Language Model: Towards Effective and Generalizable Fine-tuning". The novelty is limited. Please be careful give the comparison.

- The selected PEFT baselines are old, ConvPass, FacT, NOAH, SCT, RepAdapter are not included.

- It is confusing why w/o Internal and w/o External shows minor improvements as compared with WeatherPEFT in Table 8.

**Questions:**

None

---

> ### Author Response · Authors · 2025-11-21
> **Author Responses  (1/3)**
>
> We sincerely appreciate the reviewer’s constructive criticism. We have made every effort to address your comments below.
> > Weakness 1 & 2 (Motivation): The description is incorrect from 078-082. The difference between weather and vision is unknown; The motivation is incorrect and literature review is inadequate.
>
> Thanks for the feedback regarding the clarity and validity of our motivation. Our work is indeed driven by the unique challenges of applying PEFT to weather tasks. Below, we further clarify and substantiate our motivation:
> ### On Task-Specific PEFT
> We apologize for any confusion caused by the wording of our original claim. Our statement that "***most** existing PEFT approaches **tend to** apply the same set of trainable parameters across different downstream tasks*" is intended to highlight a **prevailing tendency** in many mainstream PEFT methods like LoRA, which uniformly updates the entire PEFT module across all inputs rather than dynamically tuning task-specific parameters. This tendency may limit adaptation effectiveness in highly heterogeneous weather tasks. Our intention is **not** to imply that task-specific or selective PEFT does not exist.
>
> We sincerely appreciate the references to SCT [1] and other methods. We will revise our Related Work and Intro section to properly cite relevant literature, clarifying that our motivation is not simply to add "selection", but to introduce context-aware recalibration (TADP) and stabilized selection (SFAS) that explicitly handle the physical shifts in weather tasks.
> ### On Difference between Weather and Vision Tasks
> While we follow recent weather works [2,3] in adopting image-like representations, the underlying meteorological semantics remain substantially different:
> - **Heterogeneous Physical Variables**: Unlike the homogeneous RGB channels of natural images, weather data channels represent distinct physical quantities (e.g., Temperature and Humidity) governed by fluid dynamics equations. Crucially, the correlations between these variables change depending on the task. For instance, the coupling between humidity and temperature is critical for precipitation forecasting (thermodynamics) but less dominant for large-scale wind circulation (geostrophic balance). Existing PEFT methods fail to explicitly recalibrate these variable-specific physical couplings.
> - **Resolution as Physics:** In Vision, resizing an image preserves the semantic content. In weather, changing resolution (e.g., from 5.625° to 0.25°) fundamentally alters the governing physics, transmitting from hydrostatic, large-scale dynamics to non-hydrostatic, convective-scale processes. This requires the model to adapt not just to a new data shape, but to different physical regimes.
> - **Spherical and Multi-Scale Geometry**: Weather data is inherently spherical and multi-dimensional, often requiring simultaneous reasoning across vast spatial scales (e.g., global circulation vs. local convection), which differs from object-centric vision tasks.
> ### Empirical Verification
> To substantiate that our method is specifically tailored for the meteorological challenge, we further compare WeatherPEFT with general vision PEFT baselines on a vision task, i.e, Cityscapes [4] → ACDC [5] domain generalization benchmarks for semantic segmentation, using Dinov2-L [6] and evaluating via mIoU. We also provide corresponding results on precipitation forecasting task.
> |Methods|Params (M)|Night|Snow|Fog|Rain|Mean|
> |-|:-:|:-:|-|:-:|:-:|-|
> |Full-Tuning|304.20|52.4|70.5|80.9|74.4|69.5|
> |Linear-Probing|0|54.3|69.3|79.1|68|67.6|
> |Convpass|3.64|**56**|71.7|80.2|**74.9**|70.7|
> |FacT-TT|2.85|56.1|71.3|81|72.9|70.3|
> |MOA|6.39|53.2|70.6|80.3|72.8|69.3|
> |LoRA|3.14|52.3|74.4|79.5|74|70.1|
> |AdaptFormer|3.17|53.8|**74.8**|80.3|74.6|**70.9**|
> |VPT|3.15|53.4|74.4|80.4|70.5|69.7|
> |Ours|2.90|**56**|70.9|**81.2**|74.5|70.7|
>
> |Methods|Params (M)|12h-SEEPS|12h-ACC|12h-RMSE|24h-S.|24h-A.|24h-R.|36h-S.|36h-A.|36h-R.|
> |-|-|-|:-:|:-:|:-:|-|-|:-:|-|-|
> |Linear-Probing|0|0.581|0.464|0.266|0.72|0.265|0.293|0.79|0.171|0.303|
> |Convpass|4.92|0.485|0.606|0.237|0.615|0.439|0.269|0.697|0.326|0.285|
> |FacT-TT|2.73|0.525|0.553|0.249|0.662|0.371|0.279|0.747 |0.246|0.294|
> |MoA|8.62|0.515|0.563|0.246|0.665|0.354|0.281|0.749|0.235|0.296|
> |LoRA|3.63|0.495|0.592|0.24|0.634|0.415|0.273|0.723|0.294|0.289|
> |AdaptFormer|4.62|0.499|0.577|0.243|0.643|0.378|0.278|0.731|0.258|0.293|
> |VPT|3.75|0.522|0.55|0.25|0.666|0.356|0.281|0.75|0.235|0.296|
> |**Ours**|3.38|**0.368**|**0.742**|**0.198**|**0.515**|**0.559**|**0.247**|**0.615**|**0.443**|**0.268**|
>
> Results show that WeatherPEFT performs comparably to generic methods like AdaptFormer on standard vision tasks. However, on the weather task, WeatherPEFT significantly outperforms all PEFT baselines. This validates our motivation: standard PEFTs are insufficient for weather foundation models, necessitating the domain-specific design we propose. We will incorporate these clarifications and results into the revised manuscript.

---

> > ### Author Response · Authors · 2025-11-21
> > **Author Responses (2/3)**
> >
> > > Weakness 3 (Novelty): SFAS module is similar to Child-Tuning. The novelty is limited.
> >
> > We thank the reviewer for this thoughtful comment and fully acknowledge the contribution of Child-Tuning [7]. However, while both methods share the high-level intuition of selecting parameter subsets, WeatherPEFT is distinct in following aspects:
> > 1. **Dynamic Annealed Selection vs. Static/Bernoulli Sampling:** Child-Tuning relies on either random Bernoulli sampling (Child-TuningF) or Fisher Information but keeps the mask static throughout the training process (Child-TuningD). In contrast, we introduce a novel annealed stochastic strategy (Eq. 8) that combines Fisher Information with a linear decay factor. Unlike Child-TuningD's static mask, SFAS dynamically updates the parameter subset. Crucially, the stochastic intervention is high at the start to prevent the model from overfitting to initial noise artifacts due to distinct distribution shifts, and progressively decays to allow precise, deterministic Fisher-guided optimization in later stages.
> > 2. **Weather-driven architectural design:** Child-Tuning is developed for the unified embedding spaces of LLMs. Our work is motivated by the unique challenges of weather adaptation, where data are highly heterogeneous and physically structured, distinct from Child-Tuning’s focus on purely reducing fine-tuning cost in NLP. Our approach is not merely a subset selection method and it is a holistic framework. SFAS does not operate in isolation but is applied to the **meteorology-aware representations** recalibrated by the TADP module. TADP first extracts internal and external patterns from the encoder to condition the backbone. SFAS then acts as a stabilizer during backpropagation to preserve invariant physical priors. This "Prompt-then-Select" synergy is absent in Child-Tuning.
> > 3. **Empirical Superiority in Weather Tasks:** To rigorously demonstrate the impact of these architectural differences, we compare Child-TuningD against our method on the Precipitation Forecasting task with a similar parameter budget.
> >
> > |Method|Params (M)|12h-SEEPS|12h-ACC|12h-RMSE|24h-SEEPS|24h-ACC|24h-RMSE|36h-SEEPS|36h-ACC|36h-RMSE|
> > |-|:-:|:-:|:-:|:-:|:-:|:-:|:-:|:-:|:-:|:-:|
> > |SFAS Only|1.26|0.459|0.634|0.231|0.612|0.443|0.269|0.716|0.294|0.289|
> > |Child-TuningD|3.39|0.407|0.694|0.214|0.565|0.500|0.259|0.672|0.364|0.280|
> > |**Ours**|3.38|**0.368**|**0.742**|**0.198**|**0.515**|**0.559**|**0.247**|**0.615**|**0.443**|**0.268**|
> >
> > As shown above, while Child-TuningD improves over the "SFAS Only" baseline (likely due to the higher parameter count), WeatherPEFT significantly outperforms Child-Tuning-F across all metrics. This confirms that the combination of TADP's context-awareness and SFAS's annealed dynamic selection is essential for effective adaptation in the weather domain. We will incorporate this discussion and the comparative results into the revised manuscript to clarify the novelty and domain-specific advantages of our method.
> > > Weakness 4 (Baselines): The selected PEFT baselines are old.
> >
> > We sincerely appreciate the reviewer raising this important point. To address this, we further provide the comparison between WeatherPEFT and the recommended PEFT baselines (SCT [1], Child-Tuning [7], Convpass [8], FacT [9], and RepAdapter[10]) on the Regional Precipitation Forecasting task. To ensure a comprehensive evaluation, we also expand the comparison to include MoA (Mixture of Adapter), HydraLoRA [11], VeRA [12], and SAM [13]. The results are summarized as below:
> > |Method|Params (M)|12h-SEEPS|12h-ACC↑|12h-RMSE|24h-SEEPS|24h-ACC↑|24h-RMSE|36h-SEEPS|36h-ACC↑|36h-RMSE|
> > |-|:-:|:-:|:-:|:-:|:-:|:-:|:-:|:-:|:-:|:-:|
> > |Convpass|4.92|0.485|0.606|0.237|0.615|0.439|0.269|0.697|0.326|0.285|
> > |FacT-TT|2.73|0.525|0.553|0.249|0.662|0.371|0.279|0.747 |0.246|0.294|
> > |RepAdapter|3.75|0.534|0.532|0.254|0.675|0.34|0.283|0.757|0.222|0.297|
> > |SCT|3.94|0.481|0.607|0.237|0.616|0.439|0.269|0.706|0.316|0.286|
> > |Child-TuningD|3.39|0.407|0.694|0.214|0.565|0.500|0.259|0.672|0.364|0.280|
> > |MoA|8.62|0.515|0.563|0.246|0.665|0.354|0.281|0.749|0.235|0.296|
> > |HydraLoRA|5.77|0.510|0.571|0.245|0.65|0.393|0.276|0.734|0.268|0.292|
> > |VeRA|0.98|0.524|0.551|0.250|0.663|0.365|0.280|0.744|0.256|0.293|
> > |SAM|3.39|0.421|0.673|0.220|0.598|0.457|0.267|0.704|0.299|0.289|
> > |**Ours**|3.38|**0.368**|**0.742**|**0.198**|**0.515**|**0.559**|**0.247**|**0.615**|**0.443**|**0.268**|
> >
> > It can be observed that methods such as ConvPass, while efficient in standard vision tasks, they exhibit performance gaps when applied to meteorological data. It is notable that task-adaptive PEFT methods (SCT, SAM, and Child-TuningD) consistently outperform other baselines. Despite these improvements, WeatherPEFT achieves superior performance across all metrics. This confirms that adaptivity alone is insufficient and coupling it with domain-specific awareness further improves performance. We will incorporate these comprehensive comparisons into the revised manuscript to strengthen our evaluation.

---

> ### Author Response · Authors · 2025-11-21
> **Author Responses (3/3)**
>
> > Weakness 5 (Results): Confused about the minor improvements of ablating Internal & External Extraction.
>
> Thank you for raising this insightful question. We would like to clarify that performance differences tend to remain small under a low parameter budget, as illustrated in Table 1. This may be particularly pertinent to the Downscaling task, where the optimization landscape is highly constrained by the minimal trainable parameters, leading to a "performance saturation" that compresses the variance between different methods. For clarity, the key results extracted from Table 1 that support this observation are provided below:
>
> |Method|T2m-RMSE|T2m-Bias|U10-RMSE|U10-Bias|V10-RMSE|V10-Bias|T850-RMSE|T850-Bias|Z500-RMSE|Z500-Bias|
> |-|:-:|:-:|:-:|:-:|:-:|:-:|:-:|:-:|:-:|:-:|
> |LoRA|1.190|0.006|1.130|0.000|1.118|-0.002|0.998|-0.001|50.061|0.984|
> |SFAS Only|0.161|0.010|1.090|0.001|1.081|0.002|0.973|0.002|47.000|0.848|
> |Ours|1.119|0.003|1.057|0.000|1.051|-0.001|0.950|0.004|44.922|0.413|
>
> Building on this observation, we further conducted an ablation study under the larger-parameter setting to verify the stability of this effect. The results are summarized below:
>
> |Method|T2m-RMSE|T2m-Bias|U10-RMSE|U10-Bias|V10-RMSE|V10-Bias|T850-RMSE|T850-Bias|Z500-RMSE|Z500-Bias|
> |-|:-:|:-:|:-:|:-:|:-:|:-:|:-:|:-:|:-:|:-:|
> |w/o Internal|0.970|0.000|0.913|-0.002|0.912 |-0.002|0.860|-0.002|36.870|0.611|
> |w/o External|0.958|0.000|0.903|-0.001|0.903|-0.003|0.854|-0.001|36.415|0.640|
> |w/o Randomness|0.954|0.000|0.900|-0.002|0.901|-0.002|0.852|-0.001|36.277|0.620|
> |Ours|0.916|0.000|0.873|-0.001|0.875|-0.002|0.834|-0.002|35.076|0.504|
>
> The results of larger-parameter setting demonstrate that the performance gaps become significantly more pronounced as capacity increases. These findings collectively demonstrate: while the gains appear compressed at the extreme low-parameter limit, the Internal and External pattern extraction mechanisms are essential for robust scaling. They allow the model to efficiently utilize additional capacity to capture complex meteorological dynamics, preventing the premature plateauing observed in the ablated variants. We will include this expanded analysis in the revised appendix to clarify the contribution of each module.
>
> [1] Sct: A simple baseline for parameter-efficient fine-tuning via salient channels. IJCV. 2024.
>
> [2] Learning skillful medium-range global weather forecasting. Science. 2023.
>
> [3] Accurate medium-range global weather forecasting with 3D neural networks. Nature. 2023.
>
> [4] The cityscapes dataset for semantic urban scene understanding. CVPR. 2016.
>
> [5] ACDC: The adverse conditions dataset with correspondences for semantic driving scene understanding. ICCV. 2021.
>
> [6] Dinov2: Learning robust visual features without supervision. TMLR. 2024.
>
> [7] Raise a Child in Large Language Model: Towards Effective and Generalizable Fine-tuning. EMNLP. 2021.
>
> [8] Convolutional Bypasses Are Better Vision Transformer Adapters. ECAI. 2024.
>
> [9] Fact: Factor-tuning for lightweight adaptation on vision transformer. AAAI. 2023.
>
> [10] Towards efficient visual adaption via structural re-parameterization. arXiv. 2023.
>
> [11] Hydralora: An asymmetric lora architecture for efficient fine-tuning. NeurIPS. 2024.
>
> [12] VeRA: Vector-based Random Matrix Adaptation. ICLR. 2024.
>
> [13] On the effectiveness of parameter-efficient fine-tuning. AAAI. 2023.
>
> **We sincerely appreciate your insightful feedback and have addressed your concerns to the best of our ability. Should you have any further questions or recommendations, we would be glad to engage in further discussion.**

---

> ### Author Response · Authors · 2025-11-25
> **Looking forward to your feedback**
>
> Dear Reviewer hX2x,
>
> We are eager to ensure that we have adequately addressed your concerns and are prepared to offer further clarifications or address any additional questions you may have.
>
> Should you find that our revisions have satisfactorily addressed your main concerns, **we would be most grateful if you would reconsider the evaluation of our paper to enhance its standing**.
>
> We would like to express our heartfelt gratitude for the time and effort you have dedicated to reviewing our work. It has been a pleasure to engage with you throughout this process.
>
> Best regards,
>
> The authors

---

> ### Author Response · Authors · 2025-11-27
> **A summary of our revision**
>
> Dear Reviewer hX2x,
>
> Thanks for your valuable comments. As the discussion period opens on November 20$^{th}$, we would be grateful if you could allocate some time to review our revision.
>
> We understand that you have a multitude of responsibilities. To facilitate a swift evaluation of our revisions, we have summarized the corresponding changes as follows:
>
> - **Motivation & Novelty:** We supplement the **Section 1 & 2** to explicitly contrast weather tasks with general tasks and distinguish WeatherPEFT from existing PEFTs. Furthermore, we add a vision domain generalization experiment in Table 4 of **Section 5.4** and results on the Visual Task Adaptation Benchmark (VTAB-1k) in Table 13 of **Appendix B.7** to validate the specificity of our performance gains to the weather domain.
> - **Baseline:** We expand Table 3 in **Section 5.3** to include a wider suite of recommended PEFT baselines.
> - **Results:** We extend the fine-grained ablation to include larger-parameter settings in Table 9 of **Appendix B.3** to show the scalability of improvements.
>
> Please let us know if you have any additional concerns or questions. We kindly request your feedback on whether our responses have satisfactorily addressed your concerns. Thank you for your time and consideration!
>
> Best regards,
>
> The authors

---

> ### Author Response · Authors · 2025-11-28
> **Author Responses to Further Comments (1/2)**
>
> Dear Reviewer hX2x,
>
> We sincerely appreciate your continued engagement with our work. We notice that the follow-up comment is posted under **another reviewer’s thread** (Reviewer Nt4E). We want to ensure this and the previous reviewer-specific response reaches you directly to prevent any potential misunderstandings regarding our previous rebuttals. Therefore, we are posting this in your original thread, and we have also pasted our full response in Reviewer Nt4E's thread to ensure you see it immediately.
>
> We have read your comments carefully. While we appreciate that you have raised your score to 2 based on the new experimental results, we respectfully disagree with the premise that applying domain-specific insights to a specialized field like meteorology constitutes "incremental" work or that the distinction between weather and generic vision tasks is trivial.
>
> ### 1. Application-Driven Innovation in Physical Sciences
>
> We wish to gently remind the reviewer that our submission targets the ICLR Primary Area: **applications to physical sciences (physics, chemistry, biology, etc.)**. In this community, the scope of contribution is not solely defined by proposing a **"fundamentally new PEFT paradigm"** (like creating LoRA from scratch), but rather by **how AI techniques can be innovated to solve specific scientific problems**. As noted in recent literature on the philosophy of ML research [1], "Application-driven innovation... is not merely to applications but to advancing ML methods". This kind of application-driven research is essential to pushing the boundaries of AI beyond just methodological advances, providing critical contributions to both AI and domain-specific fields. Leading works such as GraphCast [2], Pangu-Weather [3], and ClimaX [4] have been celebrated not for inventing entirely new architectures from scratch, but for effectively adapting Transformers or GNNs to respect the laws of physics.
>
> Our work is the **first** to explore the underexplored intersection of PEFT and Weather Foundation Models. We argue that dismissing domain-specific adaptation as merely "applying a widely used method" overlooks the scientific value of tailoring ML to distinct physical regimes. **Diverse academic perspectives drive the field forward**. We hope the value of "**Application-Driven Innovation**"  alongside pure methods-driven research can be recognized.
>
> [1] Application-driven innovation in machine learning. arXiv. 2024.
>
> [2] Learning skillful medium-range global weather forecasting. Science. 2023.
>
> [3] Accurate medium-range global weather forecasting with 3D neural networks. Nature. 2023.
>
> [4] ClimaX: A foundation model for weather and climate. ICML. 2023.
>
> ### 2. Clarification on "Uniform Treatment" of Parameters
>
> Regarding the criticism of our motivation statement: "Existing PEFT methods tend to treat parameters uniformly..." We use the phrase "tend to" in its standard academic sense: to indicate a prevailing tendency in mainstream methods (e.g., standard LoRA, Adapter, SSF) which update modules uniformly across inputs.
> According to the **Oxford English Dictionary, "tend to"** means to be likely to do something or to happen in a particular way because this is what **often** or **usually** happens. Our intention is not to imply that task-specific or selective PEFT does not exist. However, as we have further clarified in the Introduction of the revised manuscript:
> > "While task-specific selection methods exist in the broader PEFT literature, they primarily focus on reducing fine-tuning costs in general domains through static selection mechanisms ([5], [6], [7]). Consequently, as evidenced by the results of these methods in Table 3 and 13, they fail to dynamically recalibrate for the complex, variable-specific couplings and physical regime shifts that characterize meteorological data, leading to suboptimal performance."
>
> [5] Sct: A simple baseline for parameter-efficient fine-tuning via salient channels. IJCV. 2024.
>
> [6] Raise a Child in Large Language Model: Towards Effective and Generalizable Fine-tuning. EMNLP. 2021.
>
> [7] On the effectiveness of parameter-efficient fine-tuning. AAAI. 2023.

---

> ### Author Response · Authors · 2025-11-28
> **Author Responses to Further Comments (2/2)**
>
> ### 3. Unique Challenges in Weather Tasks
> Regarding your comment on the uniqueness of weather tasks, we fully agree that different downstream tasks, whether in vision or other domains, have their unique characteristics. However, we respectfully but firmly disagree with the assertion that the uniqueness of weather tasks is equivalent to the variations found between generic vision tasks (e.g., in VTAB-1k). Weather data possesses unique physical heterogeneity that generic vision tasks do not, which we have emphazied in the previous response (**Heterogeneous Physical Variables, Resolution is Physics, not just Scale, and Spherical and Multi-Scale Geometry**).
>
> Our work does not intend to redefine the entire PEFT landscape but to tackle the challenges presented by weather data through a novel adaptation of existing methods. This is not merely a replication of generic PEFT methods but a domain-specific innovation that builds on the foundation of those methods.
>
> ### 4. Clarification on Novelty of WeatherPEFT
>
> Finally, regarding novelty and the comparison to methods like Child-Tuning. WeatherPEFT is not just "sparse training". It is a holistic framework where TADP (Task-Adaptive Dynamic Prompting) adapts the model’s forward pass to task-specific characteristics, and SFAS (Stochastic Fisher-Guided Adaptive Selection) dynamically governs the subsequent parameter updates recalibrated by meteorology-aware adaptation during backpropagation.
>
> The core design of our TADP is to extract task-specific characteristics (such as variables and resolutions) from the encoder's embedding layer to introduce context-aware recalibration. In weather tasks, the embedding layer is rich with varying physical information, allowing TADP to dynamically adapt the model to the specific "physics" of the input. In the standard vision task like VTAB-1k, the embedding layer of the backbone is typically frozen and processes homogeneous RGB data. In this frozen setting, TADP can only extract information from a static layer. Consequently, the "dynamic prompt" effectively becomes a static constant. This neutralizes the primary advantage of TADP's adaptivity, resulting in performance that is comparable to, but not significantly better than, other baselines in this specific context. This proves our method is not a generic wrapper, but a specialized solution for physical sciences.
>
> Unlike Child-Tuning’s static or Bernoulli sampling, SFAS uses an annealed stochastic, dynamic strategy that combines Fisher Information with a linear decay factor. As the model representations are continuously recalibrated by meteorology-aware adaptation (TADP) during the forward pass, SFAS ensures the backward pass dynamically identifies and updates the optimal parameter subsets aligned with these recalibrated states. This "Prompt-then-Select" feedback loop is absent in static or decoupled selection methods.
> Furthermore, the annealed stochastic intervention is explicitly designed for the specific distributional shifts inherent in weather tasks. We introduce high stochasticity at the start to prevent the model from overfitting to initial noise artifacts caused by these distinct distribution shifts. This stochasticity progressively decays, allowing the optimization to transition smoothly into a precise, deterministic Fisher-guided phase once the physical regime is stabilized.
>
> **We hope this clarifies that our contribution is a targeted, scientifically motivated innovation for the physical sciences, rather than a generic increment on existing PEFT literature**.
>
> Best regards,
>
> The authors

---

### Author Response · Authors · 2025-11-23
**General Response to All Reviewers**

Dear Reviewers,

We sincerely thank all reviewers (`hX2x`, `Nt4E`, `maT4`, `L3dy`) for their constructive feedback and insightful suggestions. Your comments have significantly strengthened the motivation, empirical rigor, and clarity of our work. We have made every effort to faithfully address your comments in the responses and have uploaded the updated manuscript. As suggested by the reviewers, we summarize the key updates and new experiments included in this revision:
- **Clarified Motivation and Distinctness from General Tasks and PEFTs:** We supplement the **Section 1 & 2** to explicitly contrast weather tasks with general tasks and distinguish WeatherPEFT with existing PEFTs (Reviewers `hX2x`, `Nt4E`).
- **Additional Vision Experiment:** We add a vision domain generalization experiment in Table 4 of **Section 5.4** and results on the Visual Task Adaptation Benchmark (VTAB-1k) in Table 13 of **Appendix B.7** to validate the specificity of our performance gains to the weather domain (Reviewers `hX2x`, `Nt4E`, `maT4`, `L3dy`).
- **Additional Baseline Comparison:** We expand Table 3 in **Section 5.3** to include a wider suite of recommended PEFT baselines (Reviewers `hX2x`, `Nt4E`).
- **Additional Fine-grained Ablation Experiment:** We extend the fine-grained ablation to include larger-parameter settings in Table 9 of **Appendix B.3** to show the scalability of improvements (Reviewer `hX2x`).
- **Additional TADP Synergistic Analysis:** We add a comparison of WeatherPEFT against combinations of Generic PEFTs + SFAS in Table 11 of **Appendix B.5** to validate the specific architectural value of TADP (Review `Nt4E`).
- **Additional Computational Analysis:** We provide a quantitative analysis of the computational cost of our method in Table 12 of **Section B.6** to show the favorable trade-off (Reviewer `maT4`).

We have incorporated these suggested modifications in the revised version, which are highlighted in **blue**. We are grateful for the opportunity to improve our work. If you are satisfied with the revisions, we kindly request your approval to consider an improved score.

Thanks for all the reviewers' time again.

Best regards,

The Authors

---

### Author Response · Authors · 2025-12-01
**Summary of Rebuttal Discussion**

Dear Area Chairs, Senior Area Chairs, and Program Chairs,

We sincerely thank you for managing the review process. Due to the rollback to the pre-discussion state, we would like to briefly summarize the discussion so far.  Overall, we are pleased that the majority of engagement has been highly positive. Our paper initially received ratings of **8 (maT4), 6 (L3dy), 4 (Nt4E), and 0 (hX2x)**. Throughout the discussion period, we made every effort to systematically address reviewers' concerns and substantially improved the manuscript. As a result, the final ratings increased to **8 (maT4), 8 (L3dy), 4 (Nt4E), and 2 (hX2x), with an average score of 5.5**, before the reset.

### 1. Reviewer maT4 (Score: 8, Maintained)
Reviewer `maT4` consistently supported acceptance and emphasized the paper’s clarity, societal relevance, and strong results. Their main points concerned:
- The computational complexity of our method.
- A suggestion to explore additional downstream tasks.

In responses, we provided:
- Additional quantitative analysis on the computational cost of our method.
- Additional vision domain generalization experiments.

The reviewer acknowledged our response and confirmed they would **maintain their high score**.

### 2. Reviewer L3dy (Score:6 → 8, Raised)
Reviewer `L3dy` initially raised concerns about:
- Whether WeatherPEFT is specific to weather tasks, rather than a generic PEFT variant.
- The resolution sensitivity of our method.

We responded by providing:
- Additional vision domain generalization experiments to demonstrate the domain-specific design of WeathPEFT.
- Discussion on resolution sensitivity of WeathPEFT.

On this basis, the reviewer  **raised their score from 6 to 8**. They subsequently suggested evaluating on a more generic vision benchmark, and we promptly added a VTAB-1k experiment and reported the results. Due to the timing of the freeze, there was no further back-and-forth, but their concerns were directly addressed in our revisions.

### 3. Reviewer Nt4E (Score: 4)
Reviewer `Nt4E` highlighted the TADP module and our applications of PEFT to weather as interesting. Their primary concerns were:
- Difference between weather and general domains
- Specificity of our method to the weather domain.
- The novelty of SFAS and the impact of TADP.
- Outdated baselines.

We addressed these points by providing:
- Explanation of the specificity of weather domain.
- Additional vision domain generalization experiments to demonstrate the domain-specific design of WeathPEFT.
- Classification of the novelty of SFAS and additional synergistic analysis to validate the value of TADP.
- Comparisons on recommended baseline.

However, **we do not have the opportunity to receive further follow-up** from the reviewer before the freeze. We respectfully hope **these responses can be evaluated on their merits**, as they directly address the reviewer's concerns regarding motivation, novelty, and baselines.

### 4. Reviewer hX2x (Score:0 → 2, Raised)
Reviewer `hX2x` initially gave an extreme score of 0 while simultaneously noting that the paper is well written and the experiments are extensive and comprehensive, and the appendix is clear and solid. Their main objections focused on
- An allegedly incorrect claiming of motivation and related-work statement.
- Specificity of weather tasks from vision tasks.
- Outdated baselines.
- Novelty of SFAS.
- Small gains of fine-grained ablation study.

In our responses, we addressed these by providing:
- Classification of the wording of our motivation and explicit discussion on existing task-specific PEFTs.
- Explanation on the specificity of weather domain and additional vision domain generalization experiments to demonstrate the domain-specific design of WeathPEFT.
- Comparisons on recommended baseline.
- Classification on the novelty of SFAS.
- Extension of fine-grained ablation experiment on larger-parameter settings to show the scalability of improvements.

In the follow-up comments (posted under Reviewer `Nt4E`’s thread), Reviewer `hX2x` stated that they accept our clarifications on the wording, fully agree with the new experimental results, and view our dynamic, annealed selection as a reasonable extension, and raised their score from 0 to 2.
However, the reviewer maintained their rejection based on a subjective dismissal of the application domain, stating `"I don't buy into such a method to do the research"`, characterizing the contribution as merely an `"application-oriented adapter variant"` rather than `"a fundamentally new PEFT paradigm"`.

In response to these follow-up comments, we further explained the role of application-driven innovation in the “applications to physical sciences” track and why domain-specific adaptation is considered a valid form of contribution, reiterated the unique physical challenges of weather tasks, and highlighted the holistic design and novelty of our method.

Best regards,

The Authors

---

### Meta-Review · Area_Chair_87vk · 2026-01-06

**Summary:**

The initial concerns focused on four main areas:

- Baselines: Reviewers Nt4E and hX2x felt the initial baselines were outdated or insufficient, requesting comparisons against more baselines.

- Domain Specificity vs. Generality: Reviewers L3dy, Nt4E, and hX2x questioned whether the method was truly specific to weather or just a generic adapter applied to a new domain. They requested evidence of how it performs on general vision tasks to isolate the source of improvement.

- Novelty: Reviewer hX2x and Nt4E questioned the novelty of SFAS compared to existing Fisher-based selection methods (like Child-Tuning).

- Computational Cost: Reviewer maT4 queried the added complexity of the proposed modules.

**Reviewer Concerns:**

Addressed Concerns (or at least partially addressed):

- Baselines: The authors significantly expanded the experimental suite in the revision. They added comparisons against MoA, HydraLoRA, VeRA, SAM, SCT, Child-Tuning, and others. The results consistently showed WeatherPEFT outperforming these methods on weather tasks. I consider this concern fully addressed.

- Domain Specificity: The authors conducted new experiments on general vision benchmarks (Cityscapes and VTAB-1k). The results showed that WeatherPEFT performs comparably to generic adapters on vision tasks but significantly better on weather tasks. This effectively supports the authors' claim that their architecture captures meteorological inductive biases (heterogeneous variables, physical resolution dependence) rather than just being a "better generic adapter."

- Computational Cost: The authors provided a training time analysis showing a modest ~10% overhead compared to LoRA.

Outstanding Concerns:

- Novelty/Philosophy: Reviewer hX2x maintained a rejection (raising the score only to 2), arguing that the method is an "application-oriented adapter variant" and does not constitute a "fundamentally new PEFT paradigm." While the authors clarified the differences (annealed dynamic selection vs. static masks), the reviewer remained philosophically opposed to the "application-driven" nature of the contribution. However, given the submission is to the "Applications to Physical Sciences" track, tailoring methods to respect physical constraints is a valid and valuable contribution, so this outstanding concern should be weighed lightly against the empirical success.

**Reviewer Scores:**

- Reviewer maT4 (Current: 8): This reviewer was satisfied with the original paper and the clarifications on computational cost. Score would remain 8.

- Reviewer L3dy (Current: 8): (according to the author) This reviewer raised their score from 6 to 8 during the rebuttal after the authors addressed the resolution sensitivity and domain specificity questions. Score would remain 8.

- Reviewer Nt4E (Current: 4): This reviewer did not respond to the extensive new experiments (new baselines and vision benchmarks) before the freeze. Since their primary criticisms were the lack of modern baselines and questions on domain generality—both of which were thoroughly addressed with positive results—it is highly probable they would have raised their score. Estimated Adjusted Score: 5 or 6.

- Reviewer hX2x (Current: 2): This reviewer raised their score from 0 to 2 after acknowledging the new experiments but insisted on rejection due to a belief that the method is not novel enough. It is unlikely they would have moved significantly higher given their philosophical stance on domain-specific PEFTs.

---

### Decision · Program_Chairs · 2026-01-26

Accept (Poster)